# The Insulin-like Growth Factor Family as a Potential Peripheral Biomarker in Psychiatric Disorders: A Systematic Review

**DOI:** 10.3390/ijms26062561

**Published:** 2025-03-12

**Authors:** Carlos Fernández-Pereira, Roberto Carlos Agís-Balboa

**Affiliations:** 1Neuro Epigenetics Lab, Health Research Institute of Santiago de Compostela (IDIS), Santiago University Hospital Complex, 15706 Santiago de Compostela, Spain; carlosfernandezpereira@gmail.com; 2Translational Research in Neurological Diseases (ITEN) Group, Health Research Institute of Santiago de Compostela (IDIS), Santiago University Hospital Complex, SERGAS-USC, 15706 Santiago de Compostela, Spain; 3Neurology Service, Santiago University Hospital Complex, 15706 Santiago de Compostela, Spain

**Keywords:** IGF-1, IGF-2, IGFBP, peripheral, schizophrenia, major depressive disorder, bipolar disorder, autism spectrum disorder, obsessive–compulsive disorder, attention-deficit hyperactive disorder

## Abstract

Psychiatric disorders (PDs), including schizophrenia (SZ), major depressive disorder (MDD), bipolar disorder (BD), autism spectrum disorder (ASD), among other disorders, represent a significant global health burden. Despite advancements in understanding their biological mechanisms, there is still no reliable objective and reliable biomarker; therefore, diagnosis remains largely reliant on subjective clinical assessments. Peripheral biomarkers in plasma or serum are interesting due to their accessibility, low cost, and potential to reflect central nervous system processes. Among these, the insulin-like growth factor (IGF) family, IGF-1, IGF-2, and IGF-binding proteins (IGFBPs), has gained attention for its roles in neuroplasticity, cognition, and neuroprotection, as well as for their capability to cross the blood–brain barrier. This review evaluates the evidence for IGF family alterations in PDs, with special focus on SZ, MDD, and BD, while also addressing other PDs covering almost 40 years of history. In SZ patients, IGF-1 alterations have been linked to metabolic dysregulation, treatment response, and hypothalamic–pituitary–adrenal axis dysfunction. In MDD patients, IGF-1 appears to compensate for impaired neurogenesis, although findings are inconsistent. Emerging studies on IGF-2 and IGFBPs suggest potential roles across PDs. While promising, heterogeneity among studies and methodological limitations highlights the need for further research to validate IGFs as reliable psychiatric biomarkers.

## 1. Introduction

### 1.1. Psychiatric Disorders and the Need for Peripheral Biomarkers

Psychiatric disorders (PDs) represent a significant global health burden, affecting millions of individuals and their families worldwide. These mental conditions often result in chronic disability, diminishing the quality of life, and creating substantial societal costs [1]. Despite recent advancements in the understanding of their underlying biological mechanisms, the diagnosis of PDs still remains a subjective clinical assessment based on observed symptoms, which can vary considerably among individuals and be very unspecific [2]. The actual paradigm therefore has several limitations, including delayed or inaccurate identification of PDs and difficulties in predicting treatment responses. In this context, there is an urgent need for objective and reliable biomarkers, measurable indicators of biological processes, or pathophysiological conditions that can help psychiatrists not just in the process of diagnosis but in the monitoring of clinical progression and in the evaluation of treatment response [3]. In recent decades, the search for biomarkers in psychiatry has gained attention, focusing on molecular, genetic, and imaging approaches. However, peripheral biomarkers in plasma or serum are particularly attractive due to their less invasive, cheaper, and faster accessibility and potential to reflect central nervous system (CNS) processes [4]. In this framework, the insulin-like growth factor (IGF) family, including IGF-1, IGF-2, and their binding proteins (IGFBPs), has emerged as a potential candidate for biomarker research in PDs, mainly for its implications in neuroplasticity and cognition [5].

### 1.2. Rationale and Objectives

The aim of our review is to synthesize all original research available on the peripheral IGF system in psychiatric disorders (SZ, MDD, BD, BPD, OCD, ASD, and ADHD) with two main objectives. First, to cover the rationale behind these studies. That is why these studies were performed and what they were trying to test. Second, to consider the potential of the IGF system as a biomarker for PDs. Additionally, but to a lesser extent, there is a third objective: to cover articles that have measured IGF peripheral members and depressive symptoms without a proper MDD diagnosis, such as in population-based studies or in concomitant diseases including fibromyalgia or Parkinson’s disease. We believe these are also relevant approaches to ponder the role of the IGF system in PDs.

### 1.3. The Insulin-like Growth Factor (IGF) Family

The insulin-like growth factor (IGF) family is mainly formed by three components: ligands (IGF-1 and IGF-2), transmembrane surface receptors (IGF-1R and IGF-2R), and IGF binding proteins (IGFBP-1 to 6) [6]. Some authors also consider other proteins which are directly or indirectly related to the IGF family. First, the so-called “IGFBP-related proteins (IGFBPs-rP)” that include IGFBP-7, IGFBP-8, and IGFBP-9 [7]. Second, there are also IGFBPs proteases, like pregnancy-associated plasma protein-A (PAPP-A), which selectively cleaves the IGF ligand from IGFBPs, regulating the amount of free or bioactive IGFs that can bind to IGF receptors [8].

#### 1.3.1. IGF-1

IGF-1 is a 70 amino-acid peptide hormone produced in the liver in response to growth hormone (GH) stimulation in what is called the hypothalamic–pituitary–somatotropic (HPS) axis [9,10]. The HPS axis begins at the hypothalamic level, where the GH releasing hormone (GHRH) stimulates GH release from the pituitary to circulation. The opposite action would be exerted by somatostatin (SST). The GH then circulates bound to the GH binding protein (GHBP) until it reaches the GH receptor (GHR) at the liver, which in turns activates IGF-1 production and release into circulation from hepatocytes (Figure 1A) [11].

IGF-1 acts by binding to the IGF-1 receptor (IGF-1R), triggering intracellular pathways that promote growth, survival, and differentiation of cells (Figure 1B). It has a profound impact on neuroplasticity, including synaptogenesis and myelination, and plays a role in learning, memory, and cognitive function. IGF-1 exerts its actions in target tissues, having different functions, as indicated in Figure 1C [12].

#### 1.3.2. IGF-2

IGF-2 is a 67 amino-acid hormone peptide [13] produced by almost all cell types and especially the brain. Moreover, IGF-2 is the most expressed IGF ligand in the adult CNS. During brain development, it is mainly expressed across the choroid plexus, leptomeninges, and the hypothalamus [14]. Therefore, IGF-2 plays important roles during fetal growth and development, when IGF-2 concentration is most elevated [15]. The genetic coding of IGF-2 is complex, since the *IGF-2* gene is an imprinted gene that expresses from the paternal allele in most human tissues, only maternally expressed in the brain during adulthood, as was shown in rodent models [16]. In addition to triggering IGF-1R signaling with less binding affinity than IGF-1 (Figure 1B), IGF-2 also binds IGF-2R, a single transmembrane glycoprotein, by which the main function appears to be the internalization of IGF-2 into lysosomes for further degradation [17]. Nonetheless, IGF-2 has been recently postulated as a potential memory enhancer, playing an essential role in memory consolidation, synaptic stability, and neuroprotection [18].

#### 1.3.3. IGFBPs

The action of IGFs is tightly regulated by IGFBPs (1 to 6), a group of six proteins that bind to IGF-1 and IGF-2 in circulation (Figure 1B). These binding proteins modulate the availability of IGFs to their receptors, prolonging their half-life, and enhancing or inhibiting their actions [19]. IGFBP-3 is the most concentrated in the adult bloodstream with estimations of 100 nM/L, whereas the other IGFBPs are in the range of less than 20 nM/L [20]. Moreover, 75–80% of the peripheral IGF ligands form ternary complexes with IGFBP-3 or IGFBP-5 and a glycoprotein termed acid labile subunit (ALS). This way, IGFs can extend their half-life in circulation from 10–12 min if circulating unbound for almost 14–15 h. The other 20–25% of the peripheral IGFs form binary complexes with the remaining IGFBPs [21]. IGFBP-7 shows 100 times less binding affinity than IGFBP-3 for IGFs, probably due to its one conserved cysteine at the C-domain [22]. Moreover, IGFBPs also have IGF-independent roles in cellular signaling, affecting processes like cell migration and inflammation [23] that could be indirectly related to PDs.

## 2. Searching Strategy

We followed the PRISMA methodology to search, filter, and select the articles. The criterion for eligibility was initially set as “articles that measured IGF peripheral members in humans with an official diagnosis of a psychiatric disorder (SZ, MDD, BD, BPD, OCD, ASD or ADHD)”. Having this in mind, we first conducted a systematic search in the PubMed database by including the complete name of each PD: “Schizophrenia”, “Depression”, “Bipolar Disorder”, “Borderline Personality Disorder”, “Obsessive Compulsive Disorder”, “Autism Spectrum Disorder”, and “Attention Deficit and Hyperactivity Disorder”, followed by insulin-like growth factor “IGF”. We then identified the entries counting to 1061 articles. After, we deleted 81 duplicates by using an Excel sheet. A high number of duplicates was expected, since many articles include keywords related to multiple PDs. Moreover, many studies assess one or more PD when evaluating potential peripheral biomarkers, precisely because there is some degree of symptomatic overlap in certain cases. We then screened 980 articles by using *abstrackr* software (http://abstrackr.cebm.brown.edu/account/login, accessed on 29 December 2024) and excluded 861 that were not related to the subject, including only animal models, in vitro models, or gene expression. Then, 119 articles were sought for retrieval and assessed for eligibility. We then excluded articles for five reasons. Reason 1: “Meta-analysis”. Reason 2: “Articles that measured IGF peripheral members in the non-psychiatric offspring of patients with a diagnosis of PD”. Reason 3: “articles that measured IGF either in CSF or brain expression in patients with a diagnosis of PD”. Reason 4: “Articles that measured peripheral IGF members and depressive symptoms but without a proper psychiatric diagnosis”. These articles mainly included two types. The first type was population-based studies, measuring the relation between depressive symptoms and peripheral IGF-1 without a comorbid disease but a studying condition such as ageing or exercise. The second type included patients with a major concomitant disease, such as fibromyalgia or Parkinson’s disease. Reason 5: “Articles that openly studied IGF in ASD with specifiers associated with known single-gene mutations such as Rett syndrome, Phelan–McDermid disorder or Fragile X syndrome”.

Therefore, although articles from Reasons 1 to 4 were excluded in the main PRISMA flowchart, we briefly addressed them as part of objective 3 (see Section 1.2). Moreover, after extensive reading, we decided to include articles that were not initially found in the systematic search (*n* = 64). These records were identified in the PubMed database (*n* = 22) and were found referenced in the articles that were initially found in the systematic search. Two of these articles were included in Reason 3, since they measured IGF members in CSF [24,25]. What is more, we included an article published in a non-indexed journal [26]. The searching process ended 24 November 2024. The PRISMA flowchart can be checked in Figure 2. All modifications to the standard searching strategy were made to improve the exposition of our work.

## 3. Results

Specific data related to each searching process can be checked in Appendix A. The original articles mainly considered in this review can be checked for each PD in Appendix A. For SZ (Appendix A), MDD (Appendix A), BD (Appendix A), OCD and BPD (Appendix A), ASD (Appendix A), and ADHD (Appendix A).

## 4. IGF Peripheral Levels in Schizophrenia Patients

### 4.1. Schizophrenia (SZ)

Schizophrenia (SZ) is a multifactorial chronic PD with a prevalence of around 1% of the worldwide population that has been dramatically increasing in almost 70% of cases from 1990 [28]. SZ is characterized by so-called positive symptoms that mainly consist of hallucinations, paranoid delusions, or disorganized speech, as well as negative symptoms, like decreased motivation or emotional expression, and a third type of general symptoms related to cognition, perception, and general well-being [29]. The clinical course for SZ begins with a first psychotic episode (FE) when symptom severity surpasses a certain threshold, and after suffering from two or more psychotic episodes, termed multiple episodes (ME) [30]. Pharmacological treatment is the gold standard in SZ therapy and includes several types of antipsychotics (Aps), such as haloperidol, or the atypical antipsychotics (A-Aps), but generally also includes other type of drugs, such as antidepressants (Ads), mood stabilizers (ms), or benzodiazepines [31]. Medication only offers a palliative for acute symptomatology in approximately 40–60% of SZ patients. In this sense, pharmacological treatment does not cure SZ, and patients who do not respond satisfactorily are known as treatment-resistant [32]. The gold standard to evaluate symptom severity in SZ patients is the Positive and Negative Syndrome Scale (PANSS), which is divided into three subscales designated for the evaluation of the positive (PANSS-P), negative (PANSS-N), and general (PANSS-G) symptoms. As a way to estimate the overall symptomatology, the final total score (PANSS-T) is considered [33]. In general terms, SZ patients who have experienced a reduction of 50% in the PANSS-T scale from baseline values, are adequately responding to treatment, and therefore are called treatment-responders [34].

### 4.2. IGF-1 in SZ Patients

#### 4.2.1. First Studies and the Potential Influence of Antipsychotics in Weight Gain

The first set of studies evaluated serum levels of IGF-1 and IGFBP-1 as potential markers that could be behind the observed weight gain in SZ patients treated with clozapine or olanzapine [35,36,37,38,39]. These two A-Aps considerably improved side effects, such as extrapyramidal symptoms, but were apparently inducing some metabolic abnormalities. IGF-1 is known for the uptake of glucose and amino acids into muscle, whereas IGFBP-1 is used as an indirect measure of insulin secretion, since IGFBP-1 synthesis in the liver is inhibited by insulin. In their first study, Melkersson et al. found that the median levels of IGF-1 were significantly decreased in clozapine-treated SZ patients compared to SZ patients treated with classical Aps, while IGFBP-1 was unaltered. Nonetheless, IGFBP-1 levels were below what they considered “normal” ranges [35]. The first hypothesis was that reduced IGF-1 levels could be indicating a GH deficiency, which had been previously linked to weight gain. IGFBP-1 was inversely correlated with insulin and body mass index (BMI) in the group treated with classical Aps but not in the clozapine group. Moreover, IGFBP-1 significantly correlated with IGF-1, but just in the clozapine group. Nevertheless, no significant correlation was found between serum concentrations of IGFBP-1 and clozapine [35]. In olanzapine-treated SZ patients, IGF-1 was found to be within “normal ranges”. Therefore, weight gain in the olanzapine group was not linked to a GH deficiency. Conversely, IGFBP-1 was significantly decreased in olanzapine-treated SZ patients. However, no statistical correlation was found between IGFBP-1 levels with insulin, leptin, triglycerides, cholesterol, duration of treatment, or olanzapine, but was still inversely correlated with BMI [36]. The effects of long-term treatment with classical Aps showed no alteration in IGF-1 levels in SZ patients. Moreover, IGF-1 was similar across patients treated with different classical APs, such as perphenazine, zuclopenthixol, haloperidol, thioridazine, and remoxipride, and no significant association was found with prolactin or BMI [37]. Finally, the serum concentrations of clozapine or olanzapine were not associated with IGF-1 and IGFBP-1, which were within the “normal range” for SZ patients. Curiously, no correlation was found between insulin and IGF-1 in each treated group, whereas IGFBP-1 was inversely correlated but just in the clozapine-treated group. Nevertheless, IGFBP-1 was below the median levels for most SZ patients, but no significant difference was found between the clozapine and olanzapine groups [38]. Therefore, Melkersson et al. concluded that it could be probable that insulin resistance might have been induced by clozapine but not olanzapine. These studies had to face some limitations, such as no control group or longitudinal measures, which we believe are the reasons why they have never been added to any meta-analysis.

Howes et al. conducted the first longitudinal research evaluating IGF-1, IGFBP-1, and GH levels in SZ patients, both at baseline and after 2.5 months of clozapine treatment [39]. The theoretical basis was similar, since IGF-1 was introduced as a glucoregulatory factor that would act similar to insulin by reducing peripheral glucose levels. It was stated that low levels of IGF-1 would indicate higher glucose levels, so a subsequent loss of control in glucose homeostasis could finally lead to weight gain. Nevertheless, no significant difference was found in the levels of IGF-1, IGFBP-1, or GH before and after treatment with clozapine [39]. Moreover, no significant correlation was found between changes in the BMI or clozapine serum levels with both IGF-1 and IGFBP-1. Therefore, it was concluded that clozapine would not decrease GH secretion leading to weight gain or glucose imbalance. Howes et al. made an interesting statement suggesting that lower IGFBP-1 at baseline could be either explained by a pre-existing elevation in insulin, perhaps related to prior Ap intake, or because of an intrinsic pathophysiological process related to SZ [39]. In this sense, IGFBP-1 alterations in SZ patients were proposed as a possible inner alteration linked to SZ, rather than as a side effect from Aps.

#### 4.2.2. IGF-1 Deficiency Hypothesis in SZ Patients

In 2004, Gunnell and Holly wrote a letter to the journal, *Schizophrenia Research*, proposing a hypothesis on how “deficient IGF-1 levels” would increase the susceptibility to SZ diagnosis. The lower levels of IGF-1 would directly affect different cell processes, such as apoptosis, neurogenesis, myelinization, synaptogenesis, or dendritic branching during brain development, or indirectly by reducing brain capacity to respond to environmental perturbations, such as meningitis or fetal hypoxia in the infant years. This hypothesis can be fully read in the original review [40].

Several studies have since tested this hypothesis (Figure 3A), with some results in favor and others in contradiction (Figure 3B). Researchers have been referring to this hypothesis and adding layers, suggesting potential pathophysiological mechanisms that could also be altering the IGF peripheral system in SZ patients, such as cortisol, insulin-glucose metabolic dysregulations, dyslipidemia, inflammation, and cognitive alterations (Figure 3C). Data from these papers can be checked in Table 1.

#### 4.2.3. IGF-1 and Metabolic Dysregulations in the Insulin–Glucose Homeostasis in SZ Patients

SZ has been associated with an increased risk of insulin resistance, glucose intolerance, and type 2 diabetes. Previous authors have considered it a possible anomaly derived from the use of Aps [35,39]. Two lines of evidence suggest that those alterations may not be entirely related to Aps. First, there is evidence of insulin resistance in SZ patients before the era of Aps [41]. Second, first episode (FE) SZ patients who were drug-naïve showed insulin resistance and glucose intolerance when compared to controls [42]. Therefore, IGF-1 alteration could be inherently related to SZ. IGF-1 is essential for optimal sensitivity to insulin, and deficient IGF-1 levels may lead to insulin resistance [43]. Following this rationale, Venkatasubramanian et al. proposed that lower IGF-1 levels and higher insulin resistance could be expected in SZ patients, regardless of Aps influence [44]. Supporting this, IGF-1 plasma levels were significantly reduced in FE drug-naïve SZ patients when compared to controls. Insulin and cortisol as well as insulin resistance were all found significantly elevated in SZ patients. What is more, a significant and inverse correlation was found between IGF-1 and insulin levels. Nonetheless, FE SZ patients had normal concentrations of glucose, cholesterol, and triglycerides. Interestingly, IGF-1 significantly and negatively correlated with the Scale for the Assessment of Positive Symptoms (SAPS) for hallucinations [44]. As a consequence, it has been suggested that the hippocampus would be more susceptible to IGF-1 deficits, since it is highly concentrated in IGF-1R [45] and had been related to auditory hallucinations [46].

In another study, Wu et al. measured glucose–insulin parameters, lipidic profile, cortisol, GH, IGF-1, and IGFBP-3 serum levels in three groups of obese patients with a diagnosis of SZ, non-SZ obese subjects, and non-obese controls [47]. IGF-1 was found to be significantly decreased in the SZ obese group of patients when compared to both obese subjects and controls, but no significant difference was found in the last two groups. IGF-1 decrease was not explained by the GH reduction, since the three groups had no significant alterations despite GH being slightly reduced in both obese groups. Among glucose–insulin parameters, obese patients with SZ had significantly higher fasting insulin and the Homeostatic Model Assessment of Insulin Resistance (HOMA-IR) index levels, whereas glucose was lower. Cortisol levels were unaltered between the three groups. Regarding the lipid profile, both obese groups had similar levels of triglycerides, high-density lipoprotein (HDL), and leptin, whereas the SZ group had significantly lower levels of cholesterol and low-density lipoprotein (LDL). Therefore, SZ diagnosis was apparently behind the significant lowering of IGF-1. On the other hand, no significant difference was found for IGFBP-3. Even though, the molar ratio of IGF-1–IGFBP-3 was significantly reduced in the SZ obese group of patients when compared with the other two groups. At the same time, a significant reduction was also observed in the obese group compared to controls. So, this phenomenon could be initially related to obese conditions but could be worsened by SZ diagnosis, either because of intrinsic mechanisms or because of Aps exposure [47]. Briefly, the IGF-1–IGFBP-3 ratio was used to estimate the free portion of IGF-1 and the portion bound to IGFBP-3. A reduction in the IGF-1–IGFBP-3 ratio would indicate a less bioactive IGF-1, usually investigated in GH-related disorders [48].

Another study explored the question of whether alterations in glucose metabolism were a side effect of Aps or an intrinsic mechanism of SZ. To this end, plasma levels of glucose, insulin, IGF-1, GH, leptin, and cortisol were measured in the offspring of SZ patients. Interestingly, IGF-1 was significantly reduced in the offspring of SZ patients compared to controls, whereas insulin and the HOMA-IR index were found to be significantly increased. Glucose, cortisol, GH, and leptin levels were found to be non-significant between both groups [49].

Demirel et al. studied IGF-1 based on the high prevalence (20–60%) of metabolic syndrome (MS) in SZ patients, which could be a side effect of Ap use or part of the inherent etiopathology of SZ [50]. The characteristics for positive MS in SZ patients were defined according to the National Cholesterol Education Program Adult Treatment Panel III-A (NCEP ATP III-A) criteria [51]. In spite of a higher prevalence for MS in SZ patients, no significant difference was found for glucose, insulin, HOMA-IR index, and IGF-1 serum levels. Only BMI and waist circumference (WC) were found to be significantly increased in SZ patients compared to controls. Importantly, SZ patients were taking Aps and were almost obese (29.77 ± 5.38 kg/m^2^), while controls were within the normal weight range (25.74 ± 4.56 kg/m^2^). IGF-1 was negatively correlated with triglycerides in the control group, and positively associated with HDL levels in SZ patients. Therefore, Demirel et al. proposed that IGF-1 may have a lipolytic effect and that may be interesting in case of high odds for MS in SZ patients [50].

#### 4.2.4. IGF-1 and the Hypothalamic–Pituitary–Adrenal (HPA) Axis in SZ Patients

In 2010, a new study found that serum IGF-1 was significantly reduced in drug-naïve SZ patients when compared to controls. What is more, IGF-1 was significantly increased in SZ patients after a period of 3 months with Aps. In this study, special attention was paid to cortisol, since hypercortisolemia, a common feature in SZ patients [52], was suggested to lead to a deficit in IGF-1 [53]. Moreover, hypercortisolemia could be reversed by Aps [54]. Therefore, Venkatasubramanian et al. suggested that returning cortisol to normal levels could be the reason behind the increase on IGF-1 levels with APs [55]. SZ patients had significantly higher cortisol levels before treatment and were statistically reduced after APs. What is more, the decrease in cortisol and the increase in IGF-1 levels with Aps was significantly correlated. Therefore, a new hypothesis was postulated stating that IGF-1 deficiency in SZ patients could be the secondary effect of an alteration in the HPA axis and that Ap treatment could restore IGF-1 through cortisol reductions. Again, the IGF-1 increase with Aps was significantly associated with the reduction in the positive symptomatology (SAPS), but not with negative symptoms (SANS). Curiously, changes in cortisol levels were not correlated with changes in both symptoms (SAPS or SANS) [55].

In 2023, Arinami et al. evaluated the previous cortisol hypothesis in treated SZ patients. In the meantime, the HPA axis was related to reduced neurogenesis and synaptic plasticity in the hippocampus, and two meta-analyses concluded that cortisol levels are significantly increased in SZ patients [56,57]. In this line, cortisol levels were found to be significantly higher in SZ patients in contrast to controls and MDD patients. Therefore, IGF-1 was found to be significantly elevated in both SZ and MDD patients in comparison to controls as was predicted to be a mechanism to compensate for central reduced neuroplasticity and myelination in both PDs. Age was significantly and negatively correlated with IGF-1, but not with cortisol levels in the SZ group. Regarding sex distribution, men had significantly higher cortisol levels than women, whereas IGF-1 was unaltered. Notably, no association was found between IGF-1 and cortisol. Differences with the study of Venkatasubramanian et al. [55] were age, BMI, and Ap treatment in SZ patients. So, Arinami et al. suggested that perhaps the correlation between cortisol and IGF-1 may just represent a state of response to Ap treatment. Finally, IGF-1 was negatively and significantly correlated with the Brief Psychiatric Rating Scale (BPRS) [58].

#### 4.2.5. IGF-1 and Inflammation in SZ Patients

There is plenty of evidence pointing to an overactive inflammatory system in SZ patients [59], in which several cytokines (TNF-α, IFN-γ, IL-2, IL-4, and IL-10) have been found to be increased in drug-naïve FE SZ patients [60], whereas contradictory findings have been found for SZ patients under treatment [61].

In this context, Petrikis et al. suggested an inflammatory-related hypothesis to explain significant elevation in serum IGF-1 in drug-naïve FE SZ patients [62]. Proinflammatory cytokines, which had been found to be significantly increased in SZ patients, such as TNF-α, IL-6, or IL-1β [63], could block IGF-1R binding, displacing IGF-1 and, as a consequence, increasing IGF-1 peripherally. Nonetheless, different premises were explored, such as the interrelation between cortisol, insulin, or cognitive decline and IGF-1 in SZ patients. IGF-1 increase could be indicating an incipient cognitive decline in SZ patients since previous studies have shown a significant increase in IGF-1 in SZ patients in early stages of Alzheimer’s (AD) [64] and Parkinson’s diseases [65]. Carro and Torres-Aleman suggested that IGF-1 elevation would be an early response in neurodegenerative diseases, but would gradually decrease with disease progression [66]. On the other hand, SZ patients had significantly higher fasting insulin, c-peptide, and HOMA-IR, but not fasting glucose or cortisol levels. Therefore, IGF-1 increase was independent of metabolic parameters (glucose, insulin, and HOMA-IR) or cortisol levels. No correlation was found for IGF-1 and symptom severity (PANSS) [62].

Conversely, Karanikas et al. found no significant alteration in IGF-1 serum levels in FE patients, despite higher TNF-α and IFN-γ levels [67]. IGF-1 had been found to be a potential regulator of lymphocyte T helper 17 (Th17) cells, which secret IL-17, a cytokine that stimulates target cells to release proinflammatory cytokines, the induction and the aggregation of neutrophils. Moreover, IL-17 deteriorates the integrity of the blood–brain barrier (BBB), which allows Th17 cells to cross the BBB, where they express granzyme B, a potent cytotoxin that leads to an inflammatory response in the CNS [68]. IGF-1 and IL-17 were significantly elevated in a cohort of drug-naïve FE SZ patients compared to controls; however, there was no significant correlation among them, nor were they correlated with age, sex distribution, or waist-to-hip (WHR) ratio. IGF-1 levels significantly increased after 10 weeks of risperidone monotherapy, while IL-17 was not changed. Interestingly, changes in IGF-1 levels were significantly associated with the reduction in negative symptoms (PANSS-N) [69]. The hypothesis of a significant IL-17 elevation in SZ patients is controversial, since some studies have found significant reductions [70,71], and the most recent study of SZ patients shows an IL-17 increase as well as being unaltered after treatment [72]. Therefore, it was proposed that IGF-1 could be a state marker of acute psychosis, since IGF-1 returns to normal values after Aps, whereas IL-17 would be a trait marker because it remains unchanged in SZ patients even after treatment. Nonetheless, there was a considerable difference in sample size between SZ patients (*n* = 113) and controls (*n* = 58) [69].

#### 4.2.6. IGF-1 in FE: Differences Between SZ and BD Patients

Palomino et al. studied IGF-1 plasma levels in SZ and BD patients at their FE (baseline) and after 1, 6, and 12 months of Ap treatment [73]. IGF-1 was not significantly altered between FE patients (SZ + BD) and controls. However, a portion of SZ patients had significantly higher IGF-1 than BD patients in the FE. After 1 month of Aps, SZ patients had significantly elevated IGF-1 compared to controls, but not after 6 or 12 months. Curiously, IGF-1 levels in FE patients significantly and negatively correlated with tobacco, but not with alcohol or cannabis. Moreover, IGF-1 was found to be positively associated with negative symptoms (PANSS-N), both at baseline and after 12 months. IGF-1 and PANSS-G only correlated at baseline. However, the sample size used after 12 months was almost half (*n* = 27) in comparison to baseline (*n* = 50) [73]. It was suggested that IGF-1 increase might indicate a cognitive decline based on recent reports in Huntington’s disease [74]. Conversely, IGF-1 was thought to decrease in elderly people in association with cognitive decline [75]. Finally, increased peripheral IGF-1 levels could be a response mechanism to compensate for a structural deficit in myelinization in SZ patients [73,76]. In summary, the exposure of 1 month to Aps elevated IGF-1 in SZ patients but not in BD patients [73]. Therefore, Palomino et al. studied FE SZ patients for the first time and their results were not in accordance to the Gunnell and Holly hypothesis [40].

#### 4.2.7. IGF-1 and Cognitive Alterations in SZ Patients

Chao et al. studied the relation between IGF-1 serum levels and cognitive performance in SZ patients [77]. First, IGF-1 was found to be significantly reduced in SZ patients when compared to controls. Second, SZ patients exhibited worse cognitive performance than controls. Third, the executive function and attention were positively associated with lower IGF- l. SZ patients entered the study after a 3-month wash-out period from Aps [77]. They suggested that an elevation of dopamine 2 receptors in the striatum could have led to a significant reduction in *IGF-1* gene expression [78]. IGF-1 deficits also correlated with impaired cognition in other conditions, such as GH deficiency [79], Parkinson’s [80], or even in normal aging [81]. Therefore, Chao et al. suggested that reductions in IGF-1 levels could be contributing to the observed cognitive deficits in SZ patients through the downregulation of hippocampal neurogenesis and synaptic plasticity [77].

#### 4.2.8. IGF-1 in Remitted, Treatment Resistant, and Chronic SZ Patients

Remitted SZ patients had significantly reduced IGF-1 plasma levels when compared to both treatment-resistant SZ patients and controls. Moreover, no significant difference was found for insulin, GH, HDL, total cholesterol, triglycerides, or the HOMA-IR index between the three groups. Glucose levels were significantly higher in SZ patients than controls, while plasma LDL in the remitted group was significantly higher in contrast to resistant-patients. The educational level was reduced in SZ patients compared to controls, and the treatment-resistant group had a higher incidence of smoking and a lower working status and age of SZ onset than remitters. Nonetheless, in terms of chlorpromazine equivalent dosages, there was no difference between SZ groups. IGF-1 positively and significantly correlated with all subscales, but with positive symptoms (PANSS-P) and with the Global Assessment Scale (GAS), which was negatively associated with IGF-1 levels. In summary, parameters predicting treatment resistance in SZ patients were IGF-1, LDL, and age at disease onset; whereas glucose, insulin resistance, or sex did not affect the predictivity of treatment resistant [82].

On the contrary, no alteration in IGF-1 serum levels was found in chronic SZ patients when compared to controls. These patients were receiving multiple Aps, and chronicity was defined as a diagnosis for more than 10 years. No correlation was found between BMI and IGF-1, while IGF-1 was significantly and negatively correlated with age in the whole sample. In a similar tendency, IGF-1 was found to be significantly correlated with both PANSS-T and PANSS-G. Moreover, years of illness and diabetes mellitus incidence were not related to IGF-1 levels [83].

#### 4.2.9. IGF-1 Meta-Analysis in SZ Patients

In 2023, Pejcic et al. conducted the first meta-analysis involving 12 studies [39,44,47,50,55,67,69,73,77,82,83,84], 596 SZ patients, and 422 controls. Nonetheless, no significant difference was concluded for IGF-1 levels between SZ patients and controls. Moreover, no differences were found in drug-naïve or drug-free SZ patients and controls. However, patients that were already under treatment showed significantly lower IGF-1 levels than controls, although these results showed no robustness in the sensitivity analysis. These findings were in contradiction with the IGF-1 deficiency hypothesis [40]. Nonetheless, the authors themselves suggested that the interpretation of their analysis should be taken cautiously since the number of studies per analysis was small and there was a marked heterogenicity among them [85]. In 2024, another meta-analysis of 14 studies included 863 SZ patients and 750 controls, and revealed that IGF-1 levels were significantly lower in SZ patients when compared to controls, supporting the deficiency hypothesis of IGF-1 in SZ patients. Moreover, IGF-1 levels increased after Aps [86]. Unfortunately, we did not have access to the Teja dissertation [84] or the studies included in the last meta-analysis [86].

### 4.3. IGF-2 in SZ Patients

#### 4.3.1. IGF-2 First Study and the Atherogenic Profile in SZ Patients

In 2007, Akanji et al. measured serum levels of both IGF ligands and IGFBP-3 in SZ patients, which were under stable therapy with haloperidol [87]. First, no significant difference was found for IGF-1 and IGFBP-3 levels. Nonetheless, IGF-2 was found to be significantly elevated in SZ patients compared to controls. Several parameters, such as total cholesterol, HDL, apolipoprotein B, and uric acid, were significantly increased in SZ patients. IGF-1 and IGF-2 levels did not significantly correlate with parameters like age, BMI, or WHR. Curiously, IGF-2 was found to be unaltered in SZ patients after repeating the statistical analysis stratifying by BMI (>25 vs. ≤25 kg/m^2^). IGF-2 was significantly and positively correlated with total cholesterol, LDL, and IGFBP-3 levels. IGF-2 was linked to the fetal origin of SZ due to its important functions during the uterine phase, and were related to the atherogenic profile. Nevertheless, this study was composed only by male subjects and lacked status on symptom severity. Noteworthily, Akanji et al. pointed out that the study of IGF-2 is limited in rodents, since it is undetectable in adult circulation; whereas, in humans, it remains considerably high in comparison to IGF-1 [87].

#### 4.3.2. IGF-2 and Cognition in SZ Patients

The first study evaluating the role of IGF-2 in SZ patients, without IGF-1, was published in 2020. The cohort was formed by SZ patients who were either drug-naïve or washed-out from Aps for at least 3 months. Serum IGF-2, IGFBP-3, and IGFBP-7 were found to be significantly reduced in SZ patients compared to controls. Moreover, IGF-2 significantly and negatively correlated with negative symptoms (PANSS-N), whereas no significant difference was found for IGF-2 between males and females in both SZ patients and controls. Nonetheless, no significant correlation was found between IGF-2, IGFBP-3, and IGFBP-7 with age, BMI, age at onset, or illness duration. Importantly, IGF-2 was found to be positively associated with working memory, attention, and executive function, but not with other cognitive domains, such as speed processing, visual memory, or verbal learning; whereas IGFBP-3 and IGFBP-7 were not correlated with symptom severity or cognitive tests [88].

#### 4.3.3. IGF-2 Elevation Through Ap Actions over C/EBPβ

In further studies, IGF-2 was found to be significantly reduced in SZ patients when compared to controls, even after correcting by sex, age, BMI, age at onset, and duration of illness. IGF-2 levels were significantly increased after 8 weeks of Ap monotherapy (risperidone, aripiprazole, olanzapine, or clozapine). Moreover, the increase in serum IGF-2 was significantly correlated with a reduction in the negative symptomatology (PANSS-N) [89]. The transcription factor CCAAT enhancer binding protein β (C/EBPβ) mRNA levels were found to be reduced in SZ patients [90], and was also proved to regulate *IGF-2* gene expression [91]. Therefore, Chao et al. proposed that Ap effects on C/EBPβ could be a potential mechanism behind IGF-2 elevation in SZ patients [89]. To the best of our knowledge, this hypothesis has not been tested yet.

#### 4.3.4. IGF-2, IGFBP-3, and IGFBP-7 in FE SZ Patients

In a previous study, our group found for the first time that IGF-2 plasma levels, but not IGFBP-7, were significantly decreased in drug-naïve FE SZ patients when compared to controls. After a short period with A-Aps (aripiprazole, olanzapine, or risperidone), both IGF-2 and IGFBP-7 were significantly increased. On the other hand, IGF-2 and IGFBP-7 were significantly elevated in ME SZ patients who have been treated for years, in contrast to controls and FE patients. Interestingly, we did find a significant and positive correlation between the arithmetical difference on IGF-2 levels before and after treatment in the FE group and the percentage of reduction in positive symptoms (PANSS-P). In the ME group, IGFBP-7 levels were found to be positively and significantly associated with both negative symptoms (PANSS-N) and total scores in the Self-Assessment Anhedonia Scale (SAAS). Finally, no correlations were found between IGF-2 or IGFBP-7 with age, sex, albumin, glucose, or triglycerides. Nonetheless, IGF-2 negatively correlated with total cholesterol levels in FEs but not in MEs [92].

We support Yang et al. [88] in their suggestions that IGF-2 differences among these studies could be a matter of a dissimilar sample size, Ap treatment, source of measure, or ethnicity. There is no meta-analysis available yet on IGF-2, probably due to the small number of studies.

### 4.4. Other IGFBPs Studies in SZ Patients

#### 4.4.1. IGFBP-3 and Exercise in SZ Patients

The levels of both IGF-1 and IGFBP-3 were not significantly altered after 10 or 20 weeks of physical training in SZ male patients. Curiously, IGFBP-3 increased after 10 weeks of exercise or control conditions, whereas IGF-1 followed the opposite tendency. Nonetheless, these patients were also following pharmacological treatment [93].

#### 4.4.2. IGFBP-1 in Cognition and Treatment-Resistant SZ

Serum IGFBP-1 levels were found to be significantly increased in SZ patients when compared to controls. Curiously, treatment resistant patients exhibited significantly higher IGFBP-1 than chronic medicated patients [94]. This was in contradiction with previous reports showing significantly lower IGFBP-1 [36,39]. What is more, IGFBP-1 was found to be positively correlated with attention in treatment resistant patients and negatively correlated with immediate memory in the chronical medicated group, which points to the fact that IGFBP-1 may also impact cognition in different ways in SZ patients [94].

#### 4.4.3. Still Unexplored IGFBPs in SZ Patients

To the best of our knowledge, peripheral IGFBP-4, IGFBP-5, and IGFBP-6 have not been explored in the context of SZ. In 2011, a study that was searching for a biological blood signature in SZ patients found that IGFBP-2 levels were significantly increased compared to controls. However, after using cortisol levels as a covariate for ANCOVA analysis, IGFBP-2 was found to be non-significant [95]. More recently, *IGFBP-2* expression, but not *IGFBP-4* or *IGFBP-5*, was found to be significantly reduced in the subventricular zone of post-mortem brain samples in SZ patients compared to controls [96]. Nonetheless, *IGFBP-4* gene expression was significantly downregulated in the hippocampus of SZ patients [97]. What is more, linkage disequilibrium score regression analyses observed a negative genetic association between plasma IGFBP-6 protein and SZ [98]. In this regard, we believe there is evidence supporting the study of these IGFBPs in SZ patients.

**Table 1 ijms-26-02561-t001:** Data from 25 articles that have measured peripheral levels of the IGF family in schizophrenia (SZ) throughout years (1999–2024).

Ref.	DSM	Group	Sample Size	Age (Years)	Sex (F/M)	Treatment		Sample Source	Techn	Country	IGF Ligands		Statistics		IGFBPs		Statistics	
[35]	III-R	SZ	28	44 ± 12	12/16	Class Aps	>6 m	Fasting Serum	RIA	Sweden	IGF-1 (ng/mL)	211 ± 75 *	ns (↓)	*p* = 0.246	IGFBP-1 (ng/mL)	20 ± 15	ns	*p* = 0.157
		SZ	13	36 ± 6	6/7	Cloz						181 ± 79				16 ± 8		
[36]	III-R	SZ	14	43 ± 10	7/7	Olanz	>2 m	Fasting Serum	RIA	Sweden	IGF-1 (ng/mL)	0.18 ± 1.10	N/A		IGFBP-1 (ng/mL)	14.7 ± 7.8	↓	*p* < 0.05
[37]	III-R	SZ	47	42 (27–80)	26/21	Classical APs	>6 m	Fasting Serum	RIA	Sweden	IGF-1 (ng/mL)	207.5 ± 66.9 *	“Normal Range”					
[38]	IV	SZ	18	46.9 ± 14.8	5/13	Cloz	>6 m	Fasting Serum	RIA	Sweden	IGF-1 (ng/mL)	0.1 (−2.0–2.4)	“Normal Range”		IGFBP-1 (ng/mL)	12 (5–40)	↓	*p* < 0.05
		SZ	16	39 (23–52)	6/10	Olanz						0.2 (−1.3–1.3)				14 (6–27)		
[39]	IV	SZ0	19	31 ± 5.8	10/9	Baseline	2.5 m	Blood	CLIA, ELISA	UK + Afrikaan	IGF-1 (nmol/L)	25.5 ± 8.7	ns	*p* = 0.57	IGFBP-1 (µg/L)	3.1 ± 3.7	ns	*p*= 0.24
		SZ1				Cloz						24.7 ± 9.1				2.4 ± 2.8		
[44]	IV	SZ	44	33 ± 7.7	21/23	DN		Fasting Plasma	AU400	India	IGF-1 (ng/mL)	123.7 ± 50.0	↓	*p* < 0.006				
		HC	44	32.5 ± 7.6	21/23							159.1 ± 67.9						
[87]	IV	SZ	53	40.5 (22.0–69.0)	All M	Halo	“Stable”	Serum	ELISA	Kuwait	IGF-1 (ng/mL)	178.46	ns	*p* = 0.15	IGFBP-3 (ng/mL)	5300	ns	*p*= 0.21
		HC	52	41.0 (27.0–60.0)								152.31				5157.14		
											IGF-2 (ng/mL)	1445.38	↑	*p* = 0.02				
												1228.46						
[47]	IV	SZ OB	71	40.4 ± 7.5	All M	Cloz		Fasting Plasma	ELISA	China	IGF-1 (ng/mL)	134.7 ± 10.4	↓	*p* < 0.01	IGFBP-3 (ng/mL)	4496 ± 124	ns	*p* > 0.05
		HC OB	50	39.7 ± 1.17								201.0 ± 14.4				4874 ± 166		
		HC	51	40.9 ± 1.14								232.8 ± 12.4				4696 ± 134		
[55]	IV	SZ0	33	33.8 ± 8.2	13/20	DN		Fasting Serum	EAIC	India	IGF-1 (ng/mL)	113.9 ± 44.7	↓ ^HC^	*p* < 0.0001				
		SZ1	33	33.8 ± 8.2	13/20	A-Aps	3 m					141.5 ± 58.8	↑ ^sz0^	*p* < 0.0001				
		HC	33	32.2 ± 8.0	13/20							175.2 ± 63.0						
[95]	IV	SZ	71	31 ± 10	29/42	DF		Serum	HM MAIA	USA and Europe					IGFBP-2 (ng/mL)	52.9 ± 27	↑	*p* = 0.045
		HC	59	30 ± 8	28/31											43.29 ± 19.51		
[49]	IV	Offs SZ	32	27.6 ± 6.4	15/17			Fasting Plasma	RIA	China	IGF-1 (ng/mL)	168.6 ± 53.5	↓	*p* = 0.028				
		HC	37	26.6 ± 3.4	17/20							195.1 ± 44.8						
[73]	IV	SZ_FE	27	24.7	7/20	DN		Fasting Plasma	ELISA	Spain	IGF-1 (ng/mL)	182.42 ± 96.13	ns	*p* = 0.659				
		SZ1m	21		N/A	A-Aps + MS	1 m					237.60 ± 122.39	↑	*p* = 0.039				
		SZ6m	19				6 m					193.92 ± 74.41	ns	*p* = 0.216				
		SZ12m	16				12 m					178.02 ± 97.63	ns	*p* = 0.88				
		HC	27	25.7	13/30							171.60 ± 96.13						
[50]	IV	SZ	50	36.5 ± 11.2	13/37	A-Aps		Fasting Plasma	RIA	Turkey	IGF-1 (ng/mL)	176.06 ± 81.65	ns	*p* = 1.0				
		Sibl SZ	50	35.7 ± 11.1	19/31							175.04 ± 72.14						
		HC	50	35.5 ± 9.2	18/32							175.04 ± 64.01						
[93]	IV	SZ Re0	12	32.9 ± 2.3	All M	Treated	base	Fasting Serum	ELISA, CLIA	Brazil	IGF-1 (ng/mL)	173.70 ± 34.53	ns		IGFBP-3 (ng/mL)	3290.97 ± 440.38	ns	
		SZ Re1					10 w					143.65 ± 24.23				3388.79 ± 434.08		
		SZ Re2					20 w					151.34 ± 26.69				3666.98 ± 500.35		
		SZ Co0	9	33.5 ± 2.6			base					220.33 ± 33.74				3939.81 ± 404.66		
		SZ Co1					10 w					174.96 ± 23.67				4424.55 ± 398.88		
		SZ Co2					20 w					195.06 ± 26.08				4211.39 ± 459.77		
		SZ CT0	13	33.4 ± 12.2			base					197.22 ± 35.39				4278.81 ± 447.21		
		SZ CT1					10 w					181.11 ± 24.83				4322.47 ± 440.81		
		SZ CT2					20 w					170.56 ± 27.35				4153.81 ± 508.11		
[67]	IV-TR	FE	25	25.48 ± 5.4	All M	A-Aps (8)	1.75 ± 0.8	Serum	ELISA	Greece	IGF-1 (ng/mL)	5.57 (5.29, 5.84)	ns	*p* = 0.82				
		UHR	12	24.5 ± 3.1		DN						5.68 (5.4, 6.02)		*p* = 0.259				
		HC	23	27.04 ± 2.9								5.56 (5.37, 5.72)						
[62]	IV	PE	40	32.4 ± 9.8	13/27	DN		Fasting Serum	ELISA	Greece	IGF-1 (ng/mL)	109.66 (15.22–313.48)	↑	*p* = 0.039				
		HC	40	31.9 ± 8.3	15/25							86.96 (18.76–160.36)						
[89]	IV	SZ	30	30.5 ± 8.5	14/16	WO	3 m	Fasting Serum	ELISA	China	IGF-1 (ng/mL)	114.96 ± 65.85	↓	*p* = 0.001				
		HC	26	34.4 ± 9.9	14/12							183.43 ± 86.42						
[88]	IV	SZ	32	30.0 ± 8.5	16/17	DN + WO	3 m	Fasting Serum	ELISA	China	IGF-2 (ng/mL)	199.2 ± 67.2	↓	*p* = 0.04	IGFBP-3 (ng/mL)	849.7 ± 227.1	↓	*p* = 0.049
		HC	30	33.5 ± 9.1	16/14							355.7 ± 70.4				1034.5 ± 390.7		
															IGFBP-7 (ng/mL)	26.8 ± 9.8	↓	*p* = 0.041
																33.0 ± 12.9		
[77]	IV	SZ0	30	30.3 ± 8.5	17/13			Fasting Serum	ELISA	China	IGF-2 (ng/mL)	203.13 ± 64.62	↓ ^HC^	*p* = 0.002				
		SZ1	30			A-Aps	8 w					426.99 ± 124.26	↑ ^SZ0^	*p* < 0.001				
		HC	31	34.2 ± 9.3	15/16							442.34 ± 105.33						
[69]	IV	SZ0	113	29.2 ± 9.3	68/45	DF	2 w	Fasting Plasma	ELISA	China	IGF-1 (ng/mL)	214.45 ± 33.42	↑ ^HC^	*p* = 0.017				
		SZ1	89		N/A	Risper	10 w					202.29 ± 32.40	↓ ^SZ0^	*p* < 0.01				
		HC	58	30.2 ± 6.4	N/A							N/A						
[83]	V	SZ	65	49 ± 10	32/33	Aps	“C”	Fasting Serum	IRMA	Japan	IGF-1 (ng/mL)	109 ± 38	ns	*p* = 0.27				
		HC	20	46 ± 7.4	12/8							120 ± 39						
[82]	V	SZ R	55	36.6 ± 8.5	18/37	Aps		Fasting Plasma	RIA	Turkey	IGF-1 (ng/mL)	137.54 ± 40.28	↓ ^HC, TR^	*p* < 0.001				
		SZ TR	62	33.9 ± 8.6	16/46							165.11 ± 40.95						
		HC	60	33.9 ± 7.9	20/40							173.37 ± 38.85						
[92]	V	SZ_FE	15	31.7 ± 15.5	4/11	DN		Fasting Plasma	ELISA	Spain	IGF-2 (ng/mL)	66.93 ± 39.99	↓ ^HC, ME^	^HC^*p* = *0*.0017	IGFBP-7 (ng/mL)	54.04 ± 23.49	↓ ^ME^	^ME^*p* = 0.0017
		SZ_FE0	11	32.4 ± 16.2	3/8	DN								^ME^*p* < *0*.0001				
		SZ_FE1				A-Aps						153.45 ± 30.98	↑ ^FE0^	*p* = 0.0078		82.47 ± 27.04	↑ ^FE0^	*p* = 0.0137
		SZ_ME	40	39.9 ± 11.3	13/27	Treated						144.06 ± 30.84	↑ ^FE^	^FE^*p* = 0.0065		78.28 ± 18.55	↑ ^HC^	^HC^*p* = 0.0214
		SZ_MER	19	40.0 ± 12.1	5/14	Treated						148.64 ± 33.41	↑ ^HC^	*p* = 0.0192		81.98 ± 15.61	↑ ^HC^	*p* = 0.0185
		SZ_MENR	21	39.7 ± 10.7	8/13	Treated						140.95 ± 28.64	ns^HC, MER^	*p* > 0.05		74.93 ± 20.67	ns^HC, MER^	*p* > 0.05
		HC	45	41.2 ± 10.6	20/25							114.24 ± 55.00				64.9174 ± 25.36		
[58]	V	SZ	71	38.2 ± 9.9	33/38	Treated		Fasting Serum	ELISA	Japan	IGF-1 (ng/mL)	159.7 ± 49.6	↑	*p* = 0.01				
		HC	71	41.4 ± 9.3	33/38							137.9 ± 41.3						
[94]	IV	SZ TRS	31	40.6 ± 9.2	All M	Treated	>6 m	Fasting Serum	LSCD	China					Log IGFBP-1 (pg/mL)	4.49 ± 0.27		
		SZ CM	49	40.6 ± 10.3			6 m									4.27 ± 0.38	↓ ^TRS^	*p* = 0.048
		HC	53	40.0 ± 9.3												3.78 ± 0.46	↓ ^TRS, CM^	*p* < 0.001

**Explanation by column.** w: weeks. *DSM*: version of The Diagnostic and Statistical Manual of Mental Disorders (DSM) used to diagnose SZ or SZ axis. *Group*: SZ: Schizophrenia; 0: measure at baseline; 1: measure after a given time; HC: healthy control; OB: obese; Offs: offspring; FE: first episode; 1 m: 1 month; 6 m: 6 months; 12 m: 12 months; Sibl: siblings; C: chronical; Re: SZ patients following resistance training; Co: SZ patients following concurrent training; Ct: SZ patients not following physical training; PE: psychosis episode; TR: treatment resistant; ME: multiple episode; R: responder; NR: non-responder. UHR: Ultra High Risk of suffering a psychotic episode. *Treatment*. Class: classical antipsychotics; Cloz: clozapine; Olanz: olanzapine; Aps: antipsychotics; DN: drug-naïve; halo: haloperidol; A-Aps: atypical antipsychotics; DF: drug-free; MS: mood stabilizers; WO: washed-out from treatment; Risper: risperidone; y: years; m: months; d: days. Techn: technique used to measure peripheral levels of IGF. RIA: radioimmunoassay; ELISA: enzyme-linked immunoassay; CLIA: chemiluminescence immunoassay; AU400: Olympus AU400 analyzer; ECLIA: electro-chemiluminescence immunoassay; HM MAIA: HumanMAP multiplexed antigen IA; EAIC: enzyme-amplified immuno-chemiluminescence; LSCD: luminex liquid suspension chip detection method; IRMA: immunoradiometric assay. *Statistics*. ns: non-significant; ↑: significantly elevated; ↓: significantly reduced. *Superscripts* indicate the reference statistical comparison. * Mean ± SD and statistical tests were calculated according to data from the article, in brackets it is indicated what was found in the article for median values.

## 5. IGF Peripheral Levels in Depression

### 5.1. Major Depressive Disorder (MDD)

Major depressive disorder (MDD) represents the classic condition among depressive disorders. The etiology of MDD is the result of a complex relation between genetic, biological, environmental, and social factors [99]; it is one of the most prevalent PDs with an estimated 12%, being almost double that in women [100]. MDD patients suffer from low mood, anhedonia, decreased interest in activities, feelings of guilt or worthlessness, generally accompanied by lack of energy, poor concentration, changes in the pattern of sleep or appetite, and suicidal thoughts in their worst [101]. Treatment options include antidepressants (Ads), psychotherapy, or electroconvulsive therapy (ECT). There are different classes of Ads, depending on their mechanism of action, such as selective serotonin reuptake inhibitors (SSRIs), like fluoxetine or escitalopram, serotonin-norepinephrine reuptake inhibitors (SNRIs), like venlafaxine or duloxetine, serotonin modulators, like trazodone or vortioxetine, tricyclic (TCAs), including imipramine or amitriptyline, and monoamine oxidase inhibitors (MAOIs), like phenelzine [102]. The Hamilton Rating Scale for Depression (HAMD or HDRS) is the gold standard to evaluate symptom severity in MDD patients [103]. The traditional neurobiological hypothesis of depression used to be explained in accordance to what was known from the mechanism of action of Ads, which implied that depression resulted from a deficit in one or more neurotransmitters, such as serotonin or norepinephrine. However, the etiology of depression is still complex and uncertain, Duman proposed a neurotrophic hypothesis from which depression could also be a consequence of reductions in the brain expression of certain entities, called neurotrophins or neurotrophic factors, with a brain-derived neurotrophic factor (BDNF) having a central role [104,105]. From this, the IGF family has also be found to be possessing neurotrophic activity in certain brain areas which are implicated in the development of depression [106].

### 5.2. IGF-1 in Depression

#### 5.2.1. IGF-1 First Studies in MDD Patients: GH Challenges and Dexamethasone Bias

During the late 1980s, a considerable number of studies made by the *Psychiatric and Endocrinology Departments of Würzburg University in Germany* investigated potential neuroendocrinology anomalies in MDD patients. Diurnal GH hypersecretion and attenuated GH secretion in response to different stimulus were considered two lines of evidence that suggested an altered HPS axis in MDD patients [107]. All of these studies found significant higher IGF-1 plasma levels in MDD patients [108,109,110,111,112,113,114,115]. However, there were some limitations associated with these studies. First, a small fraction had been diagnosed with bipolar disorder. Second, GH secretion in response to GHRH [108,110,111,113], clonidine [111,115], and TRH [113] were tested. Third, the most cited articles in these set over the years performed 0.5–1 mg dexamethasone (DEX) test [109,114] to check if there were anomalies in the HPA axis in MDD patients. Some strengths were the inclusion of a control group and variable drug-free periods before sample extraction.

Importantly, Lesch et al. found a significant correlation between cortisol and IGF-1, which led them to establish the hypothesis of cortisol upregulating IGF-1 peripherally, either directly, by raising GH release by increasing GHRH, or indirectly, due to SST inhibition [109]. Rupprecht et al. found that there was no significant difference in IGF-1 among MDD patients during severe depression and after recovery with Ads. DEX challenge did not alter IGF-1 in controls, whereas MDD patients showed significantly higher IGF-1 levels after post-DEX, both during acute depressive episode and after recovery.

Interestingly, cortisol non-suppressors after DEX showed significantly higher IGF-1, whereas it was not the case of cortisol suppressors [114]. In 1990, IGF-1 was found to be normalized after 4 weeks of Ad treatment with moclobemide or maprotiline [115].

In MDD teenagers, no significant difference was found for IGF-1 compared to controls when realizing a GHRH response test [116]. Another study found no significant difference for IGF-1 in MDD patients. This time, the IGF-1 comparison was established between GHRH and clonidine stimulated MDD patients and controls [117]. Most of these studies cannot be found in a regular systematic search, since IGF-1 was previously known as Somatomedin-C.

In 1997, Deuschle et al. found that plasma IGF-1 was significantly increased in MDD patients, but not IGFBP-2, IGFBP-3, GH, and GH-BP [118]. In responders, IGF-1 was significantly reduced in contrast to non-responders after 22–55 days of Ad treatment. Moreover, IGF-1 levels were significantly and positively correlated with 24 h average cortisol levels. Similarly, it was suggested that cortisol regulation over IGF-1 could be directed by the genomic regulation of the liver, modulating IGF-1 synthesis, or by the inhibition of IGF-1R, which would indirectly increase IGF-1 circulating levels. Nonetheless, these patients were also subjected to a DEX test. Age was negatively and significantly correlated with IGF-1, which lead to older MDD patients having significantly reduced IGF-1. Reduced GH in the elders could be explaining this result. This was not the case for IGFBP-2, IGFBP-3, or GH-BP. Curiously, only IGFBP-2 was significantly and negatively correlated with BMI, but only in the MDD group. MDD patients were washed-out from Ads for at least 6 days [118].

#### 5.2.2. Saliva Cortisol and Serum IGF-1/IGFBP-3 in MDD Patients

IGF-1 and IGFBP-3 serum levels were significantly decreased in MDD responders after 35 days, but not 14 days, of amitriptyline monotherapy, whereas only IGF-1 was significantly reduced in a paroxetine-treated group. MDD patients were subjected to a wash-out period of 6 days. IGF-1 levels were significantly and positively correlated with saliva cortisol levels in MDD responders, whereas both IGF-1 and IGFBP-3 were negatively correlated with age. It was then hypothesized that the lowering of IGFBP-3 could be a mechanism to increase the bioavailability of IGF-1, through the reduction of IGF-1 half-life in circulation that would promote IGF-1 inclusion into target cells. Interestingly, since free IGF-1 levels did not change before or after treatment, despite changes in total IGF-1 and IGFBP-3, it was suggested there was a counter-regulatory action from other IGFBPs, indirectly raising the question of whether it could be of interest to measure other IGFBPs. Based on the discrepancy between free IGF-1 and total IGF-1, Weber-Hamann et al. suggested that IGFBP-proteases might have a role in reducing IGFBP affinity for IGF-1, therefore increasing free IGF-1 in circulation. Nonetheless, no comparison with a control group was made [119].

#### 5.2.3. IGF-1 and Cortisol in MDD Patients: A Counter-Regulatory Neuroprotective Mechanism

IGF-1 was found to be significantly elevated in MDD patients both at baseline and after 6 weeks of Ad treatment when compared to controls, even after correcting by age, sex, BMI, and HAMD. Curiously, after dividing by sex distribution, the statistical meaning was only preserved in men. Nonetheless, there was no significant difference in IGF-1 between men and women. At baseline, non-responders had significantly higher IGF-1 than responders (HAMD-21 < 10), but not after treatment. However, there was no significant change after treatment with Ads for each group. After changing the responding criteria (HAMD-17 ≤ 7), IGF-1 levels remained significantly higher in non-remitters after treatment, but no significant correlation was found between IGF-1 and depressive symptoms (HAMD). Kopczak et al. proposed that the peripheral elevation of IGF-1 could be a counter-regulatory mechanism to stimulate central neuroprotective roles in depressed patients (Figure 4A). Moreover, they simultaneously measured serum cortisol levels without a DEX test for the first time. Nonetheless, only in the responder group, IGF-1 and cortisol were significantly and positively correlated at baseline [120]. Cortisol was thought to reduce IGF-1 levels in SZ patients [55]. Hypercortisolemia would reduce IGF-1 levels either through the reduction of IGF-1 transcription or the expression of IGF-1R and GH-R. In this case cortisol was not altered in MDD patients. The reasons behind the abnormal relation between the HPA and HPS axes could be resistance to glucocorticoid receptors and impaired pituitary feedback to cortisol, leading cortisol not to reduce IGF-1 levels. Moreover, IGF-1 could mediate cortisol activation, since GH was found to be capable to inhibit 11ß-hydroxysteroid dehydrogenase type 1, an enzyme that actives cortisone into cortisol [120].

Tajiri et al. found higher IGF-1 and cortisol serum levels in MDD patients with increased symptom severity (HAMD), but no significant correlation was found between IGF-1 and cortisol. Curiously, higher IGF-1 was significantly correlated with melancholic and suicide scores in the HAMD scale. Also curious was that IGF-1 was inversely and significantly correlated with the Global Assessment of Functioning (GAF) scale. All patients were taking Ads, mostly SSRIs [121].

#### 5.2.4. IGF-1 and Duman’s Neurotrophic Hypothesis in MDD Patients

In 2016, Bot et al. found that plasma IGF-1 was significantly elevated in MDD patients without Ads, whereas patients who were under Ad treatment had significantly lower IGF-1 than controls, independent of drug type or response to treatment [122]. Therefore, they commented that finding higher peripheral IGF-1 levels in MDD patients was unexpected in regard to the Duman’s neurotrophic hypothesis (Figure 4B) [104]. IGF-1 should have been found significantly reduced in MDD patients, as it is for BDNF [123]. Bot et al. first proposed that the peripheral increase of IGF-1 could be a countermeasure to compensate for the loss of neurogenesis at CNS in MDD Patients. Supporting this difference, IGF-1 can cross the BBB, whereas BDNF cannot. Second, the alteration of GH levels could be behind the IGF-1 increase in MDD patients, either directly or indirectly. Third, as is the case with Ad treatment [75], IGF-1 may increase in early stages of depression and then decrease in later stages, since these patients would have been exposed to Ads more frequently [122]. Another study found no significant alteration in IGF-1 serum levels between controls and MDD patients who were not taking Ads for at least 3 months before sampling. Nonetheless, it has to be said that symptom severity was considerably lower (19.7 ± 2.6 HDRS) in these MDD patients [124] when compared to other studies [118,119,120]. In this publication, Rosso et al. again commented on the apparent contradiction between the neurotrophic hypothesis of Duman and their results [124].

#### 5.2.5. IGF-1 Potential Diagnostic Value and Cognitive Assessment in MDD Patients

Posteriorly, the studies of Levada and Troyan explored IGF-1 as a neurotrophic factor in which reduced expression could impair neuroplasticity in depression [125]. First, they found that serum IGF-1 was significantly higher in MDD patients compared to controls. Then, they established a mean diagnostic cut-off value of 178 ng/mL for MDD (AUC = 0.820, 83% sensitivity, and 71% specificity). However, IGF-1 was significantly and negatively correlated with age and positively with the number, duration, and severity of depressive episodes, according to the Clinical Global Impression Severity (CGIS) and the Montgomery–Asberg Depression Rating Scale (MADRS). Therefore, age-dependent cut-off values were calculated according to their findings. After 2 months of vortioxetine treatment (10–20 mg/day), IGF-1 levels significantly decreased in MDD patients. IGF-1 was inversely correlated with the Digit Symbol Substitution Test (DSST), which in cognitive terms implied that higher IGF-1 levels were associated with less processing speed, executive function, learning, memory, and concentration [126].

In their next study, Levada and Troyan again found significantly higher IGF-1 in MDD patients, whereas BDNF was significantly lower. After 8 weeks with vortioxetine monotherapy, both neurotrophins significantly reverted their concentrations. IGF-1 was again significantly and negatively correlated with the number, duration, and severity of depressive episodes. This time, with a negative association with the DSST and the Rey Auditory Verbal Learning Test (RAVLT). Interestingly, an ROC curve analysis including BDNF and IGF-1 offered an excellent diagnostic value for depression (AUC = 0.916, *p* < 0.0001), which was higher than BDNF or IGF-1 separately [127].

#### 5.2.6. IGF-1 and Relaxin-3 in MDD Patients

New evidence has found significantly elevated IGF-1 and reduced relaxin-3 in MDD patients compared to controls. Despite having low symptom severity (HDRS 19.02 ± 2.41), a significant and positive association was found between HDRS and IGF-1. Interestingly, IGF-1 and relaxin-3, which is a neuropeptide that is believed to modulate depressive-like behaviors, were found to be significantly and negatively correlated in MDD patients [128]. To the best of our knowledge, no study has further explored the relation between IGF-1 and relaxin-3 in the context of MDD patients.

#### 5.2.7. IGF-1 in Relation to Hormones, HPA Axis and Insulin Resistance in MDD Patients

In their first study, Arinami et al. found a significant increase in serum IGF-1 in treated-MDD patients. The same cohort exhibited significantly lower estradiol, but not testosterone or cortisol. This time, IGF-1 and symptom severity (HDRS) were not significantly related. However, IGF-1 was significantly associated with Dehydroepiandrosterone sulfate (DHEAS) levels [129]. In their next study, MDD patients maintained significantly higher IGF-1 levels, but cortisol levels were also higher. Despite age being significantly and positively correlated with cortisol and negatively correlated with IGF-1, no significant correlation was found between each other. However, both IGF-1 and cortisol were statistically and positively associated to symptom severity (HAMD), whereas total imipramine equivalent dose was not significantly associated to cortisol or IGF-1. Interestingly, no changes were found between men and women for IGF-1 [58]. Finally, IGF-1 levels were significantly elevated in non-remitted MDD patients in comparison to remitters (HAMD ≤ 7) and controls. Moreover, no significant difference was found for the HOMA-IR index among the three groups, but it was significantly and positively correlated with IGF-1 in non-remitted MDD patients. To some extent, this evidence may suggest that IGF-1 could be behind insulin resistance development in MDD patients who do not fulfill remission after Ads [130].

#### 5.2.8. IGF-1 in Drug-Naïve First Episode (FE) MDD Patients

Noteworthily, a recent study found that FE drug-naïve MDD patients showed significantly higher IGF-1 serum levels than controls. Curiously, no significant correlation was found between IGF-1 and age, sex, or symptom severity. This study offers a very special and rare case, evaluating IGF-1 in FE MDD. Previous studies have established wash-out periods, but that might not entirely represent FE and/or drug-naïve conditions, since MDD washed-out patients may have been exposed to multiple episodes (ME) and/or Ads for long periods [131].

#### 5.2.9. IGF-1 in Women Under Different Conditions in MDD Patients

Some studies have evaluated IGF-1 in women under different premises. First, serum IGF-1 was explored as a peripheral indicator of GH status, since its deficiency may lead to lower mineral bone density. However, IGF-1 was not significantly altered in MDD patients compared to controls. These patients were washed-out for Ads for at least 2 weeks. Curiously, MDD women had significantly higher levels of urinary cortisol [132].

In 1999, Franz et al. found a trend for elevated serum IGF-1 in non-treated MDD patients (*p* = 0.07). IGF-1 was significantly and negatively correlated with age and menstrual phase. As covariates, age, group (MDD and controls), and menstrual phase explained 64% of the variance observed for IGF-1. Two alternatives could explain IGF-1 elevation in MDD women. First, during a 24-h lapse, there could be a peak in GH release that stimulates a drastic elevation in IGF-1. Second, IGF-1 elevation could come from a non-hepatic source, a process that would be independent of GH, and that may be influenced by other hormones, such as ACTH (adrenocorticotropic hormone), TSH (thyroid-stimulating hormone), estradiol, or PTH (parathyroid hormone). However, whereas the first option was difficult to prove empirically, the second lacked a theoretically strong basis, since non-hepatic IGF-1 fundamentally exerts autocrine and paracrine functions, and is apparently not released into circulation. One substantial suggestion was that IGF-1 peripheral elevation in MDD patients could not be simply reflecting GH alterations [133].

Conversely, a study made in Japan involving 8042 women, found that pregnant women who had higher IGF-1 peripheral levels during the first trimester were less propense to develop postpartum depression during the first month after birth. In this sense, higher IGF-1 levels during specific periods of pregnancy could protect against developing postpartum depression [134].

#### 5.2.10. IGF-1 Rhythm in FE MDD

Li et al. found that both IGF-1 and GH had significant diurnal rhythms in healthy subjects, but it was disrupted in FE MDD patients. However, after 8 weeks of escitalopram treatment, the rhythm of IGF-1 and GH were both restored to what was found in controls. Nonetheless, IGF-1 peak phase was found 4 h later. Despite rhythm dysregulation, no significant difference was found for both IGF-1 and GH between MDD patients and controls, neither at baseline nor after AD treatment [135].

#### 5.2.11. IGF-1 Meta-Analysis in MDD Patients

In 2016, the first meta-analysis for both BD and in MDD patients was based on six of the above referenced case-control studies [109,118,120,132,133,135]. In summary, it was concluded that IGF-1 peripheral levels were significantly elevated in MDD patients when compared to controls, and that there was no alteration before and after treatment with Ads [136]. Illness duration was the only parameter that was significantly and negatively correlated with IGF-1. Nonetheless, it is not clear whether this association was found only for MDD or both BD plus MDD patients. Conversely, there was no significant association with age, sex distribution, BMI, age at onset, or symptom severity (HDRS). Therefore, Tu et al. concluded that altered peripheral IGF-1 levels could be a trait marker in mood disorders as a whole, considering both MDD and BD patients, but that would not accurately represent symptom severity [136].

Shortly thereafter, a new meta-analysis made in nine studies [109,118,120,124,132,133,135,137,138] reached the same conclusions [139]. However, it included two studies that we could not find in PubMed [137,138]. Based on reports from the meta-analysis of Chen et al. [139], these studies found significantly higher IGF-1 in drug-naïve [137] and treated MDD patients [138]. From our point of view, it is not clear whether drug-naïve and washed-out MDD patients should be included in the same statistical group, since patients who have been exposed to treatment over time might not entirely reflect drug-naïve conditions despite wash-out periods [139]. Moreover, wash-out periods widely vary among studies: 6 days [118], 14 days [109], or even 3 months [124].

The meta-analysis by Shi et al. included IGF-1 and other neurotrophins in the context of Duman’s neurotrophic hypothesis. Curiously, IGF-1, along with fibroblast growth factor 2 (FGF-2) (5 studies), was among the least studied factors, with only 7 studies, in contrast to BDNF (97), glial cell-line derived neurotrophic factor (GDNF) (10), vascular endothelial growth factor (VEGF) (23), nerve growth factor (NGF) (11) and S100 calcium-binding protein B (S100B) (11). In contrast to BDNF and NGF, IGF-1, VEGF, and S100B were found to be significantly higher in MDD patients when compared to controls, whereas GDNF and FGF-2 were found to be unaltered [140]. Mosiołek et al. published a review exploring the effects of Ads in both IGF-1 and BDNF serum levels [141]. They concluded that IGF-1 response to Ads needs further research, although four studies [120,122,126,127] indicate a decrease after Ads [141]. Qiao et al. also included a large meta-analysis of 13 studies [58,109,118,120,122,124,126,127,129,130,132,133,135] that involved 2792 MDD patients and 1146 controls, concluding that IGF-1 levels are significantly increased in MDD patients [131].

#### 5.2.12. IGF-1 in the Cerebrospinal Fluid (CSF) in MDD Patients

IGF-1 CSF is about 1% of peripheral concentrations. MDD patients showed significantly reduced IGF-1 CSF compared to controls. After Ad treatment, MDD patients experienced a significant increase in IGF-1 CSF levels, regardless of type of drug, time of exposure, or depressive symptomatology (HDRS). MDD patients were washed-out for at least 6 days [24]. Curiously, IGF-1 CSF followed an opposite trend compared to peripheral blood. However, the relation between CSF and peripheral IGF-1 in MDD patients still remains unclear since this was the only study available.

#### 5.2.13. IGF-1 in Alternative Therapies and Exercise in MDD Patients

MDD patients who were interrupted with a placebo while being treated with paroxetine, but not fluoxetine or sertraline, experienced a significant increase in IGF-1 levels. Nonetheless, this was not accompanied by cortisol alterations [142]. Alternatively, a significant reduction in IGF-1 serum levels was found after 6 h of ECT in Ad-free MDD patients [143].

Physical training has been proposed as a potential therapy for MDD patients, since it shows Ad-like effects, but the underlying mechanisms are still unclear. IGF-1 seems to be a potential mediator in this process [144]. De Sousa et al. postulated the hypothesis that exercise stimulates peripheral IGF-1 production. Then, IGF-1 crosses the BBB and acts to enhance the expression of BDNF, which in turn activates its receptor (tropomyosin receptor kinase B) at the hippocampus and prefrontal cortex. Finally, ERK translocates into the nucleus inhibiting depressive-like behaviors [145]. Conversely, regular aerobic exercise for 3 months did not induce a significant change in serum levels of IGF-1, BDNF, or VEGF in MDD patients [146]. Contradictorily, a recent meta-analysis concluded that chronic exercise may reduce IGF-1 peripheral levels in MDD patients [147]. In teenagers with MDD, 6 weeks of endurance ergometer cycling training, but not muscle strengthening whole body vibration, showed a significant increase in serum levels of IGF-1, suggesting that the type of training may be important [148].

However, no significant correlation was found between IGF-1 and depressive scores and, unfortunately, these studies did not include a control group [146,148]. More recently, Kelly et al. showed that, even when controlling for Ad use, depression severity (HAMD), or physical-exercise habits, patients with mild to moderate MDD showed a significant increase in IGF-1 after acute maximal exercise in comparison to controls. However, this study has a reduced sample size in the control group, in contrast to the MDD group (*n* = 34 vs. 113, respectively) [149].

#### 5.2.14. IGF-1 Population-Based Studies and Depressive Symptoms

Population-based studies offer huge sample sizes, but lack a proper diagnosis of MDD, and used self-applied questionnaires, which are a very useful tool in the evaluation of depressive symptoms in general population, but do not imply a proper diagnosis.

It was not until 2014 when the first population-based study was made relating IGF-1 and depression. The study was made in the region of Pomerania, and found that IGF-1 baseline levels below 10% increased the 5-year risk of depression in women. For men, the increased risk of depression was found to be above 90%. Sievers et al. suggested that the differences between men and women could be explained by differences in sexual hormones, such as estradiol [150]. Another study made in the KORA-age cohort, determined that IGF-1 serum levels were positively correlated with increased depressive symptoms according to the Geriatric Depression Scale (GDS), but just in women. Higher IGFBP-3 and lower IGF-1 were significantly associated with well-being, but again only in women. The subjects included in this work were between 64 and 93 years of age by the time they were studied. Interestingly, this model included many covariates, such as age, physical activity, sleep, BMI, smoking, and cognition [151].

Serum IGF-1 levels were evaluated in relation to the incidence of depression in dwelling elderly people. For men, mid values of IGF-1 [11.5–15.5 nmoL/L] were less prone to develop minor depression in comparison to men with higher IGF-1 levels [>15.5 nmoL/L]. For women, low IGF-1 levels [≤11.5 nmoL/L] tended to increase the probability of prevalent MDD in comparison to higher levels. Intermediate WIGF-1 concentrations in women decreased the odds of minor depression at a three-year follow-up. Given the fact that some inconsistencies were found between cross-sectional and population-based studies, Van Varsseveld et al. suggested a U-shape relation between IGF-1 and depression, with both high and low IGF-1 levels more related to depression (Figure 4C) [152].

Subsequent results have again suggested a U-shaped relation between IGF-1 and depressive symptoms in the ELSA cohort. The results of Chigogora et al. [153] partially agree with previous population-based studies, in terms of low IGF-1 levels being associated with a higher risk of depression in women [150], but they also found this risk in men. They proposed that the discrepancy between case-control and population-based studies could be due to Ad treatment that would alter IGF-1 levels. Nonetheless, IGF-1 alterations could be a side effect from prior imbalance in depression [153].

#### 5.2.15. IGF-1, IGFBP-3, Depressive Symptoms and Ageing

IGF-1 plasma levels were significantly and negatively correlated with age in the elderly (>65 years). Nonetheless, no significant correlation was found for IGF-1 and depressive symptoms (CES-D) present in 22 of 247 participants [154]. Moreover, in older people (~80 years), significantly lower IGF-1 and IGFBP-3 were associated with alterations in cognitive performance when adjusting for depressive symptoms in 90 of 353 subjects [155]. Conversely, no significant difference was found for IGF-1 and IGFBP-3 between elderly subjects (~76 years) with depressive and non-depressive scores (GDS-5), regardless of sex. In this large cohort, IGF-1 significantly and positively correlated with BMI in men, and with the Mini-Mental State Examination (MMSE) in women [156]. Moreover, the correlation between depressive symptoms and cognition increases in older subjects (~60 years) with lower IGF-1 plasma levels (<75 ng/mL). However, there was no significant correlation between depressive scores (GDS) and IGF-1 in both lower and higher IGF-1 groups [157]. Curiously, in older participants (>50 years), higher IGF-1 levels were found to be associated with lower levels of loneliness, independent of age, medical history, and depression [158].

#### 5.2.16. IGF-1 and Depressive Symptoms in Concomitant Diseases

The prevalence of depressive symptoms is considerably high in many diseases. It is interesting to find that IGF-1 has also been explored in these contexts (Figure 4D), proving the potential link between depression and IGF-1. Most of these studies normally employ questionnaires that evaluate depressive symptoms, such as the Beck Depression Inventory, the epidemiological studies depression scale, the geriatric depressive scale, the symptom checklist-90, the general health questionnaire, and so on. These questionaries do not provide an official diagnosis of depression. We believe that the complete coverage of this section would exceed the scope of our review. Nonetheless, we mention core ideas since these studies were found under the terms “IGF depression” in PubMed.

**Figure 4 ijms-26-02561-f004:**
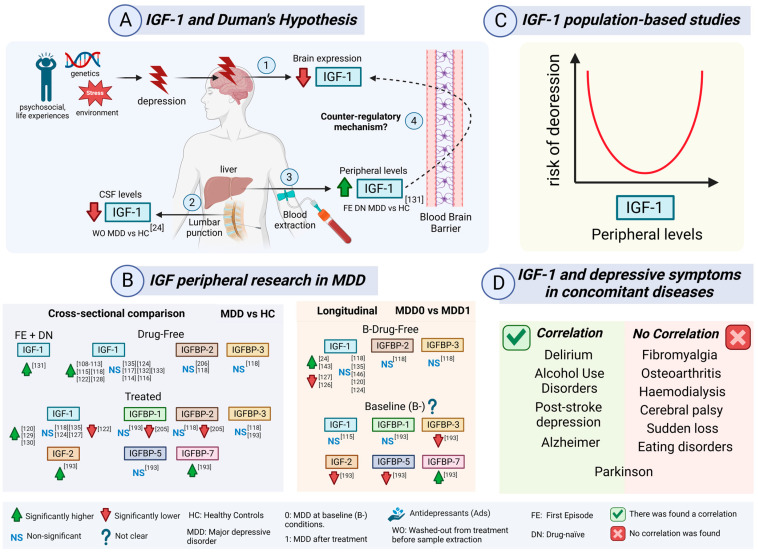
IGF peripheral research in MDD patients. (**A**) Relation to Duman’s hypothesis and IGF-1 counter regulatory mechanism. (1) Depression can result from various factors, including genetics, environmental stressors, psychosocial influences, and life experiences, all of which may contribute to a significant reduction in brain IGF-1 levels. (2) In CSF, washed-out MDD patients exhibited significantly lower IGF-1 levels. (3) Interestingly, the opposite trend was observed in the periphery of drug-naïve and first episode MDD patients. (4) Since IGF-1 can cross the blood–brain barrier, this suggests a compensatory mechanism in MDD patients to counteract brain IGF-1 deficits. (**B**) Research from case-control studies. (**C**) IGF-1 population-based studies show a U-shape between IGF-1 levels and risk of depression. (**D**) Potential correlations between IGF-1 and depressive symptoms in concomitant diseases. CSF: cerebrospinal fluid.

Briefly, a lack of correlation has been found between depressive symptoms and IGF-1 levels in fibromyalgia [159,160,161], osteoarthritis [162], patients in hemodialysis [163], with cerebral palsy [164], in people suffering from unexpected sudden loss [165], and with eating disorders such as anorexia or bulimia [166]. Conversely, an increased incidence of delirium was significantly correlated with the presence of depression and IGF-1 levels in older patients [167].

Higher IGF-1 levels have been found in depressive patients with alcohol use disorder (AUD) [168], whereas others detected no statistical effect of depression comorbidity in the plasma levels of IGF-1 in AUD patients during abstinence [169]. Patients with traumatic brain injury (TBI) and GH deficiency showed greater depressive symptoms than TBI with sufficient GH secretion, while IGF-1 levels were not different [170]. In this sense, IGF-1 levels seem not to be a good predictive marker in GH deficiency in patients with TBI, despite showing increased depressive symptoms [171]. However, when compared against controls, mild TBI patients had both significantly higher IGF-1 levels and a higher prevalence of depressive symptoms [172]. In acromegaly, patients secrete GH in excess, leading to significantly increased IGF-1 levels [173]. Nevertheless, even after treatment and the recovery or normal IGF-1 levels, acromegalic patients do not significantly improve depressive symptoms [174]. The reduction in depressive symptoms seems to be related to GH reduction rather than IGF-1 [175]. Intriguingly, achieving the primary treatment goal for acromegaly—normalization of GH and/or IGF-I levels—seems considered essential for improving depressive symptoms [176].

In neurodegenerative diseases, no significant association was found for IGF-1 and depressive symptoms in Parkinson’s patients [177], neither after deep brain stimulation in the subthalamic nucleus [178] nor after exercise [179]. Nonetheless, Shi et al. found a negative and significant correlation between IGF-1 plasma levels and depressive symptoms in Parkinson’s patients [180,181]. It was also speculated that serum proteins, such as IGF-1, could be mediating depression symptomatology in patients with Alzheimer’s disease (AD) [182]. However, older adults without dementia showed no significant differences in depressive symptoms and IGF-1 plasma levels when compared to patients with AD [183].

#### 5.2.17. IGF-1 in Post-Stroke Depression (PSD) Compared to MDD Patients

Post-stroke depression (PSD) is a common subsequent complication of stroke which worsens the rehabilitation process. IGF-1 serum levels were not significantly different between PSD and MDD patients, nor when compared against non-PSD patients and controls. Non-PSD patients had significantly reduced IGF-1 compared to MDD patients. Depressive symptoms (HDRS) were increased in MDD patients (19.45 ± 4.75) compared to PSD patients (16 ± 5.45), but without reaching statistical meaning [184]. During acute stages of ischemic stroke, PSD patients showed lower IGF-1 serum levels than non-PSD patients. The diagnosis of PSD was made at a 1-year follow up, and patients in the lowest quartile of IGF-1 levels (<92 ng/mL) had about a 62% chance of developing PSD, whereas patients in the second quartile (92–112 ng/mL) had almost a 37% chance [185]. In contrast to other biomarkers, such as VEGF, serum levels of IGF-1 were not significantly correlated with improvements in depressive symptoms either at baseline or after 3-weeks of rehabilitation after ischemic stroke. In this case, there was not an official diagnosis of PSD [186]. More recently, a meta-analysis concluded that IGF-1 levels are not significantly altered in PSD compared to non-PSD patients [187].

### 5.3. IGF-2 as a Promising Candidate in MDD Patients

Studies in rodent models showed that systemic IGF-2 injections improved depressive-like behaviors in both lipopolysaccharide-treated mice [188] and depressive rats [189]. Moreover, stress increases depressive-like behaviors as well as reduces IGF-2 expression in the hippocampus, prefrontal cortex [190], central nucleus, and amygdala [191] in mice. Moreover, *igf-2* expression levels are significantly higher in non-depressed mice [192].

We published the first case-control study on plasma IGF-2 levels in MDD patients. Similarly to IGF-1, IGF-2 was found to be significantly elevated in MDD patients in comparison to controls. After a period of 19 ± 6 days under Ads, MDD patients showed a significant decrease in IGF-2 levels. Finally, there was no significant correlation between IGF-2 levels and depressive symptoms (HDRS) or cognitive state (MMSE) at baseline or after treatment period [193].

Depression has been postulated as a potential risk factor for Alzheimer’s [194,195]. Interestingly, AD patients exhibited a significant reduction in *IGF-2* gene expression in the anterior frontal cortex [196] and IGF-2 protein levels in the hippocampus compared to controls [197]. Patients with both AD + MDD showed significantly less *IGF-2* gene expression in the hippocampus and anterior cingulate cortex in contrast to AD patients [189]. In the same line, most studies found a significant elevation in IGF-2 CSF levels in AD patients [198,199,200,201], which was suggested as a neuroprotective mechanism in response to neuronal injury [202]. In MDD patients, no study has yet to evaluate CSF IGF-2. In peripheral measures, IGF-2 has been found to be significantly increased [199] or significantly decreased compared to controls. Lack of correlation between IGF-2 with MMSE was also found for both serum and CSF IGF-2 levels in AD patients [198].

Therefore, we believe there are several reasons to continue the study of peripheral IGF-2 in MDD patients. From our perspective, it would be interesting to measure IGF-2 in washed-out or drug-naïve MDD patients in a scenario without long-term Ad treatment influence.

### 5.4. IGFBPs in MDD Patients

#### 5.4.1. IGFBP-1, IGFBP-3, IGFBP-5, and IGFBP-7 in MDD Patients

In our study, MDD patients had no significant alteration in plasma IGFBP-1, IGFBP-3, and IGFBP-5 at baseline, whereas IGFBP-7 was significantly increased compared to controls. Conversely, both IGFBP-3 and IGFBP-5 were significantly decreased after Ad therapy, whereas IGFBP-7 was significantly increased and IGFBP-1 was unaltered [193]. IGFBP-3 and IGFBP-5 reductions could be a consequence of IGF-2 decrease after Ad treatment, or it could be a first response to treatment that would lead to IGF-2 inhibition. The difference in IGFBP-1 could be explained by the fact that both IGFBP-3 and IGFBP-5 are major IGF transporters and the only IGFBPs that are bound to ALS, whereas IGFBP-7 binds IGFs with considerably less affinity than IGFBP-3 or IGFBP-5 [23]. Age was found to be positively correlated with IGFBP-1 and IGFBP-7 in the whole sample, whereas only IGFBP-7 and IGFBP-1 remained significant in the MDD group and in the control group, respectively. In fact, IGFBP-7 variance was better explained by both age and sex, rather than MDD diagnosis. Curiously, women had significantly increased levels of IGFBP-3 and IGBFP-5 than men in the whole cohort of our study [193]. In line with our results, IGFBP-7 was the only IGFBP significantly increased in the prefrontal cortex of AD patients [203].

#### 5.4.2. IGFBP-2 in MDD Patients

In 2016, Chan et al. found that IGFBP-2 was significantly and positively associated with response to imipramine monotherapy, negatively to mixed Ads, but with no significance to venlafaxine monotherapy. Time under treatment varied between 4 and 26 weeks. Interestingly, all patients were washed-out from Ads for at least 1 week [204]. The same year, IGFBP-1 and IGFBP-2 serum levels were found to be significantly decreased in atypical MDD patients when compared against controls and melancholic MDD patients. However, no significant alteration was found between melancholic patients and controls. The atypical subtype was defined as patients who mainly experienced weight gain and an overreacting character, whereas the melancholic group experienced decreased appetite, weight loss, and a lack of responsiveness [205]. Milanesi et al. found no significant difference in IGFBP-2 serum levels between MDD patients and controls. Moreover, no difference was found for drug-free and drug-naïve MDD patients [206].

#### 5.4.3. Still Unexplored IGFBP-4 and IGFBP-6 in MDD Patients

In a rat model of depression, it was found that acute lipopolysaccharide (LPS) injections significantly increased IGFBP-4 levels in the frontal cortex but not the hippocampus [207]. Moreover, the offspring of prenatal stressed rats only showed a significant increase in IGFBP-4 levels in both the hippocampus and the frontal cortex, whereas IGFBP-6 and other IGFBPs were unaltered [208]. IGF-1 might reduce IGFBP-4 levels through the activation of specific proteases, such as protease PAPP-A, and this process might differentially occur in stress-like scenarios [209]. To the best of our knowledge, no study has evaluated IGFBP-4 or IGFBP-6 in MDD patients.

Table 2 offers a summary of the main articles (the ones that passed the PRISMA flow chart) included in the review.

**Table 2 ijms-26-02561-t002:** Data from 34 articles that have measured peripheral levels of the IGF family in Depression and/or Major Depressive Disorder (MMD) throughout years (1987–2024).

Ref.	DSM	Group	Sample Size	Age (Years)	Sex (F/M)	Treatment	Time	Scale	Sample Source	Techn	Country	IGFs		Statistics		IGFBPs		Statistics	
[108] *	III	MDD	11	53.2 ± 11.6	7/4	DF	>3 d	28.5 ± 6.3 ^a^	Fasting Plasma	RIA	Germany	IGF-1 (U/mL)	1.1 ± 0.2	↑	*p* < 0.05				
		HC	11	49.2 ± 11.3	7/4								0.6 ± 0.1						
[109] *	III	MDD	34	48.2 ± 12.2	23/11	DF	14 d	26.9 ± 5.4 ^a^	Fasting Plasma	RIA	Germany	IGF-1 (U/mL)	1.41 ± 0.79	↑	*p* < 0.001				
		HC	34	44.7 ± 11.9	7/6								0.81 ± 0.31						
[110] *	III	MDD	12	53.1 ± 1.2	7/5	DF	7 d	29.1 ± 1.7 ^a^	Fasting Plasma	RIA	Germany	IGF-1 (U/mL)	1.18 ± 0.16	↑	*p* < 0.05				
		HC	12	51.0 ± 3.5	7/5								0.65 ± 0.07						
[111] *	III	MDD	10	52.9 ± 12.2	7/3	DF	7 d	28 ± 5.9 ^a^	Fasting Plasma	RIA	Germany	IGF-1 (U/mL)	1.09 ± 0.55	↑	*p* < 0.05				
		HC	10	51.2 ± 12	7/3								0.64 ± 0.23						
[112] *	III-R	MDD	15	49.6 ± 9.2	7/8	DF	14 d	28.3 ± 6 ^a^	Fasting Plasma	RIA	Germany	IGF-1 (U/mL)	1.05 ± 0.16	↑	*p* < 0.06				
		HC	15	51.1 ± 8.8	7/8								0.63 ± 0.06						
[113] *	III	MDD	10	48.1 ± 7	7/3	DF	14 d	26.4 ± 5.1 ^a^	Fasting Plasma	RIA	Germany	IGF-1 (U/mL)	1.08 ± 0.39	↑	*p* < 0.05				
		HC	10	49.6 ± 10.2	7/3								0.57 ± 0.16						
[114] *	III-R	MDD0	16	38 ± 15.1	11/5			23 ± 7.8 ^a^	Plasma	RIA	Germany	IGF-1 (U/mL)	1.34 ± 0.45						
		PreDex		N/A	N/A	DF	4 w						~0.73 *^2^	↑	*p* < 0.01				
		PostDex		N/A	N/A								~0.92 *^2^						
		MDD1	16	38 ± 15.1	11/5	Treated		6 ± 3.7 ^a^					1.12 ± 0.18						
		PreDex		N/A	N/A	DF	4 w						~0.73 *^2^	↑	*p* < 0.05				
		PostDex		N/A	N/A								~0.83 *^2^						
		HC	28	33 ± 8.7	13/15			2 ± 3.1 ^a^					1.06 ± 0.29						
		PreDex		N/A	N/A								~0.92 *^2^	ns	N/A				
		PostDex		N/A	N/A								~0.96 *^2^						
		PreDex-CS	30	N/A	N/A	DF	4 w						~0.72 *^2^	ns	*p* = 0.06				
		PostDex-CS		N/A	N/A								~0.75 *^2^						
		PreDex-NS	10	N/A	N/A	DF	4 w						~0.96 *^2^	↑	*p* < 0.02				
		Post-Dex-NS		N/A	N/A								~1.27 *^2^						
[115]	III-R	MDD0	10	45 ± 13.3	8/2	MOC	300 mg/d	26 ± 2.2 ^H^	Fasting Plasma	RIA	Germany	IGF-1 (U/mL)	1.39 ± 0.4	↑ ^HC^	*p* < 0.05				
		MDD0	13	48.4 ± 14.7	9/4	MAP	150 mg/d	28.9 ± 1.4 ^H^					1.31 ± 0.3	↑ ^HC^	*p* < 0.05				
		HC	15	N/A	N/A								0.76 ± 0.31						
		MDD1	10	45 ± 13.3	8/2	MOC	4 w	10.3 ± 1.9 ^H^					1.37 ± 0.17	ns ^MDD0^	N/A				
		MDD1	13	48.4 ± 14.7	9/4	MAP		11.3 ± 2 ^H^					1.73 ± 0.47	ns ^MDD0^					
[116]	III-R	MDD-C	9	13 ± 3	2/7	DN or DF	>3 m	57.2 ± 10.3 ^b^	Fasting Plasma	RIA	Italy	IGF-1 (ng/mL)	338.8 ± 155.7	ns	N/A				
		HC-C	9	14 ± 5	2/7								247.3 ± 171.6						
		MDD-GHRH	9	13 ± 3	2/7	DN or DF	>3 m	57.2 ± 10.3 ^b^					310.1 ± 113.9	ns	N/A				
		HC-GHRH	9	14 ± 5	2/7								378.4 ± 200						
[117]	III-R	MDD-C	12	43 ± 12	8/4	DF	>7 d	25.2 ± 1.3 ^a^	Fasting Plasma	RIA	Germany	IGF-1 (ng/mL)	193 ± 43	ns	*p* = 0.757				
		HC-C		42 ± 11									200 ± 61						
		MDD-GHRH		43 ± 12		DF	>7 d	25.2 ± 1.3 ^a^					186 ± 49	ns	*p* = *0*.206				
		HC-GHRH		42 ± 11									216 ± 62						
[132]	III	MDD	10	41.0 ± 8.0	All F	DF	14 d	N/A	Serum	RIA	US	IGF-1 (ng/mL)	189 ± 86	ns	*p* = 0.98				
		HC	10	41.0 ± 7.0									189 ± 37						
[118]	III	MDD	24	47.2 ± 16.4	11/13	WO	6 d	31.8 ± 5.8 ^a^	Plasma	RIA	Germany	IGF-1 (ng/mL)	157 ± 40	↑	*p* < 0.001	IGFBP-2 (ng/mL)	286 ± 220	ns	N/A
		HC	33	51.4 ± 19.2	11/22								120 ± 33				236 ± 134		
		MDD0	15	41.4 ± 15.4	6/9	WO	6 d	32.1 ± 5.6 ^a^					168 ± 41	ns	N/A		338 ± 252		
		MDD1				Treated	24–55 d	11.8 ± 8.3 ^a^					152 ± 33				320 ± 168		
		MDD-R0	9	42.9 ± 17	3/6	WO	6 d	30.1 ± 4.5 ^a^					174 ± 46	↑	*p* < 0.05		332 ± 310		
		MDD-R1				Treated	24–55 d	5.9 ± 2.6 ^a^					147 ± 33				301 ± 209		
		MDD-NR0	6	39.2 ± 13.8	3/3	WO	6 d	35 ± 6.3 ^a^					158 ± 33	ns	N/A		347 ± 153		
		MDD-NR1				Treated	24–55 d	20.7 ± 4.8 ^a^					161 ± 34				350 ± 90		
																IGFBP-3 (ng/mL)	2325 ± 329	ns	N/A
																	2203 ± 391		
																	2371 ± 327		
																	2334 ± 358		
																	2372 ± 284		
																	2348 ± 315		
																	2371 ± 413		
																	2563 ± 409		
[133]	III	MDD	19	34.7 ± 8.8	All F	DF		18.8 ± 3.9 ^c^	Serum	N/A	USA	IGF-1 (ng/mL)	289 ± 108	ns	*p* = 0.07				
		HC	16	36.1 ± 6.6									228 ± 58						
[142]	N/A	MDD0	37	40.0 ± 11.4	28/9	Fluox	5 d	1.46 ± 5.36 ^c^	Plasma	N/A	USA	IGF-1 (ng/mL)	161.2 ± 63.8	ns	*p* = 0.732				
		MDD1				Plac		0.95 ± 4.4 ^c^					162.5 ± 58.7						
		MDD0	34	38.7 ± 14.5	26/8	Sert		1.56 ± 5.66 ^c^					170.2 ± 73.9	ns	*p* = 0.61				
		MDD1				Plac		3.59 ± 5.86 ^c^					174.2 ± 67.6						
		MDD0	36	39.9 ± 11.1	22/14	Parox		0.68 ± 5.42 ^c^					163 ± 63.9	↑	*p* = 0.007				
		MDD1				Plac		6.22± 6.63 ^c^					186.1 ± 74.0						
[119]	IV	MDDR	25	51 ± 17	18/7	WO-6d	base	23.9 ± 5.2 ^H^	Serum	RIA	Germany	IGF-1 (ng/mL)	175 ± 40	↓	*p* < *0*.01	IGFBP-3 (ng/mL)	3.07 ± 0.55	↓	*p* < 0.01
						Ami	14 d	N/A					162 ± 49				3.09 ± 0.59		
							35 d	6.8 ± 3.6 ^H^					144 ± 45				2.87 ± 0.56		
		MDDNR	9	46 ± 16	8/1	WO-6d	base	22.1 ± 3.9 ^H^					170 ± 47	ns			2.96 ± 0.24	ns	N/A
						Ami	14 d	N/A					173 ± 51				3.17 ± 0.33		
							35 d	18.4 ± 5.6 ^H^					174 ± 49				3.07 ± 0.33		
		MDDR	27	58 ± 16	17/10	WO-6d	base	23.0 ± 3.2 ^H^					164 ± 52	↓	*p* < 0.01		2.92 ± 0.39	ns	N/A
						Paro	14 d	N/A					147 ± 57				2.89 ± 0.46		
							35 d	6.0 ± 2.9 ^H^					148 ± 57				2.80 ± 0.55		
		MDDNR	16	57 ± 14	12/4	WO-6d	base	23.7 ± 3.5 ^H^					152 ± 79	ns			2.99 ± 0.84	ns	N/A
						Paro	15 d	N/A					142 ± 70				2.92 ± 0.70		
							35 d	19.2 ± 5.2 ^H^					142 ± 70				3.00 ± 0.99		
[24]	DSM N/A	MDD0	12	59.1 ± 10	5/7	WO	>6 d	24.4 ± 5.3 ^a^	Fasting CSF	RIA	Germany	IGF-1 (µg/L)	0.235 ± 0.135	↑	*p* < 0.05				
		MDD1				Venla, Fluox, Doxe, Ami		17.1 ± 9.4 ^a^					0.305 ± 0.096						
[143]	IV-R	MDD0	8	52.9 ± 8.8	6/2	DF	>2 w	26.9 ± 6.9 ^H^	Serum	CLIA	Europe	IGF-1 (f.c.)	1.59	↑	*p*= 0.0156	IGFBP-2 (f.c.)	1.06	↑	*p*= 0.0156
		MDD1				ECT		19.7 ± 8 ^H^											
[135]	IV	MDD	15	32.25 ± 7.65	All M	base			Serum	RIA	China	IGF-1 (ng/mL)	167.3 ± 6.6	ns	*p* > 0.05				
		MDD1	12	N/A		Esci	8 w	<22 ^M^					175.11 ± 8.59						
		HC	12	31.15 ± 10.19									159.6 ± 11.8						
[146]	IV	MDD0	41	38.9 ± 11.7	30/11			19.0 ± 3.9 ^c^	Serum	ELISA	Denmark	IGF-1 (ng/mL)	86.6 ± 110	ns	*p* > 0.05				
		MDD1				A-Exer	3 m	N/A					81.7 ± 114.6						
		MDD0	38	43.8 ± 12.2	23/15			18.9 ± 4.6 ^c^					88.6 ± 104						
		MDD1				C-Exer	3 m	N/A					67.4 ± 110.3						
[120]	IV	MDD0	78	48.64 ± 13.88	35/34	Treated		26.37 ± 6.73 ^a^	Serum	ELISA	Germany	IGF-1 (ng/mL)	189.6 ± 79.7	↑ ^HC^	*p* = 3.29 × 10^−4^				
		MDD1					6 w	9.67 ± 6.54 ^a^					184.90 ± 87.29	↑ ^HC^	*p* = 0.002				
		MDDR0	39	49.95 ± 11.91	18/21			24.46 ± 6.63 ^a^					169.02 ± 60.58	ns	N/A				
		MDDR1					6 w	4.44 ± 3.04 ^a^					167.52 ± 67.24						
		MDDNR0	39	47.33 ± 15.64	17/22			28.28 ± 6.34 ^a^					210.17 ± 91.34	↑ ^MDDNR1^	*p* = 0.046				
		MDDNR1					6 w	14.9 ± 24.57 ^a^					202.26 ± 101.49	ns	*p* = 0.11				
		HC	92	48.13 ± 13.7	42/50								155.6 ± 60.0						
[122]	IV	Re D/A	502	44.2 ± 13.1	348/154	DF		12 (7–19) ^I^	Fasting Plasma	CIA	Holland	IGF-1 (nmol/L)	26.2 ± 6.9	ns	*p* = 0.09				
		Cu D/A	963	40.3 ± 12.8	654/309	DF		27 (18–35) ^I^					26.5 ± 6.8	↑ ^HC^	*p* = 0.006				
		MDD	647	42.6 ± 11.5	429/218	Treated		30 (19–40) ^I^					24.9 ± 6.9	↓ ^HC^	*p* = 0.028				
		HC	602	41.0 ± 14.6	370/232			6 (3–12) ^I^					25.8 ± 6.9						
[124]	IV	MDD0	37	42.4 ± 11.9	29/8	DF	3 m	19.7 ± 2.6 ^c^	Serum	ELISA	Italy	IGF-1 (ng/mL)	128.1 ± 48.3	ns					
		MDD1				SRIs	1 w	N/A					122 ± 46.8	ns	*p* = 0.42				
		HC	43	42.3 ± 11.3	28/15								121.2 ± 51.6						
[184]	IV	MDD	40	58.7 ± 9.92	31/9	N/A		19.45 ± 4.75 ^H^	Serum	ELISA	China	IGF-1 (ng/mL)	136.6 ± 39.02	↑ ^N-PSD^	*p* < 0.05				
		N-PSD	42	61.1 ± 6.58	17/25			3.36 ± 2.02 ^H^					109.62 ± 34.54	ns	N/A				
		PSD	39	62.44 ± 10.34	19/20			16 ± 5.45 ^H^					113.68 ± 51.46	ns	N/A				
		HC	38	57.58 ± 5.28	16/22			2.18 ± 1.98 ^H^					124.29 ± 49.48						
[205]	IV	MDD Me	231	41.7 ± 12	68% F	Treated		38.6 (9.7) ^I^	Fasting Serum	CLIA	Holland					IGFBP-1 (β.c.)	0.062	ns ^HC^	*p* = 0.135
		MDD At	128	40.7 ± 11.7	71.9% F	Treated		39.1 (8.8) ^I^									−0.256	↓ ^Me^	*p* = 1.398 × 10^−6^
		HC	414	39 ± 14.8	60.6% F			8.0 (7.1) ^I^									− 0.194	↑ ^At^	*p* = 7.296 × 10^−5^
																IGFBP-2 (β.c.)	0.023	ns ^HC^	*p* = 0.198
																	− 0.094	↓ ^Me^	3.246 × 10^−5^
																	− 0.072	↑ ^At^	*p* = 0.001
[206]	IV	MDD	43	52.18 ± 12.5	35/8	DN or DF		23 ± 3.52 ^c^	Fasting Serum	ELISA	Italy					IGFBP-2 (ng/mL)	225.82 ± 129.11	ns	*p* = 0.09
		HC	93	49.56 ± 12.9	41/52												232.9 ± 125.48		
[121]	IV	MDD	91	44.1 ± 13.1	32/59	Treated		13.9 ± 9 ^H^	Fasting Serum	N/A	Japan	IGF-1 (ng/mL)	152.0 ± 50.0	IGF-1-HAMD (R = 0.349, *p* = 0.001). No correlation with cortisol.					
[128]	V	MDD	86	32.64 ± 8.89	52/34	N/A		19.02 ± 2.41 ^H^	Serum	ELISA	Bangladesh	IGF-1 (ng/mL)	3.43 ± 4.66	↑	*p* = 0.006				
		HC	85	31.13 ± 8.72	51/34								2.08 ± 0.86						
[127]	V	MDD	41	36.4 ± 12.8	27/14	DF		28(21.5,31.5) ^M^	Serum	ELISA	Ukraine	IGF-1 (ng/mL)	289.2 ± 125.3	↑ ^HC^	*p* < 0.0001				
		MDD0	30	35.1 ± 12.9	20/10			29 (24.5–33) ^M^					288.2 ± 132.6						
		MDD1	30			Vortio	8 w	5 (2.5–10) ^M^					173.4 ± 71.2	↓ ^MDD0^	*p* < 0.0001				
		HC	32	38.0 ± 12.2	20/12			2 (0–3.5) ^M^					170.2 ± 58.2	ns^MDD1^	*p* = 0.18				
[126]	V	MDD	78	38.2 ± 11.9	48/30	DF		29 (22–33) ^M^	Fasting Serum	ELISA	Ukraine	IGF-1 (ng/mL)	228 (183–312)	↑ ^HC^	*p* < 0.0001				
		MDD0	48	N/A	N/A			29 (22–33) ^M^					236 (184–316)						
		MDD1				Vortio	8 w	6 (3–11) ^M^					170 (132–210)	↓ ^MDD0^	*p* < 0.0001				
		HC	47	37.8 ± 12.3	27/20			2 (0–4) ^M^					153 (129–186)						
[129]	V	MDD	54	42.4 ± 14.7	All M	Treated		19.3 ± 7 ^H^	Fasting Serum	N/A	Japan	IGF-1 (ng/mL)	171.5 ± 61.8	↑	*p* = 0.011				
		HC	37	39.4 ± 7.0									144.1 ± 39.2						
[193]	V	MDD	51	52.71 ± 14.57	30/21	Treated			Fasting Plasma	ELISA	Spain	IGF-2 (ng/mL)	249.86 ± 119.57	↑ ^HC^	*p* < 0.001	IGFBP-1 (ng/mL)	12.84 ±14.31	ns	*p* = 0.061
		MDD0	15	61.60 ± 10.60	7/8			24.93 ± 6.18 ^a^					241.29 ± 99.86				10.02 ± 6.61		
		MDD1						15 ± 4.11 ^a^					156.71 ± 35.64	↓ ^MDD0^	*p* < 0.01		7.35 ± 4.16	ns	*p* = 0.258
		HC	48	42.58 ± 11.54	22/26								116.69 ± 53.40		*p* < 0.01		7.57 ± 5.08		
															MDD ^	IGFBP-3 (ng/mL)	506.12 ± 139.91	ns	*p* = 0.809
															MDD0 ^		456.49 ± 118.11		
															MDD1 ^		270.14 ± 39.86	↓ ^MDD0^	*p* < 0.001
															HC ^		498.26 ± 89.57		
															MDD ^	IGFBP-5 (ng/mL)	86.95 ± 20.69	ns	*p* = 0.904
															MDD0 ^		84.24 ± 13.56		
															MDD1 ^		42.78 ± 6.14	↓ ^MDD0^	*p* < 0.001
															HC ^		86.14 ± 15.12		
															MDD ^	IGFBP-7 (ng/mL)	82.90 ± 26.44	↑ ^HC^	*p* < 0.01
															MDD0 ^		82.36 ± 25.50		
															MDD1 ^		116.19 ± 29.85	↑ ^MDD0^	*p* < 0.05
															HC ^		67.01 ± 25.14		
[58]	V	MDD	129	40.5 ± 12.8	69/60	Treated		13.9 ± 8.4 ^c^	Fasting Serum	RIA	Japan	IGF-1 (ng/mL)	160.0 ± 54.3	↑	*p* < 0.01				
		HC	71	41.4 ± 9.3	38/33								137.9 ± 41.3						
[130]	V	MDD NR	84	38.6 ± 13.2	38/46	Treated		18.1 ± 6.5 ^c^	Fasting Serum	RIA	Japan	IGF-1 (ng/mL)	166.9 ± 54.9	↑ ^HC, R^	*p* = 0.001				
		MDD R	36	44.2 ± 10.7	17/19			4.3 ± 2.2 ^c^					138.8 ± 40.0		*p* = 0.007				
		HC	99	40.8 ± 9.3	44/55								139.9 ± 42.4						
[131]	ICD-10	MDD FE	60	35.48 ± 11.77	37/23	DN		23.32 ± 5.69 ^c^	Fasting Serum	ELISA	China	IGF-1 (ng/mL)	149.81 ± 34.35	↑	*p* = 0.000				
		HC	60	34.63 ± 14.05	39/21								128.23 ± 25.83						

**Explanation by column.** * *Refs*: [108,109,110,111,112,113,114]: some patients had a BD diagnosis. *DSM*: version of The Diagnostic and Statistical Manual of Mental Disorders (DSM) used to diagnose MDD. ICD-10: International Classification of Diseases, 10th Revision. *Group*. 0: measure at baseline; 1: measure after a given time; PreDex: measure before dexamethasone test; PostDex: measure after dexamethasone test; CS: cortisol suppressors; NS: cortisol non-suppressors; -C: clonidine challenge; -GHRH: growth hormone releasing hormone challenge; R: responders to treatment; NR: non-responders to treatment; Re: remitted patients; D/A: depressive or anxiety; Cu: current disease; N-: non-; PSD: post-stroke depression; Me: melancholy subtype; At: atypical subtype; FE: first episode. *Treatment*. DF: drug-free; DN: drug-naïve; MOC: moclobemide; MAP: maprotiline; WO: washed-out from treatment; Fluox: fluoxetine; Plac: placebo; Sert: sertraline; Parox: paroxetine; Ami: amitriptyline; Venla: venlafaxine; Doxe: doxepin; ECT: electroconvulsive therapy; Esci: escitalopram; A-Exer: aerobic exercise; C-Exer: control exercise; SRIs: serotonin reuptake inhibitors; Vortio: vortioxetine; d: days; w: weeks; m: months; base: baseline. *Scale*: used to measure depressive symptom severity, such as the Hamilton Depression Rating Scale (or HAMD). ^a^: HDRS 21-item scale; ^b^: Poznanski Rating Scale; ^c^: HDRS 17-item scale; ^H^: HDRS non-specified; ^M^: Montgomery–Asberg Depression Rating Scale; ^I^: Depression severity IDS (the inventory of depressive symptomatology) score. *Techn*: technique used to measure peripheral levels of IGF. RIA: radioimmunoassay; ELISA: enzyme-linked immunoassay; CLIA: chemiluminescence immunoassay. *IGFs*: *^2^: these values were visually extracted from graphs. *IGFBPs*: (f.c.): fold change; (β.c.): β coefficient of multiple regression models. *Statistics*. ns: non-significant; *↑*: significantly elevated; *↓*: significantly reduced. *Superscripts* indicate the reference statistical comparison. N/A: data not found. ^ To improve visual clarity, the group corresponding to the IGFBP measurements is indicated.

## 6. IGF Peripheral Levels in Bipolar Disorder

### 6.1. Bipolar Disorder

Bipolar disorder (BD) is characterized by the intercalation of manic/hypomanic and/or depressive episodes that includes periods of mood stability called euthymia [210]; its prevalence is about 2.4% [211]. The search for peripheral biomarkers in BD patients has aimed to determine whether these biomarkers reflect a general trait-marker associated with the diagnosis of BD, or if they vary with mood states, functioning as state-specific markers [212]. The subjective nature and symptom overlapping with SZ or MDD complicates the diagnosis of BD. Actually, one of three BD cases might be firstly diagnosed as MDD or SZ [213]. Indeed, treatment is similar to SZ or MDD, with mood stabilizers such as lithium or valproate being the gold standard [214]. Studies evaluating the IGF family are mainly focused on BD-I and BD-II diagnosis. BD-I patients experience manic episodes and major depressive episodes through their clinical course. A manic episode is described as periods of abnormal and a persistent elevated or irritable mood, accompanied by inflated self-esteem, a decreased need for sleep, more talkativeness, distractibility, or general increased energy. These symptoms extend for at least 1 week, present most of the day, nearly every day. BD-II patients experience major depressive episodes and, at least, one hypomanic episode, but without a history of mania. Hypomanic episodes are similar to manic episodes but with less duration and intensity, persisting for at least 4 consecutive days only. Normally BD-I starts between 20–30 years and BD-II slightly later [215]. The Young Mania Rating Scale (YMRS) is the gold standard to evaluate manic symptoms for BD patients [216] and HDRS in depressive moods [103].

### 6.2. IGF-1 in BD Patients

#### 6.2.1. IGF-1 and IGFBP-1 and Weight Gain Induced by Mood Stabilizers in BD Women

The first study measuring peripheral IGF-1 and IGFBP-1 levels in BD patients was made in the context of weight gain in a cohort composed only by women, who were taking either valproic acid or lithium as treatment. IGF-1 and IGFBP-1 were found to be significantly decreased in women treated with valproic acid when compared to the lithium-treated group. Both groups did not exhibit significant differences in absolute weight, BMI, total cholesterol, HDL, LDL, glucose, triglycerides, testosterone, luteinizing hormone (LH), and TSH. On the contrary, free testosterone, androstenedione, estradiol, DHEAS, and C-peptide were significantly higher in the valproate-treated group, whereas the opposite was found for sex hormone-binding globulin (SHBG) progesterone, follicle-stimulating hormone (FSH), and prolactin in the lithium-treated patients. McIntyre et al. suggested the possibility that IGF-1 decrease could have been mediated by insulin or even the nutritional state [217]. Lower IGFBP-1 in the valproate group could be explained by higher insulin that could be inferred for higher C-peptide [218], since hepatic IGFBP-1 can be inhibited by insulin [219]. This study lacked a control group and patients were a mix of BD-I and BD-II patients [217]. Results from all these studies can be found in Table 3.

#### 6.2.2. IGF-1 in FE BD Patients

As was commented in Section 4.2.6. Palomino et al. found that IGF-1 plasma levels were significantly reduced in FE BD patients compared to FE SZ patients, but not controls. Nonetheless, no significant difference was found between controls and BD patients after 1, 6, or 12 months of treatment. Moreover, no significant correlation was found between IGF-1 and symptom severity (YMRS, HDRS) in BD patients both at FE and after 12 months [73]. This study questioned whether differences in IGF-1 were inherently related to a specific diagnosis, such as SZ or BD, or were a general trait marker of PDs with psychotic symptoms.

#### 6.2.3. IGF-1 and Other Neurotrophins in BD-I Patients

Previous research has suggested that the *IGF-1* gene has a dominant effect on BD [220]. Moreover, *IGF-1* gene expression was found to be significantly increased in BD patients who responded to lithium when compared against non-responders and controls in a lymphoblastoid cell line [221]. In that study, baseline IGF-1, but no other neurotrophins (BDNF or β-NGF), was significantly increased in BD-I patients when compared to controls. IGF-1, BDNF, and β-NGF remained unaltered after 6 months of treatment. Moreover, no significant difference was found for neurotrophins between drug-naïve and treated patients at baseline. No significant association was found between neurotrophins and the YMRS scale, neither before nor after treatment [222]. Following Trejo et al. [223], Kim et al. suggested that an IGF-1 increase in BD patients could be part of a peripheral compensatory system as a response to reduced IGF-1 central expression, possibly as an over-activated glycogen synthase kinase-3 pathway that is not appropriately regulated in BD patients. It was also stipulated that an IGF-1 increase in BD patients might be reflecting the beginning of irreversible neurodegenerative processes, but they lacked a cognitive assessment to explore this hypothesis [222].

In 2014, Liu et al. studied IGF-1 serum levels and other angioneurins (BDNF, VEGF, FGF-2, and NGF) in BD patients [224] under the theoretical basis of Duman’s hypothesis [104]. Angioneurins are factors that play an important neuroprotective role in the regulation of the BBB. IGF-1, FGF-2, and NGF were found to be significantly elevated in BD patients in comparison to controls. BD patients were in a manic episode by the time the sample was extracted and were mainly drug-naïve or drug-free. IGF-1 did not significantly correlate with manic symptoms (YMRS) [224].

Another study found significantly higher serum levels of IGF-1 in BD-I patients when compared to controls, even after correcting for age, BMI, and cortisol levels. Cortisol, which was believed to reduce neurotrophins levels, was unaltered between groups. Conversely, GDNF and Neuritin-1 (Nrn1) were found to have significantly decreased, whereas there was no significant alteration in other factors (BDNF, NGF, neurotrophin-3 (NT-3), VEGF, and FGF-2). There was no statistical difference on IGF-1 between valproate-treated and lithium-treated BD-I patients. Tunçel et al. postulated that IGF-1 alterations may be a mood state marker during mania rather than a general trait marker of BD patients [225].

#### 6.2.4. IGF-1 and Inflammation in BD Euthymic Patients

Low-grade inflammation has also been observed in BD patients [226]. A meta-analysis concluded that peripheral cytokines, such as TNF-α, are significantly increased in both manic and depressive mood states, but not euthymia [227,228,229,230]. TNF-α elevation seems to be correlated with decreased neuroprotective effects of IGF-1 at CNS [223].

Serum levels of IGF-1 were found to be significantly elevated in BD euthymic patients compared to controls, while GH, insulin, or TNF-α were not significantly altered [231]. Equally, da Silva et al. did not find a significant association between IGF-1 and subjective scales (YMRS, HAMD) [231]. The hypothetical reason for IGF-1 elevation was similar to previous works [222,224].

#### 6.2.5. IGF-1 and the Glutamatergic System in BD Patients

Serum IGF-1 was evaluated in relation to the glutamatergic system in BD patients. Alterations in the glutamatergic system includes elevated levels of glutamate, an increased ratio of glutamate–glutamine, alterations in *N*-methyl-D-aspartate receptors (NMDAR) gene expression, and changes in the activity of glutamic acid decarboxylase (GAD) [232]. IGF-1 is potentially involved in the modulation of NMDAR. The study of Ferensztajn-Rochowiak et al. compared serum levels of NMDA, GAD, and IGF-1 between BD patients during acute mania and depression, after remission, and under long-term lithium treatment. NMDA, GAD, and IGF-1 levels were significantly elevated during acute mania compared to lithium treated patients, but only IGF-1 persisted at elevated levels after the mania had remitted. Interestingly, GAD was significantly reduced after remission in both manic and depressive patients, and was also elevated during acute depression compared to lithium-treated patients. Despite BD patients having similar BMI, manic and depressive patients were significantly younger than the lithium-treated group. Importantly, lithium-treated patients were not just under lithium monotherapy but were receiving Aps and Ads, which may have influenced IGF-1. In summary, the absence of significant differences in IGF-1 between manic and depressive BD patients or after remission might support the hypothesis of IGF-1 elevation being a trait marker in BD patients [233].

#### 6.2.6. IGF-1 Meta-Analysis in BD Patients

The first meta-analysis conducted in BD patients was already discussed in the MDD Section 5.2.11 [136]. The meta-analysis of Tu et al. included only three previous studies [73,222,224], and concluded that peripheral IGF-1 levels are significantly elevated in BD patients, therefore considering IGF-1 to be a potential trait marker in BD patients. There was no significant association between IGF-1 levels with age, sex distribution, BMI, age at onset, and the YMRS. Nevertheless, potentially confounding parameters were evaluated in both BD and MDD patients [136].

The meta-analysis of Chen et al. added two more studies [73,222,224,225,231] and again concluded the significant elevation of IGF-1 in BD patients. Ethnicity was introduced as a potential confounding factor. Actually, IGF-1 was significantly higher in American and Asian patients, but not for patients with a European ancestry. Nonetheless, this study again analyzed MDD and BD patients as a whole [139].

#### 6.2.7. IGF-1 in Children with Double Diagnosis of ASD and BD

Guldiken et al. did not find any significant differences in IGF-1 between children with double diagnosis of ASD + BD and ASD alone. Conversely, VEGF and FGF-2 were found to be significantly higher in the double diagnosis group. However, the double diagnosis group had a higher prevalence of tic disorder and oppositional defiant disorder, as well as higher rates of ASD severity, self-harming behavior, sleep disturbance, and pharmacological treatment. Curiously, IGF-1 was significantly correlated with the number of previous mood episodes in the ASD + BD group [234].

### 6.3. First Studies on IGF-2 in BD Patients

In 2024, two studies explored IGF-2 in BD patients for the first time [235,236], one performed by our group [236]. Ye et al. found that serum levels of IGF-2 were significantly reduced in BD patients when compared to controls. Moreover, IGF-2 was still reduced when BD patients were separated into patients in a manic or depressive mood state [235]. Conversely, we found no significant alteration in the plasma levels of IGF-2 between BD patients in a manic state and controls [236]. From our perspective, there are several factors, such as sample size, age, pharmacological treatment, ethnicity, source of sample, and severity of manic symptoms, which were different between the studies [235,236] and that might be behind the observed differences. Therefore, the differences in these factors between both works [235,236] are shown in Figure 5.

### 6.4. IGFBPs in BD Patients

#### 6.4.1. IGFBP-1, IGFBP-3, IGFBP-5, and IGFBP-7 in BD Patients

In our work, we also found that baseline IGFBP-3 and IGFBP-5 were significantly decreased in BD patients when compared to controls. Conversely, it was not the case for IGFBP-1 and IGFBP-7. After treatment, only IGFBP-1 significantly decreased. Curiously, IGFBP-7 was the only IGFBP that was significantly and positively correlated with IGF-2 levels in both BD patients and controls and negatively correlated with age. Finally, no correlation was found for any IGF member and manic symptom severity (YMRS) both at baseline or after treatment, nor between males and females [236].

#### 6.4.2. IGFBP-2 Research in BD Patients

There are some reasons behind IGFBP-2 as the most studied IGFBP in BD patients. First, IGFBP-2 has exhibited inhibitory and stimulatory actions on IGF-1 [237]. Second, *IGFBP-2* gene expression was found to be statistically reduced in the prefrontal cortex of BD patients when compared to controls. Curiously, *IGFBP-2* downregulation was especially marked in non-lithium-treated patients [238].

Third, IGFBP-2 is not susceptible to post-prandial variations in contrast to IGFBP-1 [239] and is more abundant in the CNS than IGFBP-3 [240,241]. First, IGFBP-2 was found to be significantly and inversely correlated with BMI in BD patients suffering from a major depressive episode [242]. In the offspring of BD patients, IGFBP-2 was found to be significantly decreased in contrast to controls. Suffering a mood episode significantly decreased IGFBP-2. Moreover, a cutoff value for IGFBP-2 of 150 ng/mL was established as a potential threshold for predicting a future mood disorder at 12 years follow up [243]. Interestingly, IGFBP-2 was found to be significantly reduced in BD patients. However, no difference was found for IGFBP-2 between BD patients who were following valproate or lithium treatment. Despite the fact that *IGFBP-2* gene expression was also found to be significantly reduced in fibroblasts extracted from BD patients, no statistical association was found with IGFBP-2 serum levels. Finally, Milanesi et al. suggested that alterations in IGFBP-2 in BD patients could be related to a deregulation in the immune system and inflammatory processes [206].

#### 6.4.3. IGFBP-4 and IGFBP-6 Still Unexplored in BD Patients

The study mentioned in the SZ Section 4.4.3 detected that IGFBP-6 was significantly and negatively correlated with BD by the Linkage Disequilibrium Score Regression (LDSC) regression coefficient calculated from a large genome-wide association study (GWAS) conducted with 20,129 BD patients and 21,524 controls [98]. Weissleder et al. found no significant difference in the gene expression of IGFBP-3, IGFBP-4, IGFBP-5, and IGFBP-6 in the subventricular zone of BD patients when compared to controls [96]. Currently, peripheral IGFBP-4 and IGFBP-6 remain unexplored in BD patients.

**Table 3 ijms-26-02561-t003:** Data from 13 articles that have measured peripheral levels of the IGF family in Bipolar Disorder (BD) throughout years (2003–2024).

Ref.	DSM	Group	Sample Size	Age (Years)	Sex (F/M)	Treatment		Sample Source	Techn	Country	IGFs		Statistics		IGFBPs		Statistics	
[217]	IV	BD0 All	18	31.5	All F	Val	4–136 m	Fasting Serum	ELISA	Europe *^2^	IGF-1 (ng/mL)	124.4 ± 13.89	↑	*p* = 0.001	IGFBP-1 (ng/mL)	21.76 ± 5.68	↑	*p* = 0.034
		BD1 All	20	33.2		Lit	3–156 m					232.8 ± 41.13				34.91 ± 8.42		
[73]	IV	BD0	23	27	8/15	DN		Fasting Plasma	ELISA	Spain	IGF-1 (ng/mL)	126.15 ± 66.09	ns ^HC^	*p* = 0.143				
		BD1m	15	N/A	N/A	Treated	1 m					194.98 ± 87.19		*p* = 0.123				
		BD6m	14	N/A	N/A	Treated	6 m					173.11 ± 78.44		*p* = 0.47				
		BD12m	11	N/A	N/A	Treated	12 m					149.54 ± 60.82		*p* = 0.807				
		HC	23	25.7	13/30							155.41 ± 67.03						
[222]	IV	BD M	116	35.9 ± 11.8	74/42	DN (*n* = 79) Treated (*n* = 12)		Blood	ELISA	Korea	IGF-1 (pg/mL)	514.6 ± 259.8	↑	*p* < 0.0001				
						DF (*n* = 25)	>2 m											
		HC	123	35.5 ± 10.4	67/56							316.8 ± 270.0						
[224]	IV	BD M	70	37.9 ± 14.5	29/41	DN (*n* = 64) + DF (*n* = 6)		Fasting Serum	ELISA	China	IGF-1 (ng/mL)	162.0 ± 72.0	↑	*p* = 0.029				
		HC	50	36.8 ± 11.2	20/30							138.9 ± 80.1						
[242]	IV	BD	31	48 ± 13.09	21/10	Treated		Serum	bb-Ls	Europe					IGFBP-2 (ng/mL)	123.84 ± 67.22	Correlation with BMI	
[243]	IV	Offs BD NMD	96	16.4 ± 2.65	N/A	N/A		Serum	bb-Ls	Holland					IGFBP-2 (ng/mL)	175.5(147.5–199.9)	↑ ^HC^	*p* < 0.05
		Offs BD MD														150.1(110.8–209.4)	ns	*p* > 0.05
		HC	50	15.0 ± 1.88 ↓												133.2(117.5–159)		
[206] *^1^	IV	BD All	41	46.76 ± 14.5	27/14	Treated		Fasting Serum	ELISA	Italy					IGFBP-2 (ng/mL)	173.24 ± 77.95	*p* = 0.003	
		HC	93	49.56 ± 12.9	41/52											232.90 ± 125.48		
[231]	IV	BD E	31	41.7 ± 11.8	25/6	Treated		Fasting Serum	CLIA	Brazil	IGF-1 (ng/mL)	248.8 ± 104.9	↑	*p* = 0.001				
		HC	33	41.0 ± 11.9	27/6							169.2 ± 74.2						
[233]	IV	BD a M	19	42 ± 14	10/9	12 y of illness		Fasting Serum	ELISA	Poland	IGF-1 (ng/mL)	143 ± 51	↑ ^BD Lit^	*p* = 0.033				
		BD ra M	13			Treated	40 ± 20 d					175 ± 63	↑ ^BD Lit^	*p* = 0.019				
		BD a D	17	50 ± 14	11/6	18 y of illness						129 ± 56	ns	*p* > 0.05				
		BD ra D	12			Treated	46 ± 17 d					120 ± 52	ns	*p* > 0.05				
		BD Lit	18	61 ± 4	9/9	Lit	22 y					117 ± 37						
[225] *^1^	IV	BD All	45	34.9 ± 10.8	22/23	Treated		Fasting Serum	ELISA	Turkey	IGF-1 (ng/mL)	279.3 ± 139.5	↑	*p* = 0.0001				
		HC	45	34.9 ± 10.8	22/23							190.7 ± 56.6						
[235]	IV	BD	78	N/A	44/34	DN + WO	> 3 m	Fasting Serum	ELISA	China	IGF-2 (ng/mL)	66.08 ± 21.22	↓	*p* < 0.001				
		BD M	55	30.49 ± 8.44	34/21							67.19 ± 21.52	↓	*p* < 0.001				
		BD D	23	24.56 ± 7.24	10/13							63.43 ± 20.67	↓	*p* < 0.001				
		HC	50	29.28 ± 3.83	24/26							88.72 ± 31.55						
[236]	V	BD M	20	51.10 ± 9.85	10/10	Treated		Fasting Plasma	ELISA	Spain	IGF-2 (ng/mL)	135.25 ± 64.02	ns	*p* = 0.478	IGFBP-1 (ng/mL)	8.92 ± 4.06	ns	*p* = 0.597
		HC	20	48.80 ± 7.34	10/10							114.53 ± 50.59				8.17 ± 4.78		
		BD0	10	53.10 ± 10.18	5/5	P treated	17 d					168.91 ± 51.27	ns	*p* = 0.548		10.12 ± 4.66	↓ ^BD0^	*p* < 0.01
		BD1	10		5/5	Treated						162.97 ± 50.06				6.96 ± 3.35		
													BD M ^		IGFBP-3 (ng/mL)	264.11 ± 60.22	↓ ^HC^	*p* < 0.0001
													HC			474.33 ± 82.24		
													BD0			257.83 ± 38.98	ns	*p* = 0.429
													BD1			249.93 ± 39.63		
													BD M		IGFBP-5 (ng/mL)	33.35 ± 13.06	↓ ^HC^	*p* < 0.0001
													HC			82.46 ± 14.77		
													BD0			43.09 ± 7.94	ns	*p* = 0.169
													BD1			39.81 ± 5.51		
													BD M		IGFBP-7 (ng/mL)	143.21 ± 130.69	ns	*p* = 0.165
													HC			70.09 ± 28.89		
													BD0			205.08 ± 148.79	ns	*p* = 0.652
													BD1			154.86 ± 90.42		
[234]	V	ASD + BD	40	14.03 ± 2.97	N/A	Treated		Fasting Serum	ELISA	Turkey	IGF-1 (ng/mL)	9.10 (4.51–77.13)	ns	*p* = 0.855				
		ASD	40	13.08 ± 3.06		treated (*n* = 34)						10.87 (3.96–58.83)						

**Explanation by column.** *Ref*. *^1^: [206] Manic state patients (*n* = 5), Depressive state patients (*n* = 36). [225] Manic state patients (*n* = 13), Depressive state patients (*n* = 32). *DSM*: version of The Diagnostic and Statistical Manual of Mental Disorders (DSM) used to diagnose BD or BD axis. *Group*: BD: Bipolar Disorder; ASD: Autism Spectrum Disorder; All: mixed of Manic and Depressive mood; M: manic; D: depressive mood; E: euthymia; a: acute; ra: remitted acute; 0: measure at baseline; 1: measure after a given time; HC: healthy control; OB: obese; Offs: offspring; NMD: No Mood Disorder, MD: Mood Disorder; FE: first episode; 1 m: 1 month; 6 m: 6 months; 12 m: 12 months; Lit: lithium-treated. *Age*. *↓*: significantly reduced. *Treatment*. Val: valproic acid; P treated: means previously treated with more than 1 drug type or not specified; DN: drug-naïve; DF: drug-free; MS: mood stabilizers; WO: washed-out from treatment; y: years; m: months; d: days. *Techn*: technique used to measure peripheral levels of IGF. ELISA: enzyme-linked immunoassay; CLIA: chemiluminescence immunoassay; bb-Ls: bead-based Luminex system is a multiplexed sandwich immunoassay. *Country*. *^2^: Caucasians (*n* = 36) and Asian (*n* = 2). *Statistics*. ns: non-significant; *↑*: significantly elevated; *↓*: significantly reduced. *Superscripts* indicate the reference statistical comparison. N/A: data not found. ^ To improve visual clarity, the group corresponding to the IGFBP measurements is indicated.

## 7. IGF Peripheral Levels in Borderline Personality Disorder and Obsessive—Compulsive Disorder

### 7.1. IGF-1 in Borderline Personality Disorder (BPD)

Borderline personality disorder (BPD) represents a pattern of marked impulsivity and instability in interpersonal relationships, self-image, and affections. BPD prevalence is around 1.4%, but it seems to increase to 20% in patients with PDs and is more frequent in women [244]. Curiously, serum levels of IGF-1 were evaluated in the context of bone mineral loss in patients with a diagnosis of BPD with or without a concomitant diagnosis of MDD. BPD patients showed elevated IGF-1 levels in comparison to both controls and BDP + MDD patients, without reaching statistical significance [245]. Apparently, a significantly reduced IGF-1 level was found in a cluster of patients who did not respond adequately to a DEX test. However, this result had limitations, since only 5 of the 25 patients displayed BPD. Moreover, the statistical comparison was established between 25 cases and 223 controls [246].

### 7.2. IGF-1 in Obsessive–Compulsive Disorder (OCD)

Obsessive–compulsive disorder (OCD) consists of frequent intrusive thoughts and repetitive compulsive behaviors, which can be magnified during acute periods of stress. The estimated prevalence of OCD is around 1–3%. Generally, OCD can be separated into early onset with a mean age of 11 years and late onset at the age of 23 [247]. There are multiple potential factors contributing to OCD etiology. Some hypotheses suggest abnormalities in the cortical–striatum–thalamocortical circuit and the alteration of multiple neurotransmitters, such as serotonin, dopamine, and glutamate. The IGF family along with other neurotrophins (BDNF, NGF) may play a role in the pathophysiology of OCD [248]. In 2016, Rosso et al. measured serum levels of IGF-1 in OCD patients for the first time [124]. The theoretical basis for measuring IGF-1 in OCD was similar to MDD, since GH response to GHRH stimulation was found to be blunted in OCD [249]. Moreover, the nocturnal secretion of GH was altered in OCD patients. As with MDD patients, a potential alteration in the HPS axis in OCD patients was suggested [250]. Serum levels of IGF-1 were found to be significantly elevated in drug-naïve OCD patients at baseline when compared to controls. However, IGF-1 levels did not change in OCD patients after 10 weeks of treatment. Interestingly, no significant association was found between baseline levels of IGF-1 and the Yale–Brown Obsessive–Compulsive Scale (Y-BOCS) or age at onset. Curiously, no significant difference was found for serum levels of IGF-1 between OCD and MDD patients at baseline [124]. SSRIs are the gold standard for treating OCD patients. Nonetheless, 40–60% of OCD patients do not respond adequately to SSRI treatment. Interestingly, a significant positive association was found for baseline IGF-1 plasma levels and the percentage reduction of the Y-BOCS score after 3 months of SSRI treatment in drug-naïve OCD patients. Nevertheless, no significant correlation was found between plasma levels of IGF-1 with age, baseline Y-BOCS, or age at onset [251]. Recently, evidence from a rat model of OCD suggested a potential protective effect of melatonin by upregulating both IGF-1 and Glucagon-Like Peptide-1 (GLP-1) levels, which could be of help in the mitigation of behavioral and neurochemical alterations in OCD [252]. Table 4 offers a summary of the main articles studying IGF peripheral members in both Obsessive-Compulsive Disorder (OCD) and Borderline Personality Disorder (BPD).

**Table 4 ijms-26-02561-t004:** Data from 4 articles that have measured peripheral levels of the IGF family in Borderline Personality disorder (BPD) and Obsessive–Compulsive Disorder (1999–2024).

Ref.	DSM	Group	Sample Size	Age (Years)	Sex (F/M)	Treatment		Sample Source	Techn	Country	IGFs		Statistics	
[245]	IV	BDP	16	26.1 ± 5.1	All F	SSRIs (*n* = 3)		Fasting Serum	ELISA	German	IGF-1 (ng/mL)	189 ± 63	ns	N/A
		BPD + MDD 1	12	31.8 ± 6.5		SSRIs (*n* = 3)						161 ± 62		
		BPD + MDD 2	10	25.9 ± 5		SSRIs (*n* = 5)						164 ± 62		
		HC	20	24.2 ± 5.9								176 ± 32		
[246]	III-R	CB ADex	25	~60 *^1^	All M	Treated	Dex Test	Serum	RIA	Sweden	IGF-1 (ng/mL)	174.9 ± 33.8	↓ ^HC ADex^	*p* = 0.007
		CA ADex	42									196.4 ± 60.5	ns	*p* > 0.20
		HC ADex	223									212.8 ± 66.1		
		CB BDex	25			Treated	Dex Test					182.6 ± 44.0	ns	*p* = 0.062
		CA BDex	42									193.3 ± 67.9	↓ ^HC BDex^	*p* = 0.017
		HC BDex	223									210.3 ± 63.1		
[124]	IV	OCD	40	38.7 ± 13.3	22/18	Treated		Fasting Serum	ELISA	Italy	IGF-1 (ng/mL)	149.9 ± 60.2	↑ ^HC^	*p* = 0.04
		HC	43	42.3 ± 11.3	15/28							121.2 ± 51.6		
		OCD0	18	N/A	N/A							131.4 ± 50.3	ns	*p* = 0.215
		OCD1				SSRIs	1 w					126.6 ± 50.0		
[251]	IV	OCD0	16	28.1 ± 6.2	11/5	DN		Fasting Plasma	ELISA	India	IGF-1 (ng/mL)	129.7 ± 62.1	Y-BOCs (% red) *^2^ (r = 0.6; *p* = 0.02)	
		OCD1				SSRIs	3 m					N/A		

**Explanation by column.** *DSM*: version of The Diagnostic and Statistical Manual of Mental Disorders (DSM) used to diagnose BPD or OCD. *Group*. BDP + MDD 1: BDP patients with lifetime MDD; BDP + MDD 2: BPD patients with current MDD; CB: cluster B of patients (5 BPD, 3 histrionic, 17 narcissistic); CA: cluster A of patients (10 avoidant, 5 dependent, 18 OCD, 9 passive-aggressive); ADex: adequate response to 0.5 mg dexamethasone test; BDex: Blunted response to 0.5 mg dexamethasone test; Dex: dexamethasone challenge; 0: measure at baseline; 1: measure after a given time; HC: healthy control. *Age*. *^1^: This article was published in 1999, and it is stated that participants were born in the first 6 months of 1944, whereas samples were collected around 1994, therefore, we deduced an average of 60 years old. *Sex*. F: Female; M: Male. *Treatment*. SSRIs: selective serotonin reuptake inhibitors; DN: drug-naïve; m: months; w: weeks. *Techn*. Technique used to measure peripheral levels of IGF: RIA: radioimmunoassay; ELISA: enzyme-linked immunoassay. *Statistics*. ns: non-significant; *↑*: significantly elevated; *↓*: significantly reduced. *Superscripts* indicate the reference statistical comparison. *^2^: Y-BOCS was measured at baseline (25.7 ± 3.5) and after 3 months (18.3 ± 9.2) of SSRI treatment. % red: Y-BOCS percentage reduction was found to be inversely and significantly correlated with IGF-1 baseline levels. r: Spearman’s coefficient. N/A: data not found in the article.

## 8. IGF Peripheral Levels in Autism Spectrum Disorder

### 8.1. Autism Spectrum Disorder

Autism spectrum disorder (ASD) is a neurodevelopmental disorder characterized by persistent deficits in communication, social interaction, reciprocity, non-verbal communicative behaviors, and the development, maintenance, and understanding of social relationships. In addition, the diagnosis of ASD also requires restricted and repetitive patterns of behaviors, activities, or interests [253]. The global prevalence of ASD is around 1–2% both in children and adults [254]. The behavioral patterns of ASD start during early childhood and symptoms usually become recognized in the second year of life, or even before, and it may persist throughout life, with only a reduced proportion of patients working independently [255]. There are different subjective scales that evaluate ASD symptoms, such as the Autism Treatment Evaluation Checklist (ATEC), the Autism Spectrum Rating Scale (ASRS), the Autism Behavior Checklist (ABC), and the Childhood Autism Rating Scale (CARS), among others [256]. The IGF family has been studied in ASD (Figure 6) for its role in CNS development in both neuroplasticity modulation and behavioral effects, however different perspectives have been explored, such as growth-related anomalies and the neurotrophic approach. Results from these studies can be seen in Table 5.

### 8.2. First Studies and IGF Cerebrospinal Fluid Levels of Members in ASD Children

The rationale behind the first study of IGF-1 in the cerebrospinal fluid (CSF) was to explore its role as a neurotrophic factor involved in brain development and survival of cerebellar neurons. The levels of IGF-1 in the CSF were found to be significantly lower in children with ASD compared to controls. CSF IGF-1 was found to be significantly and positively associated with age in the ASD group, but not in the control group. Conversely, no significant correlation was found between CSF IGF-1 and head circumference. However, Vanhala et al. mentioned that their control group was not entirely “healthy”, since it would not have been ethical to extract CSF from healthy children [257]. In 2005, Vargas et al. found significantly increased levels of IGFBP-1, IGFBP-3, and IGFBP-4 in the CSF of ASD patients from all ages in comparison to controls. However, only IGFBP-1 expression was found to be significantly increased in the anterior cingulate and the middle frontal gyrus but not in the cerebellum. IGF-1 was not measured in CSF, but was found to also be increased in the anterior cingulate gyrus [25], but not in the fusiform gyrus [258].

Later, IGF-1 was found to be significantly decreased in children with ASD when compared to age-matched controls, whereas IGF-2 was unaltered. However, when separating by age group, only ASD children under 5 years had significantly lower CSF IGF-1 than controls. Equally, IGF-1 was found to be significantly and positively correlated with age in both ASD children and controls, whereas IGF-2 only correlated in controls. This time, IGF-1 positively correlated with head circumference only in ASD children, but not IGF-2. Nonetheless, no significant difference in head circumference was found between ASD children and controls [259]. Therefore, Riikonen proposed that the peripheral administration of IGF-1 to increase IGF-1 centrally acting as a potential therapy for ASD children [260]. ASD children showed a significant increase in CSF IGF-1 levels after 6 months of fluoxetine monotherapy, regardless of clinical response [261].

### 8.3. Relation Between IGFs and Growth in ASD Children

In the early 2000s, there was an open debate about whether children with ASD showed increased height, weight, or head growth, specifically during early life. Therefore, Mills et al. studied plasma levels of several members of the HPS axis, such as IGF-1, IGF-2, IGFBP-3, GHBP, and DHEA, in male children with ASD. Regarding anthropometrics, ASD children showed significantly higher head circumference, weight, and BMI than controls, but not height. IGF-1, IGF-2, IGFBP-3, and GHBP were all significantly increased in ASD children compared to controls, after adjusting for age and BMI. Weight was found to be significantly and positively correlated with IGF-1, IGF-2, IGBFP-3, and GHBP in both groups, and the same was found for height, with the exception for IGF-2 in the control group. Head circumference was significantly and positively associated with IGF-1, only in the ASD group. In this sense, higher levels of IGFs in plasma could be explaining heavier weights and longer head circumferences in the ASD group compared to controls [262].

In the same year, reduced urinary levels of IGF-1 in ASD children were found. Curiously, the reduction in IGFBP-3 was on the verge of reaching statistical meaning (*p* = 0.05). Nonetheless, no significant association was found between urinary levels of both IGF-1 and IGFBP-3 with symptom severity (CARS), which could have been of importance to at least support a cause–effect hypothesis between IGFs and ASD. Moreover, lower urinary IGF-1 levels were not associated with growth alterations in ASD children, who had normal height, weight, and head circumference. Finally, Anlar et al. proposed that reduced IGF-1 and IGFBP-3 could be reflecting alterations in GH production as a consequence of the sleep disturbance normally found in ASD [263].

### 8.4. IGF-1 in ASD Children

#### 8.4.1. IGF-1 in Reduced Bone Mineral Density in ASD Children

Another study measured serum levels of IGF-1 in the context of reduced bone mineral density in ASD children. Indeed, Neumeyer et al. found that bone mineral density was reduced in the radius and tibia of ASD children, whereas serum bone markers, such as calcium, vitamin D, phosphorus, estradiol, and total testosterone, did not differ. Both ASD children and controls had similar height and BMI, but the ASD group had a significantly higher body fat percentage. IGF-1 serum values were not given but the z-scores were found to be significantly higher in the ASD group [264].

#### 8.4.2. IGF-1 and the Neurotrophic Approach in ASD Children

Şimşek et al. found significantly higher IGF-1 serum levels in children diagnosed with ASD than controls, whereas other markers (VEGF, hypoxia-inducible factor 1-alpha HIF-1α) were not altered. Moreover, they found a significant positive correlation between IGF-1 and age in the ASD group, but not with BMI or symptom severity (CARS and ABC). Noteworthily, almost half of the ASD children were following pharmacological treatment, which could explain the increase in peripheral IGF-1 levels. They suggested that the elevation in peripheral IGF-1 levels could be a mechanism to compensate for abnormalities in myelin and white matter in ASD children [265]. Robinson-Agramonte et al. studied serum neurotrophins, like BDNF and IGF-1, in the context of their mutual signaling through the PI3K–Akt–mTOR pathway (Figure 1B). Therefore, IGF-1, but not BDNF, was found to be higher in ASD children when compared to controls. IGF-1 was found to be positively associated with age, but only in the ASD group. Moreover, IGF-1 levels significantly and positively correlated with symptoms in one scale (CARS), but not in others (CGI, ABC, Autism Diagnostic Interview (ADI), and Autism Treatment Evaluation Checklist (ATEC)). Interestingly, IGF-1 serum levels did not differ between treated and untreated ASD children. Treatments included carbamazepine, risperidone, methylphenidate (MPH), haloperidol, or valproic acid [266]. In their prior study, serum IGF-1 was not altered after non-invasive brain stimulation therapy in ASD children [267]. Another study found a significant reduction in IGF-1 serum levels in ASD children in contrast to controls. IGF-1 gradually decrease according to different genotypes in the IGF-1 (rs12579108) promoter polymorphism [268].

### 8.5. IGF Peripheral Levels in Different Stages of ASD Severity

In 2022, one study found that ASD children had significantly lower IGF-1 than controls, but not IGFBP-3. It was interesting that IGF-1 and IGFBP-3 were lower in boys than girls in both groups. Nonetheless, the sample size was triple in boys than in girls. Surprisingly, only IGFBP-3 significantly and negatively correlated with symptom severity (CARS, ABC) after correcting by sex and age. Despite this, children with severe ASD had significantly lower IGF-1 and IGFBP-3. Regarding growth, ASD children had similar height, weight, and head circumference than controls. Finally, when comparing by age group, only children of around 2 years old showed significantly higher IGF-1 than controls, whereas in groups of 3, 4, 5, and 6–7 years were not altered [26].

**Table 5 ijms-26-02561-t005:** Data from 4 studies in CSF and 10 articles in peripheral levels of the IGF family in Autism Spectrum disorder (ASD) through years (2001–2024).

Ref.	DSM	Groups	Sample Size	Age (Years)	Sex (F/M)	Treatment		Sample Source	Techn	Country	IGFs		Statistics		IGFBPs		Statistics	
[257]	III	ASD	11	3.8 ± 1.1	4/7	No treatment		CSF	RIA	Finland	IGF-1 (µg/L)	0.34 ± 0.08	↓ *^2^	*p* = 0.03				
		C	11	3.8 ± 1.3	6/5							0.40 ± 0.15						
[25]	IV	ASD	6	5.5 ± 2.5	2/4	N/A		CSF	ELISA	USA	IGF-1 was not measured. Data was not given for CSF IGFBPs but fold changes and *p*-values.				IGFBP-1 *^4^	0.4	↓	*p* = 0.036
		C	9	33.9 ± 10.8	6/3										IGFBP-3 *^4^	26.3	↑	*p* < 0.001
															IGFBP-4 *^4^	13.3	↑	*p* = 0.003
[259]	III	ASD	25	5y5m 1y11m–15y10m	5/25	N/A		CSF	RIA	Finland	IGF-1 (µg/L)	0.41 ± 0.18	↓	*p* = 0.02				
		C	16	7y4m 1y11m–15y10m	8/8							0.58 ± 0.27						
											IGF-2 (µg/L)	19.1 ± 3.10	ns	*p* = 0.33				
												20.5 ± 4.40						
[261]	N/A	ASDR0	13	5–16	1/12	Fluox	2 m	CSF	N/A	Finland	IGF-1 (µg/L)	0.55 ± 0.16	ns	*p* = 0.069				
		ASDR1										0.69 ± 0.21						
		ASDPR0										0.51 ± 0.2	↑	*p* = 0.001				
		ASDPR1										0.67 ± 0.2						
[262]	IV	ASD	71	6.6 ± 1.5	All M	treated (*n* = 8)		Plasma	RIA	England	IGF-1 (ng/mL)	149.0 ± 58.3	↑	*p* < 0.0001	IGFBP-3 (mg/L)	2.5 ± 0.6	↑	*p* < 0.0001
		HC	59	6.5 ± 1.2								113.7 ± 45.0				2.1 ± 0.4		
											IGF-2 (ng/mL)	397.2 ± 99.5	↑	*p* < 0.0001				
												306.1 ± 76.0						
[263]	IV	ASD	34	3.1 ± 0.9	8/26	N/A		Urinary	ELISA	Turkey	IGF-1 (µg/d)	0.7 ± 0.08	↓	*p* = 0.03	IGFBP-3 (mg/d)	2.2 ± 0.32	ns/↓	*p* = 0.05
		HC	29	3.3 ± 1.2	4/25							1.5 ± 0.34				3.3 ± 0.37		
[264]	IV	ASD	20	13.6 ± 0.53	All M	N/A		Serum	CMS	USA	IGF-1 (Z-scores)	0.18 ± 0.13	↑	*p* < 0.001				
		HC	20	14.2 ± 0.56								−0.73 ± 0.16						
[265]	V	ASD	40	6.98 ± 2.58	3/37	treated (yes/no) (19/21)		Serum	ELISA	Turkey	IGF-1 (ng/mL)	250.1 ± 131.5	↑	*p* < 0.001				
		HC	40	7.79 ± 2.05	3/37							141.9 ± 62.36						
[266]	V	ASD0	16	9.38 ± 2.63	3/13	NIBS (*n* = 13)		Serum	ELISA	Cuba	IGF-1 (ng/mL)	168.00 ± 85.40	ns	*p* = 0.669				
		ASD1										162.00 ± 81.40						
[267]	V	ASD	22	9.45 ± 2.94	5/17	treated (*n* = 14)		Serum	ELISA	Cuba	IGF-1 (ng/mL)	153.09 ± 77.69	↑	*p* = 0.037				
		HC	29	8.68 ± 2.82	8/21							115.31 ± 50.50						
[268]	V	ASD	200	6.6 ± 4.1	N/A	N/A		Serum	ELISA	Iran	IGF-1 (ng/mL)	31.45 ± 9.84	↓	*p* = 0.001				
		HC	198	6.8 ± 3.2								54.62 ± 11.63						
[269]	V	ASD	150	4.17 ± 1.67	37/113	DN		Fasting Serum	CL	China	IGF-1 (ng/mL)	106 (80,139)	↓	*p* = 0.021	IGFBP-3 (µg/mL)	3.6 ± 0.9	ns	*p* = 0.108
		HC	165	4 ± 1.33	41/124							113 (90,152)				3.7 ± 0.8		
[26]	V	ASD *^1^	180	8 ± 3.8	34/146	N/A		Serum	ELISA	Iran	IGF-1 (ng/mL)	39.06 ± 14.76	↓ lev ^ ↓ HC	N/A	IGFBP-3 (ng/mL)	2867.33 ± 496.49	↓ lev ^ ↓ HC	N/A
		ASDmi	69	N/A	N/A							35.46 ± 14.92				2883.46 ± 472.37		
		ASDmo	58									26.46 ± 13.77				2214.86 ± 543.20		
		ASDs	53									45.2 ± 17.69				3185.73 ± 559.67		
		HC	118	7.3 ± 3.7	23/95						IGF-2 (ng/mL)	1490.6 ± 159.9			IGFBP-4 (ng/mL)	397.86 ± 91.64		
												1455.93 ± 142.56				387.8 ± 77.96		
												1380.13 ± 178.55				357.13 ± 75.05		
												1546.07 ± 166.88				409.6 ± 102.06		
											IGFBP-1 *^3^ (ng/mL)	9.8 ± 4.2			IGFBP-5 (ng/mL)	216.8 ± 61.36		
												9.4 ± 6.5				182.2 ± 45.18		
												9.1 ± 3.4				161.26 ± 45.60		
												11.8 ± 4.6				224.86 ± 70.99		
											IGFBP-2 *^3^ (ng/mL)	175.13 ± 47.40			IGFBP-6 (ng/mL)	189 ± 60		
												167.53 ± 51.27				176.93 ± 50.15		
												150.8 ± 40.28				165.06 ± 47.74		
												178.6 ± 60.43				200.53 ± 64.72		
[234] ^	V	ASD + BD	40	14.03 ± 2.97	N/A	treated		Fasting Serum	ELISA	Turkey	IGF-1 (ng/mL)	9.10 (4.51–77.13)	ns	*p* = 0.855				
		ASD	40	13.08 ± 3.06		treated (*n* = 34)						10.87 (3.96–58.83)						

**Explanation by column.** *DSM*: version of The Diagnostic and Statistical Manual of Mental Disorders (DSM) used to diagnose ASD. *Group*. C: control group; HC: healthy controls; R: responders; PR: poor responders; mi: mild level; mo: moderate level; s: severe level; 0: measure at baseline; 1: measure after a given time. *^1^: This group does not have IGF values, but demographic data was given for all ASD children, but not for each level of severity. Ref. [262] is not indexed in PubMed. ^: [234] can be also found in Table 3. *Age*. y: years; m: months. *Sex*. M: Male. *Treatment*. Fluox: fluoxetine; NIBS: non-invasive brain stimulation; DN: drug-naïve. *Techn*. Technique used to measure peripheral levels of IGF: RIA: radioimmunoassay; ELISA: enzyme-linked immunoassay; CMS: chromatography mass spectrum. *Statistics*. *^2^: mean ± SD were calculated according to given data, since 4 ASD children had <0.16 µg/L. ns: non-significant; *↑*: significantly elevated; *↓*: significantly reduced; ↓ lev: the article states that IGFs and IGFBPs are significantly reduced on each level of ASD severity. *^3^: For organizing purposes, IGFBP-1 and IGFBP-2 were included in the *IGFs* column. *^4^: this article does not give CSF IGFBPs values, but it gives fold-change and *p*-values of each one. N/A: data not found in the article.

In the same year, another study measured the whole IGF family in the serum of ASD children. Both IGF-1 and IGF-2 were found to be significantly reduced in children with ASD compared to controls, with a decreasing gradient observed from mild to moderate to severe ASD symptom severity. The exact same tendency was found for IGFBP-1 to IGFBP-6 levels. This article was published in the journal *Archives of Psychiatry and Psychotherapy*, which is not indexed in PubMed, and therefore escaped our initial systematic search [270].

### 8.6. Animal Models Supporting IGF-2 Research in ASD Children

Since 2006, several reviews have suggested that IGF-1 has a potential therapeutical role in ASD children [260,270,271,272,273,274,275,276,277,278,279,280,281,282,283,284], whereas IGF-2 has been less studied. Nevertheless, research in animal models clearly supports further exploration in humans. Recently, it was found that systemic administration of IGF-2 reverted standard repetitive behaviors, social interaction, and cognitive deficits in two mice models of ASD [285,286]. Surprisingly, the effects of IGF-2 were found to be mediated by IGF-2R, and not IGF-1R [285]. Moreover, it was suggested that the beneficial effect of IGF-2 administration could be extended to many forms of ASD, since other studies made in animal models of Angelman syndrome (AS), which is an ASD-like neurodevelopmental disorder, showed beneficial effects on cognitive and behavioral patterns [287]. However, there is still some controversy, since another study found no behavioral benefits after IGF-2 administration in ASD mice [288].

## 9. IGF Peripheral Levels in Attention Deficit/Hyperactive Disorder

### 9.1. Attention-Deficit/Hyperactive Disorder

Attention-deficit/hyperactivity disorder (ADHD) is a neurodevelopmental disorder that affects 5–7% of the global population, with a higher tendency in children and teenagers [289]. ADHD is thought to be the result of a number of different genetic, neurobiological, and environmental factors, and has an estimated hereditability of 80% [290]. ADHD diagnosis has suffered from several modifications through time, and nowadays it is mainly divided into three subtypes: predominantly inattentive, hyperactive–impulsive, or a mix of both. Interestingly, since the release of DSM-V, ADHD and ASD coexist, giving them high comorbidity [291]. Recently, ADHD diagnosis has suffered a dramatic increase, and genes that control the expression of neurotrophic factors, such as BDNF, have been suggested to confer certain vulnerability [292]. Therefore, the study of the IGF signaling system is of interest to obtain a broader panorama of growth factors in the context of ADHD. Results from these studies can be seen in Table 6.

### 9.2. IGF-1 in ADHD Patients

#### 9.2.1. IGF-1 in the Context of Potential Growth Restriction by Methylphenidate (MPH) in ADHD Children

The peripheral IGF system was initially explored in ADHD children in the context of evaluating the potential side effects of methylphenidate (MPH), the most employed drug for ADHD treatment. In the early 1980s, evidence suggested a decrease in growth parameters, such as height and weight, after MPH treatment in hyperactive children. Therefore, IGF-1 was explored as a peripheral effector of the HPS axis in longitudinal growth. The first study found that there was no alteration in serum levels of IGF-1 between when ADHD children were under MPH treatment. Nonetheless, four children had IGF-1 below range [293]. In the same line, Toren et al. found no significant alteration in serum levels of IGF-1, GH, and GHBP between MPH-treated, drug-naïve ADHD children, and controls. ADHD children were following treatment for 7.2 ± 7.4 months by the time samples were extracted. However, the authors discussed that higher MPH dosages could alter circulating members of the HPS axis, since they worked with relatively low doses (5–20 mg/day). Interestingly, they encourage the study of IGFBP-3 as a potential indicator of GH deficiency [294].

Another study measured serum levels of both IGF-1 and IGFBP-3 in drug-naïve ADHD children at baseline and after 4–16 months of MPH. Interestingly, Bereket et al. proposed that growth deficits could be either a side effect from MPH or an inherent condition related to ADHD. Baseline IGF-1 and IGFBP-3 serum levels were within a “considered normal range” for ADHD children, which suggested that growth retardation might not be an intrinsic trait of ADHD. Then, there was no significant alteration in both IGF-1 and IGFBP-3 in different measures taken at 4, 8, or 14 months after initiating MPH treatment. These results, along with no alteration in both weight and height, postulated that MPH was not behind growth restriction in ADHD children [295].

In 2020, Kim et al. found that there was no difference in IGF-1 levels between drug-naïve ADHD, MPH-treated ADHD, and controls in both boys and girls. Moreover, growth serum markers (Thyroxine (T_4_), calcium, phosphorus, Alkaline Phosphatase (ALP), and vitamin D) or metabolic parameters (total protein, albumin, cholesterol) were not significantly different either. Only MPH-treated ADHD boys showed a significant decreased in TSH levels, but within normal ranges, and therefore clinical relevance was discarded [296]. Curiously, a retrospective study found that children with ADHD who underwent a GH provocative test between 1998 and 2013 showed no statistical difference on IGFBP-3 between boys and girls, whereas IGF-1 was found to be significantly higher in girls. Nevertheless, differences in sample size were substantial [297]. The potential influence of pharmacological treatment and comparison with controls are shown in Figure 7.

#### 9.2.2. Potential Influence of Systemic Inflammation and IGF-1 Levels with ADHD Increased Risk

More than a decade later, one study investigated the potential influence of systemic inflammation during the first postnatal month in children who were born before the 28th week of gestation and the increased risk of being identified by teachers with ADHD characteristics. Importantly, ADHD symptoms were later assessed at age 10. In this context, it was found that the top quartile concentrations of neurotrophic factors (IGF-1 and BDNF) and inflammatory markers (tumor necrosis factor-α (TNF-α), interleukin-8 (IL-8), and interleukin-1β (IL-6)) were associated with an increased risk. Nonetheless, ADHD was assessed by parents or teachers by the Child Symptom Inventory-4: Parent Checklist (CSI-4), and apparently there was no official DSM diagnosis. Moreover, it seems that blood IGFBP-1 measured in the 1st, 7th, and 14th postnatal days was related to reduced ADHD symptoms reported by both parents and teachers [298].

#### 9.2.3. Influence of Atomoxetine and MPH in IGF-1/IGFBP-3 Levels in ADHD Children

Atomoxetine (ATO) is a selective norepinephrine reuptake inhibitor used to treat ADHD to improve attention and impulse control in children over six years old. After 12 weeks of treatment, ATO was found not to alter IGF-1 and IGFBP-3 serum levels [299]. Another study showed that IGF-1 serum levels significantly increased in non-treated and MPH-treated ADHD children after 12 months, but not in the case of ATO-treated ADHD children. Conversely, IGFBP-3 remain unchanged in both the untreated and the MPH-treated, but significantly increased in the ATO group [300].

#### 9.2.4. IGF-1 in Drug-Naïve ADHD Children and Urine Levels of Phthalates

Phthalates are considered potential endocrinal disruptors that could potentially increase ADHD symptoms in children. In 2023, Wang et al. found that IGF-1 was significantly reduced in drug-naïve ADHD children compared to controls, whereas IGFBP-3 was found to be unaltered. However, despite potential interference of phthalates in growth, no other hormones (T_4_, free T_4_, Triiodothyronine (T_3_), and TSH) were found to be significantly altered between groups. Nonetheless, ADHD children were found with significantly higher levels of Mono(2-ethylhexyl) Phthalate (MEHP), Mono-n-butyl Phthalate (MnBP), and Mono-benzyl Phthalate (MBzP), but not Mono-ethyl Phthalate. (MEP), Mono-methyl phthalate (MMP), or Bisphenol A (BPA) urine levels. Moreover, IGF-1 was significantly and negatively correlated with MEHP and ADHD measurements, such as clinical symptoms, auditory attention-impulsive, and visual attention–sluggishness. Curiously, the only significant correlation for IGFBP-3 was found to be positive with clinical symptoms [301]. Conversely, the next year, IGF-1 levels and other parameters (VEGF, HIF-1α) were found to be unaltered between drug-naïve ADHD children and controls. What is more, ADHD children showed an increased IGF-1 tendency [302].

#### 9.2.5. IGF-1 and Melatonin Treatment in ADHD Adult Patients with Delayed Sleep Phase Syndrome (DSPS)

IGF-1 levels were found to be unaltered in ADHD adults with delayed sleep phase syndrome (DSPS) who were treated for 3 weeks either with just melatonin or melatonin plus 30 min of bright light therapy (BLT) in comparison to a group receiving placebo. This was interesting since almost 80% of adults with a diagnosis of ADHD exhibit an altered biological rhythm [303].

### 9.3. Supporting IGF-2 and Other IGFBPs in the Context of ADHD

To the best of our knowledge, there is still no study evaluating IGF-2 peripheral levels in ADHD, either for children or adults. Nonetheless, recent research has shown potential interest on IGF-2 in the context of ADHD, since elevated *IGF-2* DNA methylation at birth predicted ADHD symptoms in kids from 7 to 13 years old [304]. Therefore, *IGF-2* methylation might be a potential mechanism mediating vulnerability to ADHD [305]. In summary, the study of both IGF-1 and IGFBP-3 in ADHD children has been more related to possible growth impairments as a result from pharmacological treatment. Therefore, it could be interesting to explore the IGF peripheral system from a different perspective by including other IGFBPs that still remain unexplored.

**Table 6 ijms-26-02561-t006:** Data from 10 articles in peripheral levels of the IGF family in Attention Deficit and Hyperactive Disorder (ADHD) through years (1982–2024).

Ref.	DSM	Group	Sample Size	Age (Years)	Sex (F/M)	Treatment		Sample Source	Techn	Country	IGFs		Statistics		IGFBPs		Statistics	
[293]	N/A	T ADHD	8	10.99 ± 1.69	2/6	MPH	15.4 ± 14.4 m	Fasting Serum	RIA	USA	IGF-1 (U/mL)	0.76 ± 0.3	ns	*p* = 0.28 ^†^				
		NT ADHD	9	11.14 ± 1.64	2/7	Off-MPH treatment						0.9 ± 0.24 *						
[294]	III-R	NT ADHD	21	10.1 ± 2.4	All M	No treated		Serum	RIA	USA	IGF-1 (nmol/L)	19.4 ± 11.5	ns	*p* > 0.05				
		T ADHD	21			MPH	7.2 ± 7.4 m					24.1 ± 9.5						
		HC	30	10.08 ± 2.06								21.7 ± 8.5						
[295]	IV	ADHD0	14	8.12 ± 1.8	4/10	DN		Serum	IRMA	Turkey	IGF-1 (ng/mL)	163 ± 107			IGFBP-3 (mg/L)	4.87 ± 0.8		
		ADHD4m				MPH	4 m					97 ± 32	↓ ^0^	*p* < 0.05		4.13 ± 0.9	↓ ^0^	*p* < 0.05
		ADHD8m				MPH	8 m					134 ± 45	ns	N/A		4.30 ± 0.5	ns	N/A
		ADHH14m				MPH	12 m					178 ± 92	ns	N/A		4.81 ± 0.6	ns	N/A
[296]	IV	NT ADHD	41	8.81 ± 1.52	All M	DN		Serum	CLIA	Korea	IGF-1 (ng/mL)	198.49 ± 105.42	ns	*p* = 0.75				
		T ADHD	31	9.48 ± 1.44		MPH	1.79 ± 1.11 y					213.72 ± 137.09						
		HC	79	9.20 ± 1.65								189.85 ± 92.97						
		NT ADHD	23	9.22 ± 1.51	All F	DN						316.25 ± 169.19	ns	*p* = 0.45				
		T ADHD	8	9.89 ± 1.45		MPH	1.79 ± 1.11 y					289.56 ± 174.99						
		HC	33	9.36 ± 1.44								257.22 ± 131.28						
[299]	V	ADHD0	149	8.9 ± 2.78	56/93	DN		Serum	CLIA	China	IGF-1 (ng/mL)	170.43 ± 37.27	ns	*p* = 0.28	IGFBP-3 (µg/mL)	4.22 ± 0.87	ns	*p* = 0.51
		ADHD1				ATO	12 w					209.71 ± 83.53				4.62 ± 0.98		
[300]	V	NT ADHD0	22	8.8 ± 1.9	6/16	DN		Serum	ELISA	Taiwan	IGF-1 (ng/mL)	134.61 ± 54.04	↑ ^0,MPH12m^	*p* < 0.05	IGFBP-3 (ng/mL)	2746.83 ± 702.44	ns	N/A
		NT ADHD12m										174.94 ± 63.54				3099.68 ± 663.76		
		ADHD0	39	9.0 ± 2.3	9/30	MPH	12 m *^2^					121.43 ± 56.01	↑ ^0^	*p* < 0.05		2511.93 ± 790.25	ns	N/A
		ADHD12m										161.59 ± 72.31				2653.39 ± 631.24		
		ADHD0	40	9.5 ± 2.5	7/33	O-MPH						138.89 ± 73.44	↑ ^0,NT12m^	*p* < 0.05		2377.14 ± 690.58	↑ ^0^	*p* < 0.05
		ADHD12m										189.11 ± 83.77				2643.85 ± 539.67		
		ADHD0	17	9.1 ± 2.3	3/14	ATO						144.11 ± 76.27	ns	N/A		2636.61 ± 444.64	↑ ^0,O-MPH^	*p* < 0.05
		ADHD12m										181.57 ± 86.14				2962.45 ± 391.60		
[301]	V	ADHD	144	8.9 ± 2.2	34/110	DN		Fasting Serum	ELISA	Taiwan	IGF-1 (ng/mL)	141.5 ± 69.5	↓	*p* = 0.003	IGFBP-3 (mg/L)	N/A	ns	*p* > 0.05
		HC	70	9.2 ± 2.2	24/46							182.7 ± 100.8				N/A		
[302]	V	ADHD	40	9.87 ± 1.32	16/24	DN		Fasting Serum	ELISA	Turkey	IGF-1 (ng/mL)	7.42 ± 4.79	ns	*p* = *0*.074				
		HC	40	9.94 ± 1.58	16/24							6.60 ± 6.45						
[303]	IV	ADHD + DSPS	12	29.75 ± 9.03	6/6	MEL	3 w	Fasting Serum	ECLIA	Holland	IGF-1 (nmol/L)	30.30 ± 7.69	ns^PLAC^	*p* = 0.938				
			13	28.77 ± 10.99	9/4	MEL + BLT						31.28 ± 9.54	ns^PLAC^	*p* = 0.595				
			12	31.25 ± 6.70	7/5	Plac						26.69 ± 8.54						
[297]	N/A	ADHD	51	11.91 ± 2.21	6	GH provocative test Treated		Serum	RIA	Australia	IGF-1 (ng/mL)	29.1 ± 11.8	↑	*p* = 0.018	IGFBP-3 (nmol/L)	140.75 ± 34.61	ns	*p* = *0*.298
				13 ± 2.37	45							20 ± 8.1				no		

**Explanation by column.** *DSM*: version of The Diagnostic and Statistical Manual of Mental Disorders (DSM) used to diagnose ADHD. *Group*. T: treated; NT: non-treated; HC: healthy controls; 0: measure at baseline; 1: measure after a given time; m: months; DSPS: delayed sleep phase syndrome. *Sex*. M: Male. *Treatment*. MPH: Methylphenidate; ATO: atomoxetine; O-MPH: osmotic-MPH; MEL: melatonin; MEL + BLT: melatonin + bright light therapy (30 min); PLAC: placebo; DN: drug-naïve; y: years; m: months; w: weeks; d: days. *Techn*. Technique used to measure peripheral levels of IGF: RIA: radioimmunoassay; IRMA: IRMA: immunoradiometric assay; ELISA: enzyme-linked immunoassay; CLIA: chemiluminescence immunoassay; ECLIA: electro-chemiluminescence immunoassay. *^2^: MPH (14 ± 6.7 mg/d), O-MPH (32 ± 9.6 mg/d), and ATO (29.2 ± 9.7 mg/d) for 12 months. *IGFs*: *: Mean ± SD was calculated according to given values in the paper, which were 0.86 ± 0.03 (SEM, not SD). *Statistics*. ^†^ We calculated the exact *p*-value according to given data in the paper. ns: non-significant; *↑*: significantly elevated; *↓*: significantly reduced. *Superscripts* indicate the reference statistical comparison. N/A: data not found in the article.

## 10. Heterogeneity and Potential Source of Bias Among Studies

The articles that have measured peripheral IGF levels in psychiatry are subject to a source of heterogeneity based on certain parameters, which can be considered a source of bias and/or variability. These parameters are summarized in Figure 8.

## 11. Limitations and Future Perspectives

Despite the fact of the above-reviewed bibliography, there are several limitations when considering the IGF family as a potential peripheral biomarker for psychiatric disorders (PDs). First, while the *counter-regulatory mechanism* hypothesis might suggest a correlation between peripheral and central IGF-1 levels, it remains unclear whether peripheral changes reliably reflect central alterations and if these central alterations are a consequence of or a predisposition cause to PDs. In this regard, rodent models could help elucidate this relationship, but they present a second limitation: psychiatric disorders in humans are highly complex, and rodent models may not fully capture their pathophysiology. Second, species-specific differences in IGF-2 expression between humans and rodents could further hinder translational research. Third, human studies on the peripheral IGF system in PDs are often constrained by numerous confounding factors, including pharmacological treatments, diet, BMI, cognitive status, age, and ethnicity.

Addressing these challenges requires the implementation of robust statistical models that account for potential biases to develop a more patient-specific approach. Nevertheless, expanding research in this area is crucial, as IGF-1 remains one of the least studied neurotrophic factors in the context of PDs. Additionally, studies conducted in real-world clinical settings are essential to enhance the ecological validity of findings. If neurotrophic factors such as IGF-1 or BDNF are to serve as reliable biomarkers for PDs, they must be investigated within broader, everyday clinical paradigms.

## 12. Conclusions

This review explored the role of the IGF family as potential peripheral biomarkers in psychiatric disorders (PDs). In schizophrenia (SZ) patients, the initial hypothesis stipulated a deficiency in IGF levels as a consequence of neurodevelopmental disruptions that would confer vulnerability. Therefore, controversial results have been found with IGF-1 being either significantly increased, non-significant or reduced in SZ patients compared to controls. Nonetheless, antipsychotics (Aps) appear to upregulate IGF-2 and IGFBP-7. In major depressive disorder (MDD)patients, findings regarding IGF-1 are equally contradictory. Elevated IGF-1 levels were observed in drug-naïve FE MDD patients, suggesting a compensatory mechanism for impaired neurogenesis. Conversely, treated patients exhibit reduced IGF-1 and IGF-2. Alternatively, IGFBP-7 is uniquely increased in MDD, distinguishing it from other IGFBPs. However, their diagnostic value remains unclear. In bipolar disorder (BD) patients, evidence suggests that IGF alterations could be both a general trait marker or a mood state dependent marker, and longitudinal studies are needed to establish causality. Research on IGF levels in obsessive compulsive disorder (OCD) is more limited, but preliminary findings suggest altered IGF-1 reflecting underlying neuroplasticity deficits. IGF-1 could help identifying subtypes of OCD or predict responses to behavioral and pharmacological treatment, but more studies are needed to establish their significance. In all of these PDs, there are several important factors such as the state of the glucose–insulin metabolism, cortisol, cognitive alterations, age, regime and type of treatment, and stage of the disease that could be altering IGF peripheral measures. On the other hand, emerging evidence links IGF-1 with altered neurodevelopmental processes, such as autism spectrum disorder (ASD) and attention deficit hyperactive disorder (ADHD). Briefly, IGF-1 administration may improve synaptic function and social behaviors in ASD children. Additionally, the role of IGF-2 in memory consolidation highlights its relevance in ASD research. For ADHD, IGF-1 has been implicated in cognitive processes and executive function. Lower IGF-1 levels observed in some cohorts may be associated with attention and memory deficits.

The accessibility in peripheral tissues and the ability to cross the blood–brain barrier (BBB) makes the IGF family a promising candidate for biomarker research in PDs. Future research should prioritize standardized protocols, longitudinal designs, and integration of clinical and molecular data to validate the IGF family members as reliable biomarkers for PDs. In conclusion, while the IGF family shows potential as a biomarker for PDs, its clinical utility is still unknown. Addressing methodological limitations and incorporating larger, more diverse cohorts will be critical to advance in this field.

## Figures and Tables

**Figure 1 ijms-26-02561-f001:**
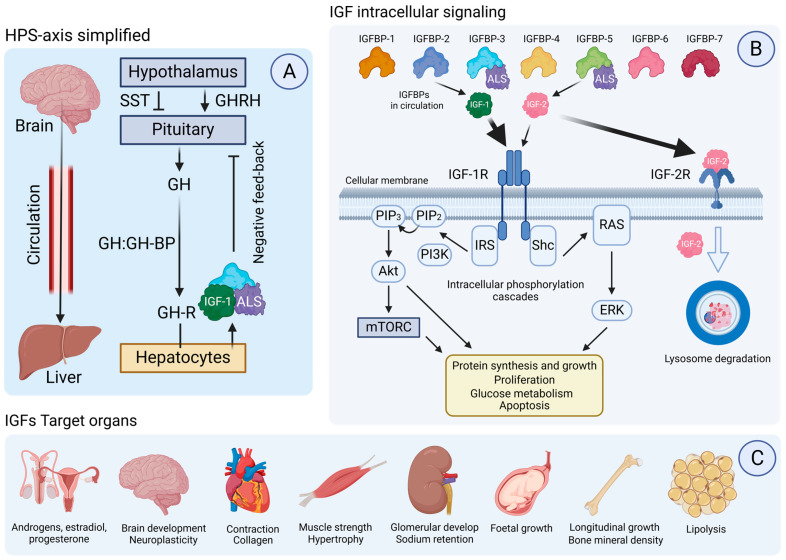
Brief schematic representation of (**A**) the Hypothalamic–Pituitary–Somatotropic (HPS) axis, (**B**) the IGF intracellular signaling, and (**C**) IGFs target organs. SST: somatostatin; GHRH: growth hormone releasing hormone; GH: growth hormone; GH-BP: growth hormone binding protein; GH-R: growth hormone receptor; ALS: acid labile subunit; PIP_2_: phosphatidylinositol 4,5-bisphosphate; PIP_3_: phosphatidylinositol (3,4,5)-trisphosphate; PI3K: phosphoinositide 3-kinase; IRS: insulin receptor substrate; Shc: Shc signaling adaptor protein; RAS: rat sarcoma; ERK: extracellular signal-regulated kinase; Akt: protein kinase B; mTORC: mammalian target of rapamycin; IGFBP: insulin-like growth factor binding protein; IGF: insulin-like growth; IGF-1R: insulin-like growth 1 receptor; IGF-2R: insulin-like growth 2 receptor.

**Figure 2 ijms-26-02561-f002:**
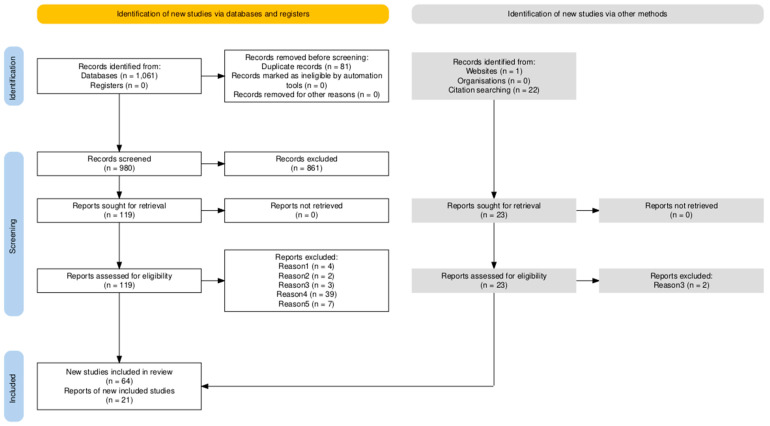
PRISMA flow diagram illustrating the study selection process for this review. The diagram shows the records identified from databases and other sources, the number of records excluded at each stage (duplicates, exclusion criteria, etc.), and the final number of studies included in the review [27].

**Figure 3 ijms-26-02561-f003:**
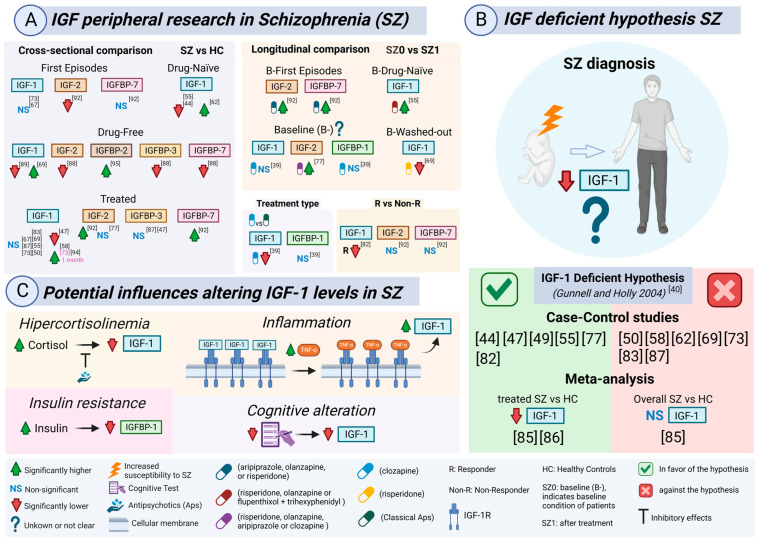
Summary of the main research on peripheral IGF levels in schizophrenia. (**A**) Shows IGF peripheral research for SZ in studies comparing SZ patients versus healthy controls and comparisons before/after pharmacological treatment. (**B**) The IGF deficient hypothesis states that reduced IGF-1 levels could be expected in SZ patients; therefore, we briefly resume case-control studies that are in favor of this hypothesis and others which are against. (**C**) Potential influences and hypothesis behind alterations in the IGF system in SZ patients. References in the figure are indicated in brackets.

**Figure 5 ijms-26-02561-f005:**
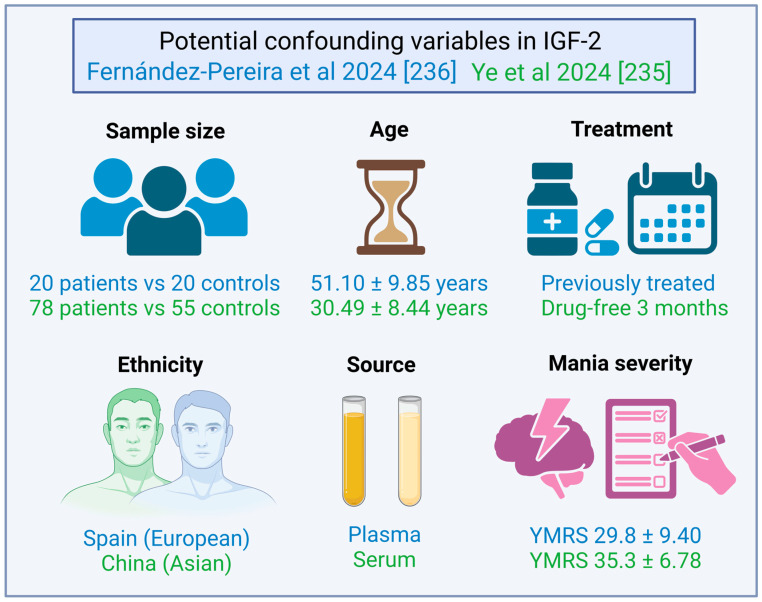
Potential confounding variables in IGF-2 studies [235,236]. Sample size, age, pharmacological treatment, ethnicity, sample source, or mania severity were different between both studies. YMRS: Young Mania Rating Scale.

**Figure 6 ijms-26-02561-f006:**
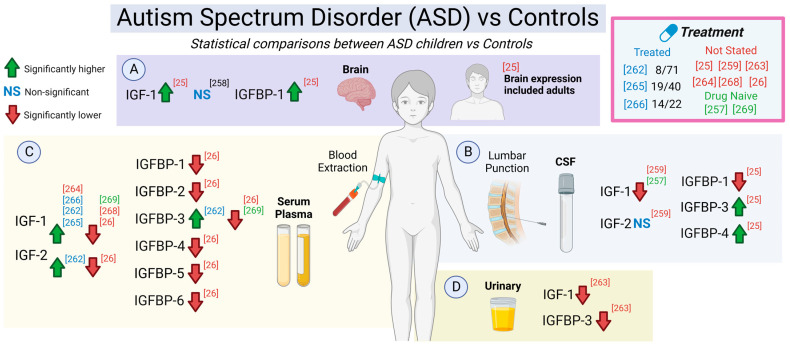
Summary of the studies measuring members of the IGF family in Autism Spectrum Disorder (ASD) children in comparison to control subjects in (**A**) brain tissue, (**B**) cerebrospinal fluid (CSF), and peripherally in (**C**) serum/plasma and (**D**) urinary levels. Controls from CSF studies were not entirely “healthy” but were considered as controls for ASD children.

**Figure 7 ijms-26-02561-f007:**
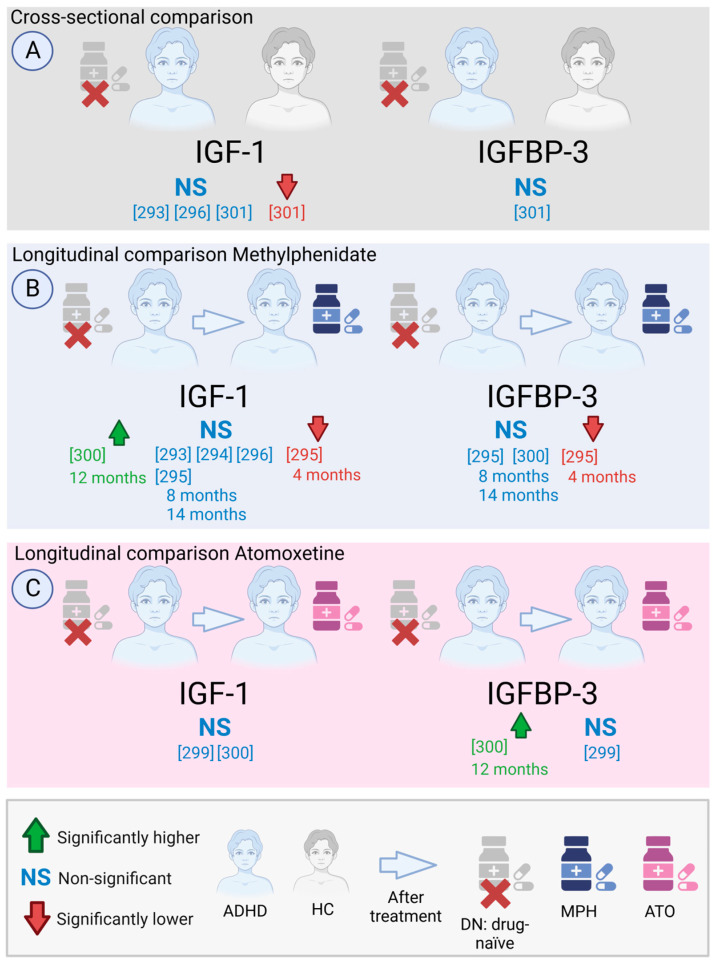
IGF-1 and IGFBP-3 peripheral levels in (**A**) cross-sectional comparison between drug-naïve (DN) ADHD children and controls. (**B**) Longitudinal comparisons in ADHD children treated with methylphenidate (MPH) and (**C**) atomoxetine (ATO).

**Figure 8 ijms-26-02561-f008:**
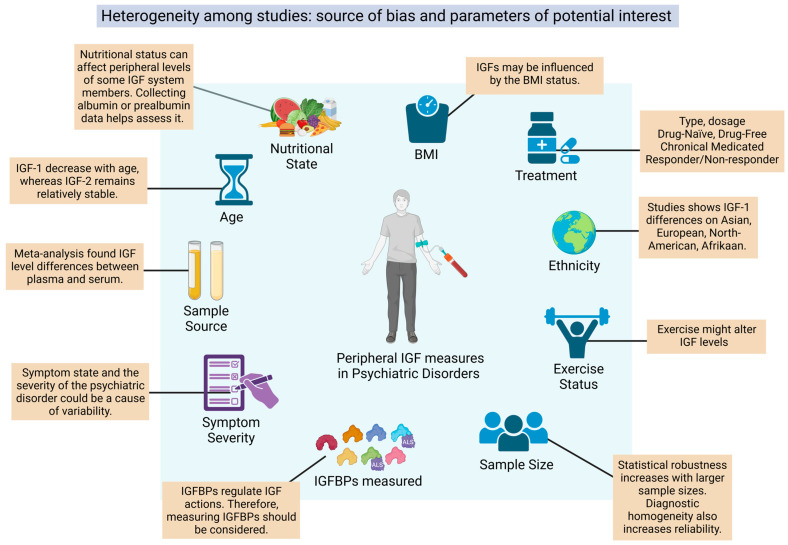
Heterogeneity, source of bias and parameters of potential interest. Studies measuring IGF peripheral levels in psychiatric disorders might present some parameters that can be considered as potential source of bias and heterogeneity among studies. These include Nutritional State, BMI (Body Mass Index), Treatment, Ethnicity, Exercise status, Sample Size, IGFBPs (Insulin-like growth factor binding proteins) measured in the studies, Symptom Severity, Sample Source, and Age.

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
