# Peer review of "The Insulin-like Growth Factor Family as a Potential Peripheral Biomarker in Psychiatric Disorders: A Systematic Review"

_ijms, 2025, doi:10.3390/ijms26062561_

Round 1

Reviewer 1 Report

Comments and Suggestions for Authors

The manuscript reviewed the alterations of IGF-1, IGF-2, and IGF-binding proteins (IGFBPs) in psychiatric disorders. It offers abundant data, covering the alterations of IGF-1, IGF-2 and IGFBPs in schizophrenia, major depressive disorders, bipolar disorders, borderline personality disorders, obsessive compulsive disorders, autism spectrum disorders and attention-deficit/hyperactivity disorders, providing useful information to the related fields. Howeverthe manuscript mainly reviewed the alterations of IGF-1, IGF-2, and IGF-binding proteins (IGFBPs) in psychiatric disorders, but not about their potential as peripheral biomarkers, which is not consistent with the title and abstract. In additionthe authors should make efforts to improve the manuscript as follows.

1.      The illustrations in Figure 2 are vague, for example, what do "SZ vs HC" and "SZ0→SZ1" mean? What are the results of "IGF peripheral research in SZ" and what conclusions can be drawn from the researches? The authors need to amplify these issues.

2.      One serious deficiency in the manuscript is that most paragraphs are mainly a list of literature resultslacks rigorous logical analysis, inductive analysis and summary. It is suggested that the author should re-analyze and integrate literature data, rather than just list the literature research results.

3.      Some subsections, such as 5.2.6. IGF-1 and relaxin-3 in MDD, convey little and possibly unnecessary information for the topic of the article, needn’t to be separated into separate subsections.

4.      The theme of each paragraph is not very clear and many sentences are verbose, such as 5.2.17.IGF-1 in post-stroke depression, it is recommended to add a topic sentence at the beginning or end of each paragraph. many similar paragraphs need to be revised.

5.      It is suggested to make a summary at the end of the article.

Comments on the Quality of English Language

The theme of each paragraph is not very clear and many sentences are verbose, it is recommended to add a topic sentence at the beginning or end of each paragraph. many similar paragraphs need to be revised.

Author Response

Answer to Reviewer 1 Round 1

Dear Reviewer 1,

First of all, thank you very much for taking the time to review this manuscript and for your valuable comments to improve the presentation of our work.

We have updated the manuscript format from PDF (named “FernandezPereira_AgisBalboa_IGF peripheral system in PDs” to a Word document (named “FernandezPereira_AgisBalboa_IGF peripheral system in PDs_V2”, in which we indicate to the EDITOR the specific locations where we would like to insert the Tables. As a result, the line references in the PDF have changed compared to the new version. In any case, we have noted the lines in the previous PDF version and marked them accordingly in the Word document of the new version.

We hope this approach allows the EDITOR to insert the tables as considered appropriate, therefore improving their presentation.

Below, we proceed to address the comments.

Comments and Suggestions for Authors

The manuscript reviewed the alterations of IGF-1, IGF-2, and IGF-binding proteins (IGFBPs) in psychiatric disorders. It offers abundant data, covering the alterations of IGF-1, IGF-2 and IGFBPs in schizophrenia, major depressive disorders, bipolar disorders, borderline personality disorders, obsessive compulsive disorders, autism spectrum disorders and attention-deficit/hyperactivity disorders, providing useful information to the related fields. However,the manuscript mainly reviewed the alterations of IGF-1, IGF-2, and IGF-binding proteins (IGFBPs) in psychiatric disorders, but not about their potential as peripheral biomarkers, which is not consistent with the title and abstract. In addition, the authors should make efforts to improve the manuscript as follows.

  1. The illustrations in Figure 2 are vague, for example, what do "SZ vs HC" and "SZ0SZ1" mean? What are the results of "IGF peripheral research in SZ" and what conclusions can be drawn from the researches? The authors need to amplify these issues.

Thank you for your comment. We changed the subtitle within Figure3A and others, indicating the type of statistical comparison, such as: “Cross-sectional comparison SZ vs HC” and “Longitudinal comparison SZ0 vs SZ1”. Any case, the legend of the figure indicates the meaning of SZ0 and SZ1, but maybe this way is clearer.

Indeed, this is a highly complex topic. As illustrated in Figure 1, numerous studies have measured IGFs in SZ, incorporating various variables and perspectives. Rather than making definitive statements, we opted to present the findings from each study, allowing readers to draw their own conclusions. However, we acknowledge the importance of including a conclusion. Therefore, we have added a new section, “10. Conclusion” to address this need, between lines [1557-1580]

“10.      Conclusion

This review explored the role of the IGF family as potential peripheral biomarkers in PDs. In SZ, the initial hypothesis stipulated a deficiency in IGF levels as a consequence of neurodevelopmental disruptions that would confer vulnerability. Therefore, controversial results have been found with IGF-1 being either significantly increased, non-significant or reduced in SZ patients compared to controls. Nonetheless, Aps appear to upregulate IGF-2 and IGFBP-7. In MDD, Findings regarding IGF-1 are equally contradictory. Elevated IGF-1 levels were observed in drug-naïve FE MDD patients, suggesting a compensatory mechanism for impaired neurogenesis. Conversely, treated patients exhibit reduced IGF-1 and IGF-2. Alternatively, IGFBP-7 is uniquely increased in MDD, distinguishing it from other IGFBPs. However, their diagnostic value remains unclear. In BD, evidence suggests that IGF alterations could be both a general trait marker or a mood state dependent marker and longitudinal studies are needed to establish causality. Research on IGF levels in OCD is more limited, but preliminary findings suggest altered IGF-1 reflecting underlying neuroplasticity deficits. IGF-1 could help identifying subtypes of OCD or predict responses to behavioural and pharmacological treatment, but more studies are needed to establish their significance. In all of this PDs, there are several important factors such as the state of the glucose-insulin metabolism, cortisol, cognitive alterations, age, regime and type of treatment, stage of the disease that could be altering IGF peripheral measures. On the other hand, emerging evidence links IGF-1 with altered neurodevelopmental processes such as ASD and ADHD. Briefly, IGF-1 administration may improve synaptic function and social behaviours in ASD. Additionally, the role of IGF-2 in memory consolidation highlights its relevance in ASD research. In ADHD, IGF-1 has been implicated in cognitive processes and executive function. Lower IGF-1 levels observed in some cohorts may be associated with attention and memory deficits.

The accessibility in peripheral tissues and the ability to cross the blood-brain barrier makes the IGF family a promising candidate for biomarker research in PDs. Future research should prioritize standardized protocols, longitudinal designs, and integration of clinical and molecular data to validate the IGF family members as reliable biomarkers for PDs. In conclusion, while the IGF family shows potential as a biomarker for PDs, its clinical utility is still unknown. Addressing methodological limitations and incorporating larger, more diverse cohorts will be critical to advance in this field.”

  1. One serious deficiency in the manuscript is that most paragraphs are mainly a list of literature results,lacks rigorous logical analysis, inductive analysis and summary. It is suggested that the author should re-analyze and integrate literature data, rather than just list the literature research results.

Thank you very much for your comment. However, we must respectfully disagree on this one. The objective of our review is to provide a comprehensive overview of the peripheral IGF system in the mentioned psychiatric disorders (PDs). We analysed more than 70 specific articles with the goal of offering not just a literal summary, but also the rationale behind these studies.

We believe it is essential to consider aspects that are often overlooked in systematic reviews or meta-analyses. For instance, in the case of SZ, factors like the role of cortisol or inflammation are frequently addressed in original research but are either excluded or dismissed in meta-analyses.

Our intention is to provide readers with a thorough understanding of the IGF system in PDs, while also indirectly highlighting its potential utility as a biomarker.

  1. Some subsections, such as 5.2.6. IGF-1 and relaxin-3 in MDD, convey little and possibly unnecessary information for the topic of the article, needn’t to be separated into separate subsections.

For sure. We understand that dedicating an entire paragraph solely to relaxin-3 may seem uncommon. However, we believe this approach is justified for the following reason: relaxin-3 has not been revisited in connection with IGF-1 in the context of MDD. Therefore, by allocating a full paragraph with its heading, we aim to provide readers with potentially valuable insights and shows possible lines for future research.

For us, it is of utmost importance to guide readers effectively toward significant topics or areas of potential interest regarding the peripheral IGF system in PDs.

  1. The theme of each paragraph is not very clear and many sentences are verbose, such as 5.2.17.IGF-1 in post-stroke depression, it is recommended to add a topic sentence at the beginning or end of each paragraph. many similar paragraphs need to be revised.

You may refer to section 2.2.17. We changed the title to a more precise one “5.2.17. IGF-1 in post-stroke depression (PSD) compared to MDD” by adding a topic sentence at the beginning and end of the paragraph.

  1. It is suggested to make a summary at the end of the article.

 As we mentioned in point 1, we have added a conclusive section.

Comments on the Quality of English Language

The theme of each paragraph is not very clear and many sentences are verbose, it is recommended to add a topic sentence at the beginning or end of each paragraph. many similar paragraphs need to be revised.

Thank you very much for your comments.

We believe that the way we did it is correct. We agree on that we covered several psychiatric disorders and the understanding of the whole text requires different areas of expertise. We wrote our review the way we considered appropriate to guide researchers into the study of the peripheral IGF system in different PDs. We included 7 figures and 6 tables, which give a summary of each section to easily access data in case needed. For sure, we will review the manuscript in order to find grammar mistakes or if we can improve the way a sentence was written.

Sincerely yours,

Fernández-Pereira and Agís-Balboa, the authors.

Reviewer 2 Report

Comments and Suggestions for Authors

The manuscript entitled "The Insulin-like Growth Factor family as a potential peripheral biomarker in Psychiatric Disorders" is really impactful for scientific community, giving a deepened literature review on this issue. I would like to congratulate the Authors for their efforts in doing such work.

Only few comments are required to improve some parts of the manuscript:

- please, add the aims of this review;

- please, repeat the column header on pages 10-12. Furthermore, add the number of pages. The same is for pages 21-24, page 30, page 36, page 40;

- considering lines 932-935, explain better the confounders in such studies and what the predictive factors are IGF-2;

- please, add the discussion and the conclusion of this manuscript, focusing on possible biases. Finally, add the future implication and perspectives of such literature review.

Author Response

Answer to Reviewer 2 Round 1

Dear Reviewer 2,

First of all, thank you very much for taking the time to review this manuscript and for your valuable comments to improve the presentation of our work.

We have updated the manuscript format from PDF to a Word document, in which we indicate to the EDITOR the specific locations where we would like to insert the Tables. As a result, the line references in the PDF have changed compared to the new version. In any case, we have noted the lines in the previous PDF version and marked them accordingly in the Word document of the new version.

We hope this approach allows the EDITOR to insert the tables as considered appropriate, therefore improving their presentation.

Below, we proceed to address the comments.

Comments and Suggestions for Authors

“The manuscript entitled "The Insulin-like Growth Factor family as a potential peripheral biomarker in Psychiatric Disorders" is really impactful for scientific community, giving a deepened literature review on this issue. I would like to congratulate the Authors for their efforts in doing such work.

Only few comments are required to improve some parts of the manuscript:

- please, add the aims of this review;

Indeed, we forgot to clearly state an aim for our review. We have now included the formal aim that has been added between lines [53-55] as it follows:

“Therefore, the aim of our review is to synthesize all original research available on the peripheral IGF system in some psychiatric disorders (SZ, MDD, BD, BPD, OCD, ASD, and ADHD), with particular emphasis on the rationale behind these studies and the potential of the IGF system as a biomarker."

- please, repeat the column header on pages 10-12. Furthermore, add the number of pages. The same is for pages 21-24, page 30, page 36, page 40;

Indeed, this issue arose because we created separate PDFs for the tables and then mixed them into a single final PDF document. Unfortunately, this led to the impossibility of page numbering where the tables are located. To address this, we have contacted the EDITOR to request assistance in ensuring the page numbers in the new PDF are corrected and appropriately aligned.

-considering lines 932-935, explain better the confounders in such studies and what the predictive factors are IGF-2;

We did not want to include a whole new paragraph explaining potential confounders or predictive factors in depth because potential confounders are explained all over the SZ and MDD sections. That is the reason why we included Figure 4 (now Figure 5, after the addition of a new Figure 2), in order to briefly show the differences in these parameters (sample size, age, treatment, ethnicity, source of sample and severity of manic symptoms). However, we believe it might be interesting to add some changes in order to improve the understanding of the paragraph. For this purpose, the following paragraph has been added (lines 935-937).

“From our perspective, there are several factors such as sample size, age, pharmacological treatment, ethnicity, source of sample and severity of manic symptoms, which were different between studies [233, 234] and that might be behind the observed differences. Therefore, the differences in these factors between both works [233, 234] are shown in Figure 5.”

We hope that now the differences between our works have been better explained, if not please tell us what changes would you have implemented.

- please, add the discussion and the conclusion of this manuscript, focusing on possible biases. Finally, add the future implication and perspectives of such literature review.”

We did not want to include a discussion section itself, since we consider that somehow the whole text is a mix between a exposition and a discussion. However, in line with your thoughts, we believe it can be useful to add a conclusive section between lines [1550-1573] as it follows:

“10.      Conclusion

This review explored the role of the IGF family as potential peripheral biomarkers in PDs. In SZ, the initial hypothesis stipulated a deficiency in IGF levels as a consequence of neurodevelopmental disruptions that would confer vulnerability. Therefore, controversial results have been found with IGF-1 being either significantly increased, non-significant or reduced in SZ patients compared to controls. Nonetheless, Aps appear to upregulate IGF-2 and IGFBP-7. In MDD, Findings regarding IGF-1 are equally contradictory. Elevated IGF-1 levels were observed in drug-naïve FE MDD patients, suggesting a compensatory mechanism for impaired neurogenesis. Conversely, treated patients exhibit reduced IGF-1 and IGF-2. Alternatively, IGFBP-7 is uniquely increased in MDD, distinguishing it from other IGFBPs. However, their diagnostic value remains unclear. In BD, evidence suggests that IGF alterations could be both a general trait marker or a mood state dependent marker and longitudinal studies are needed to establish causality. Research on IGF levels in OCD is more limited, but preliminary findings suggest altered IGF-1 reflecting underlying neuroplasticity deficits. IGF-1 could help identifying subtypes of OCD or predict responses to behavioural and pharmacological treatment, but more studies are needed to establish their significance. In all of this PDs, there are several important factors such as the state of the glucose-insulin metabolism, cortisol, cognitive alterations, age, regime and type of treatment, stage of the disease that could be altering IGF peripheral measures. On the other hand, emerging evidence links IGF-1 with altered neurodevelopmental processes such as ASD and ADHD. Briefly, IGF-1 administration may improve synaptic function and social behaviours in ASD. Additionally, the role of IGF-2 in memory consolidation highlights its relevance in ASD research. In ADHD, IGF-1 has been implicated in cognitive processes and executive function. Lower IGF-1 levels observed in some cohorts may be associated with attention and memory deficits.

The accessibility in peripheral tissues and the ability to cross the blood-brain barrier makes the IGF family a promising candidate for biomarker research in PDs. Future research should prioritize standardized protocols, longitudinal designs, and integration of clinical and molecular data to validate the IGF family members as reliable biomarkers for PDs. In conclusion, while the IGF family shows potential as a biomarker for PDs, its clinical utility is still unknown. Addressing methodological limitations and incorporating larger, more diverse cohorts will be critical to advance in this field.”

Thank you very much again for reviewing our work and for the consideration of our efforts.

Please, tell us if you believe that something else can be improved or changed in the actual format.

Sincerely yours,

Fernández-Pereira and Agís-Balboa.

Reviewer 3 Report

Comments and Suggestions for Authors

Review "The Insulin-like Growth Factor family as a potential peripheral biomarker in Psychiatric Disorders" very comprehensively and systematically indicates the potential practical/clinical importance of peripheral biomarkers in psychiatric diseases. The review presents the structure of insulin-like growth factor families, as well as their physiological role and impact on target organs. The importance of IGF in schizophrenia is explained in detail; as well as major depressive disorder and other depressive disorders, including post-stroke depression and depression in neurodegenerative diseases; bipolar disorders; Borderline Personality Disorder and Obsessive Compulsive Disorder; Autism Spectrum Disorder; Attention Deficit/Hyperactive Disorder. Explanations also include particularly sensitive categories (comorbid conditions, women and children).

For each of the psychiatric disorders considered in detail, historical and current findings are given along with textual explanations and summarization of the presented facts through pictures and tables.

Well-defined subheadings make it easier to find the desired facts.

The advantage of such a detailed presentation for all processed psychiatric disorders is that in one manuscript you can find well-processed and clearly presented data.

The disadvantage of the manuscript is the existence of concise "takeaway messages", i.e. there is no conclusion at the end of the article (like the one at the end of the abstract, only more extensive) that would be of practical use to the reader in clinical practice, in addition to the undoubted theoretical benefit and expansion of knowledge that such an article provides.

Author Response

Answer to Reviewer 3 Round 1

Dear Reviewer 3,

First of all, thank you very much for taking the time to review this manuscript and for your valuable comments to improve the presentation of our work.

We have updated the manuscript format from PDF to a Word document, in which we indicate to the EDITOR the specific locations where we would like to insert the Tables. As a result, the line references in the PDF have changed compared to the new version. In any case, we have noted the lines in the previous PDF version and marked them accordingly in the Word document of the new version.

We hope this approach allows the EDITOR to insert the tables as considered appropriate, therefore improving their presentation.

Below, we proceed to address the comments.

Comments and Suggestions for Authors

Review "The Insulin-like Growth Factor family as a potential peripheral biomarker in Psychiatric Disorders" very comprehensively and systematically indicates the potential practical/clinical importance of peripheral biomarkers in psychiatric diseases. The review presents the structure of insulin-like growth factor families, as well as their physiological role and impact on target organs. The importance of IGF in schizophrenia is explained in detail; as well as major depressive disorder and other depressive disorders, including post-stroke depression and depression in neurodegenerative diseases; bipolar disorders; Borderline Personality Disorder and Obsessive Compulsive Disorder; Autism Spectrum Disorder; Attention Deficit/Hyperactive Disorder. Explanations also include particularly sensitive categories (comorbid conditions, women and children).

For each of the psychiatric disorders considered in detail, historical and current findings are given along with textual explanations and summarization of the presented facts through pictures and tables.

Well-defined subheadings make it easier to find the desired facts.

The advantage of such a detailed presentation for all processed psychiatric disorders is that in one manuscript you can find well-processed and clearly presented data.

The disadvantage of the manuscript is the existence of concise "takeaway messages", i.e. there is no conclusion at the end of the article (like the one at the end of the abstract, only more extensive) that would be of practical use to the reader in clinical practice, in addition to the undoubted theoretical benefit and expansion of knowledge that such an article provides.

Indeed, we completely forgot to add a final summary.

We now added a conclusion in section 7 between lines [1557-1580] as it follows:

“10.      Conclusion

This review explored the role of the IGF family as potential peripheral biomarkers in PDs. In SZ, the initial hypothesis stipulated a deficiency in IGF levels as a consequence of neurodevelopmental disruptions that would confer vulnerability. Therefore, controversial results have been found with IGF-1 being either significantly increased, non-significant or reduced in SZ patients compared to controls. Nonetheless, Aps appear to upregulate IGF-2 and IGFBP-7. In MDD, Findings regarding IGF-1 are equally contradictory. Elevated IGF-1 levels were observed in drug-naïve FE MDD patients, suggesting a compensatory mechanism for impaired neurogenesis. Conversely, treated patients exhibit reduced IGF-1 and IGF-2. Alternatively, IGFBP-7 is uniquely increased in MDD, distinguishing it from other IGFBPs. However, their diagnostic value remains unclear. In BD, evidence suggests that IGF alterations could be both a general trait marker or a mood state dependent marker and longitudinal studies are needed to establish causality. Research on IGF levels in OCD is more limited, but preliminary findings suggest altered IGF-1 reflecting underlying neuroplasticity deficits. IGF-1 could help identifying subtypes of OCD or predict responses to behavioural and pharmacological treatment, but more studies are needed to establish their significance. In all of this PDs, there are several important factors such as the state of the glucose-insulin metabolism, cortisol, cognitive alterations, age, regime and type of treatment, stage of the disease that could be altering IGF peripheral measures. On the other hand, emerging evidence links IGF-1 with altered neurodevelopmental processes such as ASD and ADHD. Briefly, IGF-1 administration may improve synaptic function and social behaviours in ASD. Additionally, the role of IGF-2 in memory consolidation highlights its relevance in ASD research. In ADHD, IGF-1 has been implicated in cognitive processes and executive function. Lower IGF-1 levels observed in some cohorts may be associated with attention and memory deficits.

The accessibility in peripheral tissues and the ability to cross the blood-brain barrier makes the IGF family a promising candidate for biomarker research in PDs. Future research should prioritize standardized protocols, longitudinal designs, and integration of clinical and molecular data to validate the IGF family members as reliable biomarkers for PDs. In conclusion, while the IGF family shows potential as a biomarker for PDs, its clinical utility is still unkown. Addressing methodological limitations and incorporating larger, more diverse cohorts will be critical to advance in this field.”

Thank you very much again for reviewing our work and for the consideration of our efforts.

Please, tell us if you believe that something else can be improved or changed in the actual format.

Sincerely yours,

Fernández-Pereira and Agís-Balboa.

Reviewer 4 Report

Comments and Suggestions for Authors

Thank you very much for the opportunity to read your text. Since it is such a specific topic, I will focus primarily on methodological aspects that are of concern to me:

1. The description of the search strategy in databases (PubMed) is adequate, but it would be useful to include a table with the key terms used and how they were combined. This would improve the transparency and replicability of the study.

2. Although a systematic search is mentioned, it appears that additional studies outside those initially found were included. It is important to clarify how these additional studies were selected and whether quality assessment was performed for them.

3. No explicit inclusion/exclusion criteria are detailed for the studies considered in the review. This is crucial to justify why certain studies were included or excluded. Why didn't they choose SCOPUS? I consider it to be a more comprehensive database. Please justify your selection.

4. Heterogeneity among the reviewed studies is mentioned, but the sources of this heterogeneity (e.g., differences in measurement techniques, populations, or treatments) are not explored in depth. A summary table with key characteristics of the included studies could help to identify these differences.

5. It is unclear whether a systematic approach was applied to synthesize the results beyond describing them. The inclusion of flowcharts (such as a PRISMA) or statistical analyses (if possible) would have strengthened the results section.

6. Although some limitations are mentioned (e.g., lack of controls or longitudinal measures in some studies), a more structured discussion of the overall methodological limitations of the included studies and of the review itself would be beneficial

7. The acronyms are well defined at the beginning, but are used extensively in the text, which can make reading difficult. Consideration could be given to repeating the full term occasionally or in key sections. There are minor grammar and writing errors that could be corrected to improve the flow of the text. 

Author Response

Answer to Reviewer 4

Dear Reviewer 4,

First of all, thank you very much for taking the time to review this manuscript and for your valuable comments to improve the presentation of our work.

We have updated the manuscript format from PDF to a Word document, in which we indicate to the EDITOR the specific locations where we would like to insert the Tables. As a result, the line references in the PDF have changed compared to the new version. In any case, we have noted the lines in the previous PDF version and marked them accordingly in the Word document of the new version.

We hope this approach allows the EDITOR to insert the tables as considered appropriate, therefore improving their presentation.

Below, we proceed to address the comments.

Comments and Suggestions for Authors

Thank you very much for the opportunity to read your text. Since it is such a specific topic, I will focus primarily on methodological aspects that are of concern to me:

  1. The description of the search strategy in databases (PubMed) is adequate, but it would be useful to include a table with the key terms used and how they were combined. This would improve the transparency and replicability of the study.

We believe that the information you requested was already included in Supplementary Figure 1. However, we think it would be more appropriate to move Figure S1 into the main text as Figure 2. Actually, moved by your indications, we have changed the whole explanation in section 3 Results [lines 144-167].

  1. Although a systematic search is mentioned, it appears that additional studies outside those initially found were included. It is important to clarify how these additional studies were selected and whether quality assessment was performed for them.

Nonetheless, we believe that this point could be partially answered in section C of Figure 1 “C) Number of articles that were added after an extensive reading that were not initially found on systematic search PubMed search”.

Therefore, we included extra articles because two main reasons:

  1. They were indirectly found, which means that we found those articles because they were cited in papers that were found in our initial systematic search. However, these extra articles were not directly found as part of our initial systematic search.
  2. They were found as a consequence of searching for all PDs and IGF, as it is the case of reference [270], which was the only one and that is not even indexed in PubMed, as it is indicated in line 1097.

However, moved by your requests, we have included 6 Supplementary Tables in which we included each article considered showing the IGF measured and therefore the number of total studies included for each PD. This is indicated in the main text between lines [164-167].

  1. No explicit inclusion/exclusion criteria are detailed for the studies considered in the review. This is crucial to justify why certain studies were included or excluded. Why didn't they choose SCOPUS? I consider it to be a more comprehensive database. Please justify your selection.

We both screened the articles in a double-blinded manner, principally, the initial screening. This was in order to include/exclude the articles. We have slightly changed the definition of some article types in order to improve the explanation of our search. This can be now seen in Figure 2, between lines [150-163].

In Figure 1A, we indicate the number of references found for each article type. Nonetheless, we agree with you that we did not establish specific inclusion/exclusion criteria beyond ‘measuring IGF peripherally in PDs’ for the selected papers, as we did not conduct a proper systematic review in terms of publication. Therefore, we believe this process to be ultimately unnecessary.

PubMed is one of the most specialised biomedical databases that gives free access to most papers, whereas SCOPUS is of private use. When we thought which database to use, we did not ponder an alternative because for us is just appropriate by definition, as a premise. For sure, SCOPUS has some advantages since it covers more areas and may give bibliometric data. However, these aspects might not be interesting for us in the context of our review. We truly appreciate your suggestion, but we did it this way and there is no way to consider PubMed as a “non-reliable” database.

  1. Heterogeneity among the reviewed studies is mentioned, but the sources of this heterogeneity (e.g., differences in measurement techniques, populations, or treatments) are not explored in depth. A summary table with key characteristics of the included studies could help to identify these differences.

Unfortunately, we did not add a proper section for this. However, the reason behind was double. First, to avoid redundancy. Data from measurement techniques, sample size, treatments, country, and so on can be also checked on each Table, separately for each psychiatric disorder. Second, the number of words and the space needed to add a table like that, having in mind we have more than 50 references just between SZ and MDD, would just take too much for something that can be easily checked on each table (DSM version, group, sample size, age, gender, treatment, sample source, technique, country, IGF ligands, statistics, IGFBPs and statistics). We believe data given by this tables should be more than enough.

  1. It is unclear whether a systematic approach was applied to synthesize the results beyond describing them. The inclusion of flowcharts (such as a PRISMA) or statistical analyses (if possible) would have strengthened the results section.

The resume from our initial search in systematic terms can be checked in Figure 2. We did not want to make a systematic approach because we considered that almost 20 papers (shown in Figure 1C) would have been excluded if a systematic search was followed (shown in Figure 1B).

For sure, a meta-analysis is always an interesting approach. It was not our goal when doing this review. We appreciate your suggestion but the main purpose of our review was to give a rationale for each block of articles, as well as to show whether can be potentially used as future biomarkers in this field.

  1. Although some limitations are mentioned (e.g., lack of controls or longitudinal measures in some studies), a more structured discussion of the overall methodological limitations of the included studies and of the review itself would be beneficial

For sure. We completely agree with you on this one. We did not want to include a whole paragraph discussing this issue because we assume that somehow the whole text is partially a discussion in which heterogeneity can be interpreted.

  1. The acronyms are well defined at the beginning, but are used extensively in the text, which can make reading difficult. Consideration could be given to repeating the full term occasionally or in key sections. There are minor grammar and writing errors that could be corrected to improve the flow of the text.

Indeed. We agree on that there is an overuse of the acronyms. The reason behind is to reduce the total amount of words, since there is a considerable difference if we do not use acronyms whenever possible. This was actually measured.

We would revise the whole text in order to change acronyms for the full term at least in key sections as you mention. We will revise the whole text to check for grammar errors. Anyway, if you have detected some important mistakes, please let us know.

Thank you very much for your comments and suggestions to improve our work,

Sincerely yours,

Fernández-Pereira and Agis-Balboa.

Round 2

Reviewer 2 Report

Comments and Suggestions for Authors

I congratulate the Authors for the improvement of their manuscript. The manuscript can be published.

Author Response

Thanks to reviewer 2 for his comments that have helped improve the manuscript.

Reviewer 4 Report

Comments and Suggestions for Authors

Thank you very much, dear authors, for your attention to the comments. I would like to share with you some additional points that we need to be addressed.

1. Although they explain that the additional studies were found through citations in other articles, the quality assessment of these studies is lacking.

2. The inclusion/exclusion criteria are not detailed.

3. Although they point out that the heterogeneity data are already in the tables by disorder, a summary table would be ideal to better identify them.

4. I insist that the use of a PRISMA flow chart would help to provide methodological clarity.

5. Please, we need the limitations explicitly pointed out.

We remain attentive to the new version of your text.

Author Response

Reviewer 4 Round 2

Comments and Suggestions for Authors

Thank you very much, dear authors, for your attention to the comments. I would like to share with you some additional points that we need to be addressed.

  1. Although they explain that the additional studies were found through citations in other articles, the quality assessment of these studies is lacking.

These are articles indexed in PubMed, excluding [270], as previously indicated. We are unsure what you mean by "quality assessment." We searched for the relevant terms in PubMed, copied the results in CSV format, and analyzed the abstracts using Abstrackr to classify them into the different article types mentioned. Once the classification was completed, we proceeded to read the articles of interest and found that some studies had also measured peripheral IGF levels in psychiatric disorders but did not appear under the initial search terms. Therefore, we specifically searched for those articles in PubMed.

  1. The inclusion/exclusion criteria are not detailed.

The inclusion and exclusion criteria are presented in Figure 1. We would like to clarify the meaning of “not detailed”. Could you specify what additional information you require beyond what is shown in Figure 1?

The core of our manuscript is based on studies that measured peripheral levels of IGF family members in patients diagnosed with one of the mentioned psychiatric disorders (SZ, MDD, BD, OCD, BPD, ASD, and ADHD). However, we found this approach to be somewhat reductionist. While it may be the correct methodology for a systematic review, we believe it was not the most appropriate choice for our study. As previously mentioned, several relevant articles were not captured in our initial systematic search.

Furthermore, some studies assessed peripheral IGF system components—primarily IGF-1—in large cohorts of participants who did not have a formal MDD diagnosis but exhibited depressive symptoms. We considered these population-based studies to be valuable and worth including in our manuscript, as they explore the relationship between depression and IGF-1 from a different perspective. If we had strictly adhered to a rigid inclusion criterion that required measures on IGF peripheral levels exclusively in diagnosed MDD cases, the incorporation of these studies would not have been possible.

This, along with other previously stated reasons, led us to structure our work as a non-systematic review. Nonetheless, we chose to include our initial systematic search. We believe this information may be useful for readers interested in understanding how many PubMed entries are available when searching for each psychiatric disorder (SZ, MDD, BD, OCD, BPD, ASD, and ADHD) in combination with IGF (as it is stated in Figure 1).

  1. Although they point out that the heterogeneity data are already in the tables by disorder, a summary table would be ideal to better identify them.

We added a new paragraph between lines [1559-1662]:

  1. Heterogeneity and potential source of bias among studies

The articles that have measured peripheral IGF levels in psychiatry are subject to a source of heterogeneity based on certain parameters, which can be considered a source of bias and/or variability. These parameters are summarized in Figure 8.

(We also added the new Figure 8)

  1. I insist that the use of a PRISMA flow chart would help to provide methodological clarity

Certainly, the PRISMA methodology is highly useful. However, our review does not follow systematic guidelines. That is the reason why we have included relevant articles that did not appear in the initial systematic search. Nonetheless, Figure 1 shows the classification of articles and in Supplementary Tables we included the core articles of the review (as it is indicated, the ones that measured IGF peripherally and meta-analysis).

  1. Please, we need the limitations explicitly pointed out.

We do not fully understand what you mean by limitations. In an original article with experiments and original data, we understand that there might be some limitations with the research experimental design. However, in this case, since it is a (non-systematic) review, we are not quite sure what could be considered a limitation that is not already described throughout the text in a broader manner. However, we added a new paragraph between lines [1568-1582].

  1. Limitations and future perspectives

Despite the fact of the above-reviewed bibliography, there are several limitations when considering the IGF family as a potential peripheral biomarker for PDs. First, while the counter-regulatory mechanism hypothesis might suggest a correlation between peripheral and central IGF-1 levels, it remains unclear whether peripheral changes reliably reflect central alterations and if these central alterations are a consequence of or a predisposition cause to PDs. In this regard, rodent models could help elucidate this relationship, but they present a second limitation: psychiatric disorders in humans are highly complex, and rodent models may not fully capture their pathophysiology. Third, species-specific differences in IGF-2 expression between humans and rodents could further hinder translational research. Fourth, human studies on the peripheral IGF system in PDs are often constrained by numerous confounding factors, including pharmacological treatments, diet, BMI, cognitive status, age, and ethnicity. Addressing these challenges requires the implementation of robust statistical models that account for potential biases to develop a more patient-specific approach. Nevertheless, expanding research in this area is crucial, as IGF-1 remains one of the least studied neurotrophic factors in the context of PDs. Additionally, studies conducted in real-world clinical settings are essential to enhance the ecological validity of findings. If neurotrophic factors such as IGF-1 or BDNF are to serve as reliable biomarkers for PDs, they must be investigated within broader, everyday clinical paradigms.”

We remain attentive to the new version of your text.

Thank you very much for your comments,

We hope that now your request has been solved.

Round 3

Reviewer 4 Report

Comments and Suggestions for Authors

Many thanks to the authors for their work.

My biggest concern remains the lack of transparency in the process of selection and inclusion of the selected studies. In this type of paper review articles, the systematization of the process is fundamental. I would like to share my comments with you in this regard:

  1.  I remain fearful of the lack of review of the quality of the texts included. I understand the methodology of obtaining documents from references, but it is important to point out that an essential aspect in systemic or narrative reviews is transparency about the quality criteria used for the inclusion of studies. I suggest again a detailed table or section on these criteria to provide academic certainty to readers.
  2.  I understand that it is valid to do a narrative, non-systematic review, but the inclusion of articles without a structured process makes the study insufficiently transparent. If you do not include a PRISMA, at least include a table explaining the study selection process.
  3. I understand that Figure 2 describes the process of data collection and classification of articles into different categories, but it does not explicitly detail the inclusion and exclusion criteria.
  4. I understand that you refer to some figure regarding the inclusion or exclusion criteria, however, the problem is that these are not explicitly detailed in the text. I request a section on this subject for methodological clarity.

Author Response

Answer to Reviewer 4 Round 3
Many thanks to the authors for their work.
My biggest concern remains the lack of transparency in the process of selection and inclusion of the selected studies. In this type of paper review articles, the systematization of the process is fundamental. I would like to share my comments with you in this regard:
1.     I remain fearful of the lack of review of the quality of the texts included. I understand the methodology of obtaining documents from references, but it is important to point out that an essential aspect in systemic or narrative reviews is transparency about the quality criteria used for the inclusion of studies. I suggest again a detailed table or section on these criteria to provide academic certainty to readers.
2.     I understand that it is valid to do a narrative, non-systematic review, but the inclusion of articles without a structured process makes the study insufficiently transparent. If you do not include a PRISMA, at least include a table explaining the study selection process.
3.    I understand that Figure 2 describes the process of data collection and classification of articles into different categories, but it does not explicitly detail the inclusion and exclusion criteria.
4.    I understand that you refer to some figure regarding the inclusion or exclusion criteria, however, the problem is that these are not explicitly detailed in the text. I request a section on this subject for methodological clarity.
Dear Reviewer 4, 
Thank you very much for your insistence on applying the PRISMA methodology.
After thinking about it, we believe we have understood the changes you would like us to implement. We think these changes will not only improve the transparency and exposition of our work but also increases its reliability in terms of rigor and up-to-date way of reporting.
We added a new subsection between lines [53-60] to clarify the rationale and objectives according to PRISMA 2020 checklist 3 and 4.  
“1.2. Rationale and objectives 
The aim of our review is to synthesize all original research available on the peripheral IGF system in psychiatric disorders (SZ, MDD, BD, BPD, OCD, ASD, and ADHD), with two main objectives. First, to cover the rationale behind these studies. That is, why these studies were done and what they were trying to test. Second, to consider the potential of the IGF system as a biomarker for PDs. Additionally, but to a less extent in terms of extensiveness, there is a third objective in which we cover articles that measured IGF peripheral members and depressive symptoms without a proper MDD diagnosis such as in population-based studies or in concomitant diseases such as fibromyalgia or Parkinson’s disease. We believe these are also relevant approaches to ponder the role of the IGF system in depression.”
We then completely rewrote Section 2 [lines 142-174] to explain the search strategy following PRISMA Guidelines. Additionally, we added Figure 2 (PRISMA flowchart diagram) as you requested, while previous Figure 2 was moved to the Supplementary Material as Figure S1 in case readers may be interested. 
We are aware that some systematic reviews exclude articles not written in English. In our case, only one key article (fulfilling the eligibility criterion of measuring IGF peripheral members in humans with an official diagnosis of PD) was written in a language other than English (Chinese, in reference [269]). We decided not to exclude this article based on language reasons. A few articles were found written in Swedish, which we excluded not due to language, but because they did not meet our eligibility criteria.
We hope that your requests have been completely fulfilled.
Sincerely yours, 
The authors. 

Round 4

Reviewer 4 Report

Comments and Suggestions for Authors

Dear authors
Thank you very much for taking the comments into account. I understand that sometimes these can be annoying, however, they help to improve the article.
For my part, I appreciate that my requests have been adequately attended to.